# Interpretable Vision-Language Survival Analysis with Ordinal Inductive Bias for Computational Pathology

**Pei Liu, Luping Ji✉, Jiaxiang Gou, Bo Fu, Mao Ye**
School of Computer Science and Engineering,
University of Electronic Science and Technology of China

## Abstract

Histopathology Whole-Slide Images (WSIs) provide an important tool to assess cancer prognosis in computational pathology (CPATH). While existing survival analysis (SA) approaches have made exciting progress, they are generally limited to adopting highly-expressive network architectures and only coarse-grained patient-level labels to learn visual prognostic representations from gigapixel WSIs. Such learning paradigm suffers from critical performance bottlenecks, when facing present scarce training data and standard multi-instance learning (MIL) framework in CPATH. To overcome it, this paper, for the first time, proposes a new Vision-Language-based SA (VLSA) paradigm. Concretely, (1) VLSA is driven by pathology VL foundation models. It no longer relies on high-capability networks and shows the advantage of *data efficiency*. (2) In vision-end, VLSA encodes textual prognostic prior and then employs it as *auxiliary signals* to guide the aggregating of visual prognostic features at instance level, thereby compensating for the weak supervision in MIL. Moreover, given the characteristics of SA, we propose i) *ordinal survival prompt learning* to transform continuous survival labels into textual prompts; and ii) *ordinal incidence function* as prediction target to make SA compatible with VL-based prediction. Notably, VLSA's predictions can be interpreted intuitively by our Shapley values-based method. The extensive experiments on five datasets confirm the effectiveness of our scheme. Our VLSA could pave a new way for SA in CPATH by offering weakly-supervised MIL an effective means to learn valuable prognostic clues from gigapixel WSIs. Our source code is available at https://github.com/liupei101/VLSA.

## 1 Introduction

Histopathology whole-slide image (WSI) plays a vital role in cancer diagnosis and treatment (Zarella et al., 2018). It usually covers rich and holistic microscopic information from cellular morphology to tumor micro-environment and then to tissue phenotype (Pati et al., 2022; Chen et al., 2022). As this information can directly reflect tumor progression, digital WSIs are often used in computational pathology (CPATH) to assess cancer patients' prognosis (or survival) (Kather et al., 2019; Song et al., 2023; Jaume et al., 2024). An accurate prognosis assessment is of great significance for enhancing patient management and disease outcomes (Skrede et al., 2020).

However, the survival analysis (SA) of WSI data has always faced two critical challenges. While existing approaches have made exciting progress in overcoming these challenges, they still suffer from *performance bottlenecks* due to the inherent limitations of current SA paradigm. **(1) Scarce training data**. Owing to many real-world factors, *e.g.*, the difficulties in long-term patient follow-up and the concerns about patient privacy, the scale of WSI data for SA has always been limited (Lu et al., 2022; Liu et al., 2024a; Song et al., 2024a), typically on the order of 1,000. Most existing SA models often overlook this and generally seek for network-level solutions to improve the performance, such as adopting highly-expressive modern networks like GNNs (Chen et al., 2021a; Liu

---

✉ Corresponding author: Luping Ji (jiluping@uestc.edu.cn).

et al., 2023b; Shao et al., 2024; Wang et al., 2024) or Transformers (Huang et al., 2021; Hou et al., 2022; Liu et al., 2023a). However, when facing present small WSI data, this way is more likely to cause overfitting in deep learning models (Srivastava et al., 2014), leading to suboptimal prediction performance. **(2) Learning from gigapixel images under weak supervision**. Digital WSIs have extremely-high resolution, *e.g.*, $40,000 \times 40,000$ pixels, so each one is usually processed into a bag of multi-instances for training. With only WSI-level labels, bag-level representations are derived via a *defacto* weakly-supervised multi-instance learning (MIL) framework (Ilse et al., 2018; Liu et al., 2024b). It first i) learns task-specific embeddings from single instances and ii) then aggregates numerous instances (typically 10,000) into one single vector. Current SA schemes follow this way, yet the whole learning process is driven *entirely* by patient-level labels. We argue that such paradigm could lead to inefficient representations, since SA models are only provided with overall patient-level labels while they are not only required to i) learn prognostic embeddings at a fine-grained instance level; but also to ii) select key instances from numerous candidates (Li et al., 2023).

To overcome the limitations of current practices, this paper studies a new SA paradigm for CPATH, called Vision-Language Survival Analysis (**VLSA**). Concretely, we find that recent VL foundation models (VLMs) in CPATH, *e.g.*, CONCH (Lu et al., 2024), offer potential means to mitigate the challenges above. **First**, these VLMs are pretrained on large-scale image-text pairs by task-agnostic objectives. They show surprisingly-good performance in terms of ***data efficiency***, especially in zero-shot transfer, as highlighted in Lu et al. (2024); Javed et al. (2024). This is promising for alleviating the challenge of scarce training data. **Second**, VL contrastive pretraining aligns image and language in latent embedding space (Radford et al., 2021), which enables language to act as "prompt" for vision tasks. This implies that, with prior knowledge, language is likely to provide ***auxiliary signals*** to boost learning efficiency. Such additional signals could be particularly helpful to improve the weak supervision in MIL. Despite all these appealing merits, VLM-driven SA has not yet been studied. We believe there are two main reasons: i) the powerful VLMs for CPATH are developed just recently; ii) different from classification, there remains a gap for SA to be adapted to VLMs.

Based on these insights, this paper first proposes a VLSA framework for CPATH. Different from existing VL-based schemes, VLSA comprises four core designs as follows. **(1) Vision-end**: it leverages language-encoded prognostic priors to guide multi-instance aggregation, producing multi-level visual presentations. **(2) Language-end**: considering the intrinsic ordinality in survival risks, we propose ordinal survival prompt learning to encode continuous time-to-event labels into textual prompts. **(3) Prediction and optimization**: To make SA compatible with VL-based prediction, we take incidence function as prediction target and introduce an ordinal inductive bias into it for regularization in optimization. **(4) Prediction interpretation**: with the classic Shapley values from game theory, individual prognostic risk could be interpreted from an intuitive perspective—descriptive language. The extensive experiments on five datasets verify the effectiveness of our scheme. Notably, our comparative experiments and analyses suggest that VLSA could pave a new way for SA in CPATH by offering weakly-supervised MIL an effective means to learn valuable prognostic clues from gigapixel WSIs. We summarize the main contributions of this paper as follows:

• An interpretable Vision-Language Survival Analysis framework (VLSA) is proposed for computational pathology. To our knowledge, it is the first work that studies VL-based SA in CPATH.

• Language-encoded prognostic prior is proposed to boost weakly-supervised WSI representation learning. In view of the characteristics of SA, two ordinal inductive bias terms, *i.e.*, ordinal survival prompts and ordinal incidence function, are introduced into VLSA to enhance the performance.

• To assess model performance more rigorously, this paper conducts extensive experiments and adopts multiple metrics for discrimination and calibration evaluation. Empirical results show that VLSA could often obtain new state-of-the-art performances with less computation costs.

## 2  RELATED WORK

**Survival Analysis on WSIs** One key challenge of WSI survival analysis is effectively learning global prognostic representations from multi-instances with only patient-level survival labels. To this end, various data-driven approaches are studied. They can be roughly grouped into three categories based on the structure assumed for instances: i) *cluster-based* (Yao et al., 2020; Shao et al., 2021a), ii) *graph-based* (Chen et al., 2021a; Liu et al., 2023b; Shao et al., 2024; Wang et al., 2024),

and iii) *sequence-based* (Huang et al., 2021; Hou et al., 2022; Liu et al., 2023a). These structures are employed to learn non-local prognostic embeddings at cluster, node, or global token level. However, most studies in this field currently focus more on pure vision or vision-gene representation learning (Zhou & Chen, 2023; Jaume et al., 2024). The vision-language learning paradigm, which has witnessed remarkable successes in recent years (Zhang et al., 2024a), remains under-studied.

**Vision-Language Models for Computation Pathology** Since the inception of CLIP (Radford et al., 2021), Vision-Language models (VLMs) have attracted considerable attention and are applied to a wide range of fields. CPATH is one of them. Related studies or applications cover two main directions. **(1) Foundational VLMs for pathology**, *e.g.*, PLIP (Huang et al., 2023), QUILTNET (Ikezogwo et al., 2023), PathAlign (Ahmed et al., 2024), and CONCH (Lu et al., 2024). These models are usually pretrained on large-scale pathology image-text pairs by VL contrastive learning. Their pretrained encoders often perform significantly better than CLIP in pathology-related tasks, laying a solid foundation for CPATH. **(2) VL-based WSI classification**. Recent efforts involving VLMs are mainly seen in WSI classification (Qu et al., 2023; Li et al., 2024; Shi et al., 2024). Based on pretrained VL encoders, they further improve the performance by large margins through fine-grained text guidance or multi-scale image features. Given the good foundation and strong potential of pathology VLMs, a further study on VL-based survival analysis is strongly anticipated.

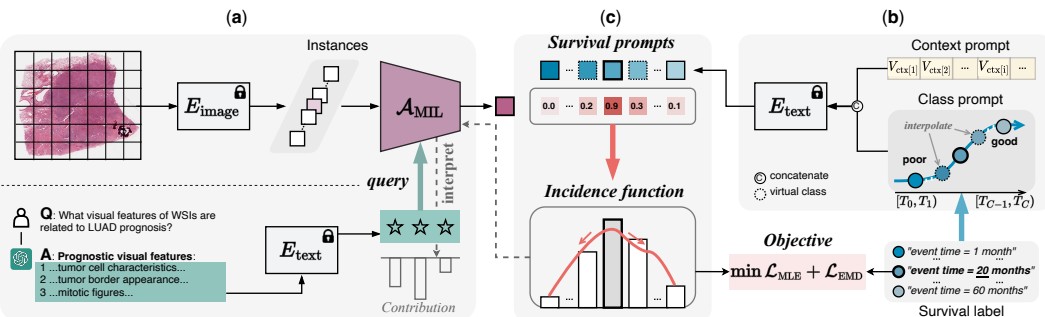

Figure 1: Overview of Vision-Language Survival Analysis (VLSA). (a) WSI representation learning with language-encoded prognostic priors (Section 3.1). (b) Ordinal survival prompt learning (Section 3.2). (c) Prediction of ordinal incidence function (Section 3.3). The survival prediction of VLSA can be interpreted by quantifying each prognostic prior's contribution to risk (Section 3.5).

## 3 METHOD

This section presents our VLSA, a vision-language-based survival analysis framework (Figure 1) for CPATH. We first introduce its three key parts: i) WSI representation learning with language-encoded prognostic priors (Figure 1(a), Section 3.1); ii) Ordinal survival prompt learning (Figure 1(b), Section 3.2); and iii) ordinal incidence function prediction (Figure 1(c), Section 3.3). Then, we give our overall training objectives in Section 3.4 and explain how to interpret the survival prediction of VLSA in Section 3.5. The preliminaries can be found in Appendix A.

### 3.1 WSI REPRESENTATION LEARNING WITH LANGUAGE-ENCODED PROGNOSTIC PRIORS

Given the WSIs of one patient, we denote their processed instances by a bag $\boldsymbol{X} = [\boldsymbol{x}_1, \ldots, \boldsymbol{x}_K]^\mathsf{T} \in \mathbb{R}^{K \times D}$, where $\boldsymbol{x}_k \in \mathbb{R}^D$ represents the $k$-th instance feature vector. Instance features are extracted by the image encoder of a VLM, written as $E_{\text{image}}$. To derive WSI representations, we introduce language-encoded prognostic priors to encourage MIL models to distill valuable prognostic clues from numerous instances under weak supervision, as shown in Figure 1(a).

**Language-Encoded Prognostic Priors** Concretely, for a specific cancer disease, we first obtain the textual descriptions about its critical prognostic features visible in histopathology WSIs, written as $\mathcal{T}_{\text{prog}} = \{\mathcal{T}_{\text{prog}}^1, \ldots, \mathcal{T}_{\text{prog}}^M\}$, using a LLM (large language model), GPT-4o. Then, we encode $\mathcal{T}_{\text{prog}}$ into textual features with the text encoder of the VLM, denoted by $E_{\text{text}}$. We further fine-tune these

textual features using a group of learnable parameters, $\boldsymbol{T}_{\text{prog}} \in \mathbb{R}^{M \times D}$. We write the result as

$$\boldsymbol{P} = E_{\text{text}}(\mathcal{T}_{\text{prog}}) + \boldsymbol{T}_{\text{prog}}. \tag{1}$$

$\boldsymbol{P} = [\boldsymbol{p}_1, \ldots, \boldsymbol{p}_M]^{\mathsf{T}} \in \mathbb{R}^{M \times D}$ represents language-encoded prognostic priors, where $\boldsymbol{p}_m \in \mathbb{R}^D$ is the $m$-th prior item and $M$ is the total number of prior items.

**WSI Representation** In the MIL aggregator $\mathcal{A}_{\text{MIL}}$, any prior $\boldsymbol{p}_m$ is utilized as a query to aggregate key instances by similarity matching and weighted averaging, producing *one-level* WSI representation ($\boldsymbol{f}_m \in \mathbb{R}^D$) corresponding to the $m$-th prognostic description ($\mathcal{T}_{\text{prog}}^m$). Finally, multi-level visual representations, $\boldsymbol{F} = [\boldsymbol{f}_1, \ldots, \boldsymbol{f}_M]^{\mathsf{T}} \in \mathbb{R}^{M \times D}$, are averaged and passed through a linear neural network layer to obtain the final WSI representation $\boldsymbol{f}_{\text{image}} \in \mathbb{R}^D$. In summary, there are

$$\boldsymbol{f}_m = \sum_{k=1}^{K} \boldsymbol{x}_k \cdot \frac{\exp\left(\alpha \cdot \cos(\boldsymbol{p}_m, \boldsymbol{x}_k)\right)}{\sum_{i=1}^{K} \exp\left(\alpha \cdot \cos(\boldsymbol{p}_m, \boldsymbol{x}_i)\right)}, \tag{2}$$

$$\boldsymbol{f}_{\text{image}} = \text{Linear}\left(\text{mean}(\boldsymbol{F})\right) = \text{Linear}\left(\text{mean}(\{\boldsymbol{f}_m^{\mathsf{T}}\}_{m=1}^{M})\right), \tag{3}$$

where $\alpha$ is a fixed temperature hyper-parameter, $\cos(\cdot, \cdot)$ is an operator that calculates cosine similarity, and $\text{Linear}(\cdot)$ indicates a simple linear layer with learnable weights and biases.

**Justification (1) Prognostic prior knowledge**: from Eq. (2), we can find that, the instances whose visual features are well-aligned with the prognostic description, would be selected and aggregated into one visual representation. This implies that, with the well-aligned VL embedding space provided by VLMs, prognostic prior could offer additional helpful signals and compensate for the inherent weak supervision in MIL. **(2) Multi-level visual representations**: they are designed to provide sufficient prognostic clues because there are usually more than one prognostic markers in WSIs, *e.g.*, nuclear atypia, perineural invasion, mitotic activity, etc. Moreover, multi-level prognostic clues could enable us to decouple and interpret individual survival predictions from intuitive perspectives (*i.e.*, descriptive language). Our interpretation method is elaborated in Section 3.5. **(3) Difference from multi-modal representation learning**: our textual description of prognostic prior is utilized to provide the weight of each instance for MIL aggregation and mainly plays a guidance role. The final representation in vision-end comes from WSI features, not a multi-modal representation fusing text and vision features. This is different from current multi-modal SA models (Chen et al., 2021b; Xu & Chen, 2023; Zhou & Chen, 2023; Xiong et al., 2024; Song et al., 2024b).

## 3.2 ORDINAL SURVIVAL PROMPT LEARNING

Different from classification, SA provides continuous time-to-event labels, *i.e.*, $\mathbb{Y} = \{t, \delta\}$, where $t$ is the last follow-up time and $\delta \in \{0, 1\}$ is an event indicator at $t$. Due to this intrinsic difference, obtaining SA prompts would be more challenging, compared with VL-based classification. To address it, we propose *ordinal survival prompt learning* that encodes time-to-event labels into textual survival prompts for VL-based prediction, as depicted in Figure 1(b).

**Time Discretization** Following the convention of discrete-time SA (Haider et al., 2020), we first uniformly discretize time into a set of non-overlapping bins, $[T_0, T_1), [T_1, T_2), \cdots, [T_{C-1}, T_C)$. For any $t \in [T_{c-1}, T_c)$, we assign its corresponding time-discrete label $c$, where $c \in [1, C]$. These time bins are equal in length; $C$ is determined by $\sqrt{N_e}$, where $N_e$ is the number of patients with $\delta = 1$. Refer to Appendix C.5 for more details and Appendix D.6 for experiments on these settings.

**Ordinal Survival Prompt** Textual prompt usually consists of context and class label in VLM-driven prediction. Thus, **(1) for context prompt**, we follow CoOp (Zhou et al., 2022) to employ learnable parameters, $\boldsymbol{V}_{\text{ctx}} \in \mathbb{R}^{L_{\text{ctx}} \times D_{\text{emb}}}$, to optimize the context of survival prompts, where $L_{\text{ctx}}$ is the length of context tokens and $D_{\text{emb}}$ is the dimension of token embedding. **(2) For class prompts**, we maintain $B$ learnable *base class prompts*, $\{\boldsymbol{V}_{\text{cls}}^{\lambda_1}, \cdots, \boldsymbol{V}_{\text{cls}}^{\lambda_B}\}$, where $\boldsymbol{V}_{\text{cls}}^{\lambda_b} \in \mathbb{R}^{L_{\text{cls}} \times D_{\text{emb}}}$ ($b \in [1, B]$) is the token embedding of the $b$-th base class prompt and $L_{\text{cls}}$ is the maximum length of class tokens. These base class prompts are initialized using common prognosis risk descriptions, *e.g.*, {"very poor", "moderate", "very good"}. Then, considering the ordinality between survival classes $\{1, 2, \cdots, C\}$, we obtain the remaining class prompts by *interpolating* virtual points between base class prompts, inspired by Li et al. (2022). The $c$-th class prompt is written as follows:

$$\boldsymbol{V}_{\text{cls}}^c = \sum_{b=1}^{B} \boldsymbol{V}_{\text{cls}}^{\lambda_b} \cdot \frac{W(\boldsymbol{D}_{c,b})}{\sum_{i=1}^{B} W(\boldsymbol{D}_{c,i})}, \tag{4}$$

where $D_{c,b}$ is an element of matrix $D \in \mathbb{R}^{C \times B}$ representing the ordering distance between the $c$-th class prompt and the $b$-th base class prompt, and $W(\cdot)$ is a function determining the interpolation weight based on distance. A closer distance usually indicates a greater interpolation weight. $D_{c,b}$ could be simply set by $D_{c,b} = |(c-1) - (b-1) \cdot (C-1)/(B-1)|$. For linear interpolation, $W(D_{c,b}) = 1 - \frac{D_{c,b}}{C-1}$. **(3) For survival prompts**, each class prompt is concatenated with $V_{\text{ctx}}$ at the token level, followed by passing through frozen $E_{\text{text}}$ to produce the final ordinal survival prompts, denoted as $F_{\text{text}} = [f_{\text{text}}^1, \ldots, f_{\text{text}}^C]^\top \in \mathbb{R}^{C \times D}$. Namely, there is

$$f_{\text{text}}^c = E_{\text{text}}\big([V_{\text{ctx}} | V_{\text{cls}}^c]\big) \ \ \forall \, c \in [1, C]. \tag{5}$$

**Justification (1) Ordinal inductive bias**: in common sense, as the length of survival time decreases, the corresponding death risk becomes increasingly high. This implies the ordinality between survival classes; such inductive bias could be considered when designing prompts for SA. More explanations and discussions could be found in Appendix B. **(2) Class prompt interpolation**: on one hand, most textual encoders are lacking in the sensitivity to numbers' ordinality (Thawani et al., 2021; Paiss et al., 2023), so it may be unsuitable to directly adopt numerical classes as textual prompts. On the other hand, when the number of classes becomes large, *e.g.*, $C = 10$, it would be intractable to manually design fine-grained prognosis risk descriptions at $C$ different levels. In view of these obstacles, we adopt a few base prompts ($\lambda \leq 4$) and then utilize an interpolation-based strategy (Li et al., 2022) to preserve the ordinality between survival prompts.

## 3.3 Ordinal Incidence Function Prediction

The prediction target of VL-based models is usually class probability, calculated based on the similarity measures between image features and a group of textual class prompts. To make survival prediction compatible with this way, we propose to take individual *incidence function* as prediction target and introduce another ordinal inductive bias term into it, as shown in Figure 1(c).

**Survival Prediction with Incidence Function** Similar to the manner of VL-based classification, we calculate SA prediction, denoted by $\hat{y} = [\hat{y}_1, \cdots, \hat{y}_C]$, as follows:

$$\hat{y}_c = \frac{\exp\big(\tau \cdot \cos(f_{\text{image}}, f_{\text{text}}^c)\big)}{\sum_{i=1}^C \exp\big(\tau \cdot \cos(f_{\text{image}}, f_{\text{text}}^i)\big)} \ \ \forall \, c \in [1, C], \tag{6}$$

where $\tau$ is a temperature parameter optimized in training. From a conventional classification perspective, $\hat{y}_c$ given by Eq. (6) can be cast as the probability that an event first occurs at time $c$; $\hat{y}$ is thus a probability distribution on the first hitting time. Such prediction is closely associated with the concept of *incidence function* (IF) (Fine & Gray, 1999) and the first hitting time model (Lee & Whitmore, 2006; Lee et al., 2018) in traditional SA. Therefore, we call Eq. (6) the prediction of individual IF and leverage IF tools for the interested task. By definition, *cumulative IF* is $\text{CIF}(c) = \sum_{i=1}^c \hat{y}_c$. *Survival function* (surviving past time $c$) can be written as $\hat{S}(c) = 1 - \text{CIF}(c) = 1 - \sum_{i=1}^c \hat{y}_i$.

**Ordinal Inductive Bias in Incidence Function** As $\hat{y}$ is a probability distributed over ordered survival classes, we consider an ordinality constraint for it (see Appendix B for further elucidations). Concretely, given that $c$ is the survival class with the largest probability, we assume there is a consecutive decline in probability for those classes away from $c$, as shown in Figure 1(c). Note that our assumption is concerned with the event that first occurs, as $\hat{y}$ is defined as the probability distribution on the first hitting time. To impose the ordinality constraint on $\hat{y}$, at first we adopt *Earth Mover's Distance* (EMD) (Levina & Bickel, 2001) as the measure to quantify the distance between $\hat{y}$ and $y$ (ground truth distribution), written as

$$\text{EMD}(\hat{y}, y) = \big(\frac{1}{C}\big)^{\frac{1}{l}} ||\text{CDF}(\hat{y}) - \text{CDF}(y)||_l, \tag{7}$$

where $\text{CDF}(\cdot)$ means cumulative distribution function. EMD is a measure aware of distribution ordinality as it considers the geometry property of distribution in distance measurement and it is smaller when the geometry (shape) of two distributions is closer. Please refer to Appendix A.3 for detailed explanations. Furthermore, we adopt a squared EMD objective (Hou et al., 2017) to regularize $\hat{y}$ in model optimization:

$$\mathcal{L}_{\text{EMD}} = ||\text{CDF}(\hat{y}) - \text{CDF}(y(c, \delta))||_2^2. \tag{8}$$

$\boldsymbol{y}(c, \delta)$ is defined as follows. For the patient with event, $\boldsymbol{y}(c, \delta = 1) = \text{Softmax}\big(\tau' \cdot (2 \cdot \mathbb{I}_{i=c} - 1)\big)$. For the patient censored at time $c$, we only know that its actual time-to-event is not less than $c$, so $\boldsymbol{y}(c, \delta = 0) = \text{Softmax}\big(\tau' \cdot (2 \cdot \mathbb{I}_{i \geq c} - 1)\big)$. $\mathbb{I}_i \in \{0, 1\}^C$ is an indicator with element '1' at index $i$. The value of $\tau'$ is from $\tau$ but not involved in optimization. We further analysis and discuss the ordinal inductive biases introduced into $\boldsymbol{F}_{\text{text}}$ and $\hat{\boldsymbol{y}}$; the details can be found in Appendix B.

## 3.4 OVERALL TRAINING OBJECTIVES

For any patient with survival label $\mathbb{Y} = \{t, \delta\}$, we denote its time-discrete label by $\mathbb{Y}_{\text{d}} = \{c, \delta\}$ (Section 3.2). To optimize the prediction of individual incidence function, a maximum likelihood estimation (MLE)-based objective (Tutz & Schmid, 2016) is utilized in training:

$$\mathcal{L}_{\text{MLE}} = -\big[\delta \cdot \log(\hat{y}_c) + (1 - \delta) \cdot \log(1 - \sum_{i=1}^{c-1} \hat{y}_i)\big]. \qquad (9)$$

For censored patients ($\delta = 0$), $\mathcal{L}_{\text{MLE}}$ minimizes $\sum_{i=1}^{c-1} \hat{y}_i$, *i.e.*, the probability that the event of interest first occurs at discrete bins $1, 2, \ldots$, or $c - 1$, according to the definition of $\hat{y}_i$ in Eq. (6). Furthermore, with the $\mathcal{L}_{\text{EMD}}$ given in Eq. (8), our overall training objective is minimizing

$$\mathcal{L} = \mathcal{L}_{\text{MLE}} + \beta \cdot \mathcal{L}_{\text{EMD}}, \qquad (10)$$

where $\beta \geq 0$ is a hyper-parameter that modulates the weight of $\mathcal{L}_{\text{EMD}}$.

## 3.5 PREDICTION INTERPRETATION

Understanding how a model makes predictions is crucial for reliable decision-making, especially in medical domains. We thereby study the prediction behavior of VLSA and propose an interpretation method based on the classic Shapley values (Appendix A.4) from cooperative game theory.

Concretely, we interpret the survival prediction of VLSA through *language-encoded prognostic priors*, *i.e.*, $\boldsymbol{P} = [\boldsymbol{p}_1, \ldots, \boldsymbol{p}_M]^\mathsf{T}$ described in Section 3.1, because i) each prior intuitively describes a certain prognostic visual feature in WSIs and ii) these language-encoded priors play an important role in learning valuable prognostic clues and making final survival predictions. However, each prior item's contribution to risk prediction cannot be computed directly since it is not linearly correlated with $\hat{\boldsymbol{y}}$. Thus, based on a principled framework for model interpretation—Shapley value (Shapley, 1953; Lundberg, 2017), we calculate the contribution of each prior item by

$$\phi_m(f_{\text{risk}}) = \sum_{\mathbb{Z} \subseteq \mathbb{P}} \frac{|\mathbb{Z}|!(M - |\mathbb{Z}| - 1)!}{M!} \left[f_{\text{risk}}(\mathbb{Z}, \boldsymbol{X}) - f_{\text{risk}}(\mathbb{Z} \setminus \{\boldsymbol{p}_m\}, \boldsymbol{X})\right], \qquad (11)$$

where $\mathbb{P}$ is a set of prognostic priors $\{\boldsymbol{p}_1, \ldots, \boldsymbol{p}_M\}$. The function $f_{\text{risk}}(\mathbb{Z}, \boldsymbol{X})$ outputs the risk prediction given the prognostic prior subset $\mathbb{Z}$ and the WSI features $\boldsymbol{X}$ as inputs. Specifically, we i) calculate $\boldsymbol{f}_{\text{image}}$ using a subset of $\boldsymbol{F}$ corresponding to $\mathbb{Z}$, ii) use it to obtain the IF prediction $\hat{\boldsymbol{y}}$ through Eq. (6), and iii) finally derive the risk via the summation of CIF: $\hat{R} = \sum_{i=1}^{C} \text{CIF}(i)$.

## 4 EXPERIMENTS

### 4.1 EXPERIMENTAL SETUP

**Datasets** Following Chen et al. (2021a), five publicly-available datasets from TCGA (The Cancer Genome Atlas) are used in experiments: **BLCA** (bladder urothelial carcinoma) ($N = 373$), **BRCA** (breast invasive carcinoma) ($N = 956$), **GBMLGG** (glioblastoma & lower grade glioma) ($N = 569$), **LUAD** (lung adenocarcinoma) ($N = 453$), and **UCEC** (uterine corpus endometrial carcinoma) ($N = 480$). These datasets cover five different cancer types, a total number of 2,831 patients, and 3,530 diagnostic gigapixel WSIs. Overall Survival (OS), which only occurs once, is set as the event of interest in this paper, following Chen et al. (2021a). We adopt a standard CLAM tool (Lu et al., 2021) to preprocess all WSIs into multi-instance bags for training. **CONCH** (Lu et al., 2024), as a state-of-the-art VLM recently developed for CPATH, is utilized to provide powerful encoders $E_{\text{image}}$ and $E_{\text{text}}$. More details on datasets can be found in Appendix C.1.

**Baselines (1) Vision-only (V)** methods contain ABMIL (Ilse et al., 2018), TransMIL (Shao et al., 2021b), ILRA (Xiang & Zhang, 2023), $R^2$T-MIL (Tang et al., 2024), DeepAttnMISL (Yao et al., 2020), and Patch-GCN (Chen et al., 2021a). **(2) Vision-language (VL)** methods are MI-Zero (Lu et al., 2023), ABMIL$_{Prompt}$, CoOp (Zhou et al., 2022), and OrdinalCLIP (Li et al., 2022). **MI-Zero** is a VL-based zero-shot approach for classification; we adapt it for SA to assess the lower bound performance of VLM on new SA tasks. **ABMIL$_{Prompt}$** is another ABMIL baseline with naive (not learnable) textual prompts for prediction in language-end. As **CoOp** and **OrdinalCLIP** are not proposed for gigapixel WSIs, we adapt them for comparison: i) in vision-end, the classic attention-based MIL, *i.e.*, ABMIL, is used for instance aggregation; ii) in language-end, CoOp optimizes both context and class prompts without ordinal inductive bias, while OrdinalCLIP improves CoOp using ordinal class prompts. As a result, OrdinalCLIP and our VLSA share the approach of generating ordinal survival prompts. Note that all methods use *the same $E_{image}$* from CONCH; VL-based methods involve an additional VL projection layer. More details are provided in Appendix C.2 and C.3.

**Evaluation Metrics** To evaluate models more comprehensively, we follow traditional SA (Qi et al., 2023a;b) to adopt multiple metrics. (1) Concordance index (**CI**), as a prevalent metric in SA, assesses models' discrimination power. (2) Mean absolute error (**MAE**) gives the absolute error of model's prediction on time-to-event. (3) Distribution calibration (**D-cal**) (Haider et al., 2020) is a statistical test to evaluate models' ability in calibrating survival distribution prediction; we mainly check if there is no significant difference in distribution between ground truth and prediction ($p > 0.05$). For any method, we adopt 5-fold cross-validation to evaluate its performance and report the average on 5 folds. Please refer to Appendix C.4 for more details on evaluation metrics.

## 4.2 COMPARISON WITH BASELINES

As shown in Table 1, there are three key observations. **(1) Our VLSA** achieves new state-of-the-art performances and it could often perform better than most baselines by a large margin. In terms of average CI, our VLSA (average CI = 0.695) leads runner-up by 2.3%. Moreover, it often predicts well-calibrated survival distributions, reflected by D-cal Count. **(2) For VL-based methods**, CoOp and OrdinalCLIP are competitive with other baselines, although they adopt the ABMIL network much simpler than others. **(3) Original CONCH encoders** (with MI-Zero$_{Surv}$) often cannot discriminate risks correctly in SA (average CI = 0.54) without supervised fine-tuning (SFT), especially on GBMLGG. This suggests the necessity of adapting CONCH to SA through SFT.

Table 1: **Main comparative results**. D-cal count is the number of datasets with $p > 0.05$ in D-cal test. † MI-Zero is adapted for SA. Best performance in **bold**, second best underlined.

| | Method | TCGA | | | | | Average | | D-cal |
|---|---|---|---|---|---|---|---|---|---|
| | | BLCA | BRCA | GBMLGG | LUAD | UCEC | CI | MAE | Count |
| **V** | ABMIL | 0.5581 (± 0.031) | 0.5825 (± 0.035) | 0.7935 (± 0.032) | 0.6121 (± 0.050) | 0.6667 (± 0.033) | 0.6426 | 29.83 | 4 |
| | TransMIL | 0.5885 (± 0.055) | 0.6140 (± 0.060) | 0.7956 (± 0.015) | 0.5708 (± 0.050) | 0.6380 (± 0.067) | 0.6414 | 30.43 | 5 |
| | ILRA | 0.5549 (± 0.053) | 0.5705 (± 0.067) | 0.7742 (± 0.014) | 0.5179 (± 0.081) | 0.6503 (± 0.064) | 0.6136 | 32.59 | 4 |
| | $R^2$T-MIL | 0.5775 (± 0.024) | 0.5473 (± 0.095) | 0.7757 (± 0.024) | 0.5711 (± 0.076) | 0.6510 (± 0.087) | 0.6245 | 32.54 | 4 |
| | DeepAttnMISL | 0.5646 (± 0.035) | 0.5346 (± 0.036) | 0.6750 (± 0.048) | 0.4678 (± 0.039) | 0.6259 (± 0.085) | 0.5736 | 52.10 | 5 |
| | Patch-GCN | 0.6124 (± 0.031) | 0.6375 (± 0.033) | 0.7999 (± 0.021) | 0.5922 (± 0.053) | 0.7212 (± 0.025) | 0.6726 | 26.70 | 2 |
| **VL** | MI-Zero$_{Surv}$ [†] | 0.5541 (± 0.034) | 0.5788 (± 0.028) | 0.3842 (± 0.063) | 0.5209 (± 0.049) | 0.6623 (± 0.059) | 0.5400 | 25.63 | 0 |
| | ABMIL$_{Prompt}$ | 0.5717 (± 0.035) | 0.6215 (± 0.084) | 0.7825 (± 0.020) | 0.5984 (± 0.052) | 0.6762 (± 0.063) | 0.6500 | 25.68 | 4 |
| | CoOp | 0.5971 (± 0.033) | 0.5994 (± 0.086) | 0.7853 (± 0.015) | 0.5750 (± 0.064) | 0.6840 (± 0.070) | 0.6482 | 28.70 | 5 |
| | OrdinalCLIP | 0.6037 (± 0.043) | 0.6202 (± 0.046) | 0.7893 (± 0.018) | 0.6053 (± 0.065) | 0.6836 (± 0.036) | 0.6604 | 28.01 | 5 |
| | **VLSA (ours)** | **0.6176** (± 0.025) | **0.6652** (± 0.057) | **0.8002** (± 0.010) | **0.6370** (± 0.027) | **0.7571** (± 0.045) | **0.6954** | **25.15** | 5 |

## 4.3 ANALYSIS OF LANGUAGE-ENCODED PROGNOSTIC PRIORS

**Ablation Study** To verify the effectiveness of our instance aggregation method, we compare three baselines: i) *attention*, classical attention-based aggregation; ii) *learnable prototypes*, using the same number of learnable vectors without prior knowledge encoding; iii) *prognostic texts*, *i.e.*, only encoding textual priors, without $T_{\mathrm{prog}}$. Their results are shown in Table 2. Our main findings are as follows. **(1) Language-encoded prognostic priors play a particularly important role** in the performance improvement of VLSA. After incorporating prognostic priors, VLSA obtains improvements ranging from 1.3% to 6.5% in terms of CI, and an improvement of **3.5%** in average CI. **(2) Fine-tuning with $T_{\mathrm{prog}}$** boosts the performance with a slight overall improvement (0.4%).

Table 2: **Ablation study on our method for instance aggregation**. FT is fine-tuning with $T_{\mathrm{prog}}$.

| Aggregation method | TCGA | | | | | Average | | D-cal |
| | BLCA | BRCA | GBMLGG | LUAD | UCEC | CI | MAE | Count |
|---|---|---|---|---|---|---|---|---|
| Attention | 0.6083 (± 0.047) | 0.6180 (± 0.046) | 0.7908 (± 0.017) | 0.6048 (± 0.063) | 0.6908 (± 0.035) | 0.6625 | 26.78 | 5 |
| Learnable prototypes | 0.5872 (± 0.048) | 0.6201 (± 0.050) | 0.7853 (± 0.013) | 0.6061 (± 0.053) | 0.6845 (± 0.052) | 0.6566 | 26.76 | 4 |
| Prognostic texts | 0.6159 (± 0.025) | 0.6614 (± 0.047) | 0.7985 (± 0.009) | 0.6314 (± 0.028) | 0.7491 (± 0.049) | 0.6912 | **25.05** | 4 |
| Prognostic texts + FT | **0.6176** (± 0.025) | **0.6652** (± 0.057) | **0.8002** (± 0.010) | **0.6370** (± 0.027) | **0.7571** (± 0.045) | **0.6954** | 25.15 | 5 |

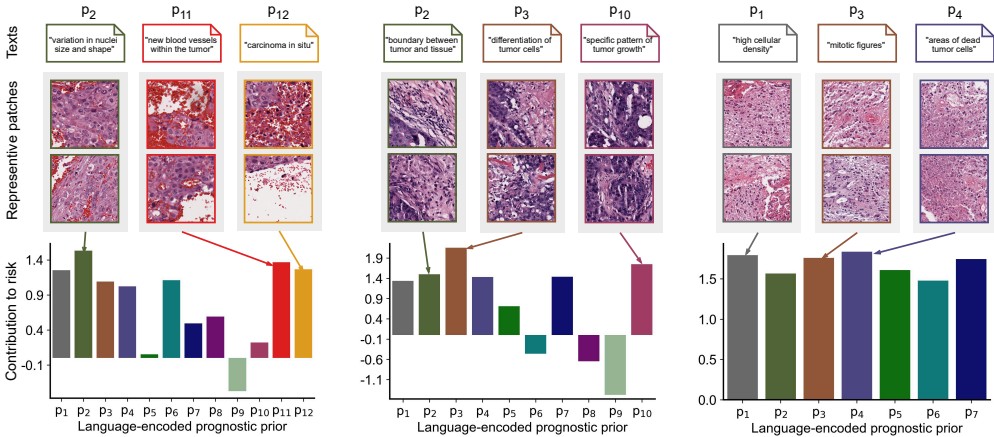

Figure 2: **Interpreting the survival prediction of VLSA via language-encoded prognostic priors**. Top row shows language descriptions (simplified for better view) about prognostic visual features in WSIs. Detailed texts are provided in Appendix C.3. Middle row gives the most representative patches corresponding to each prognostic text. Last row presents each prognostic prior's contribution to risk. We mainly examine the top three language priors in terms of contribution. The three examples are from the first three datasets. More results are shown in Appendix D.2.

**Visualization** We conduct a case study to further examine the role of language-encoded prognostic priors in our VLSA. As shown in Figure 2, we observe that valuable visual features can be highlighted by prognostic texts in most cases. These visual features, *e.g.*, "differentiation of tumor cells" and "high cellular density", could often provide important clues for cancer prognosis. We note that, such desirable behavior is largely attributed to the intrinsic property of VLMs, *i.e.*, VLMs offer *a well-aligned vision-language embedding space* to enable us to manipulate visual features with language prior. We believe that this property would be particularly important to the MIL in which a model is usually required to *aggregate numerous instances under weak supervision*.

## 4.4 ANALYSIS OF ORDINAL INDUCTIVE BIASES

**Ablation Study** We conduct ablation studies to verify the effectiveness of our two ordinal inductive bias terms when the language-encoded prognostic priors $P$ are used or not used in VLSA. Its

results are shown in Table 3. We have two main observations from these results. **(1) Ordinal survival prompt**: when $P$ is not used in VLSA, the improvements by imposing ordinality on survival prompts are 1.2% and 1.3% in average CI; when $P$ is used, the improvements are relatively large (3.4% and 3.6%) on BRCA and are often marginal on the other four datasets. **(2) Ordinal probability**: also called ordinal IF, it always shows consistent improvements in terms of average MAE, ranging from 1.2 to 1.7, when $P$ is used in VLSA or not. More discussions on the two inductive bias terms are provided in Appendix B.

Table 3: **Ablation study on the two ordinal inductive bias terms in VLSA**.

| Ordinality | | TCGA | | | | | Average | | D-cal |
| Prompt | Proba. | BLCA | BRCA | GBMLGG | LUAD | UCEC | CI | MAE | Count |
|---|---|---|---|---|---|---|---|---|---|
| *- w/o language-encoded prognostic priors* | | | | | | | | | |
| | | 0.5971 (± 0.033) | 0.5994 (± 0.086) | 0.7853 (± 0.015) | 0.5750 (± 0.064) | 0.6840 (± 0.070) | 0.6482 | 28.70 | 5 |
| ✓ | | 0.6037 (± 0.043) | **0.6202** (± 0.046) | 0.7893 (± 0.018) | **0.6053** (± 0.065) | 0.6836 (± 0.036) | 0.6604 | 28.01 | 5 |
| | ✓ | 0.5997 (± 0.033) | 0.6049 (± 0.104) | 0.7846 (± 0.016) | 0.5818 (± 0.056) | 0.6769 (± 0.065) | 0.6496 | 26.99 | 3 |
| ✓ | ✓ | **0.6083** (± 0.047) | 0.6180 (± 0.046) | **0.7908** (± 0.017) | 0.6048 (± 0.063) | **0.6908** (± 0.035) | **0.6625** | **26.78** | 5 |
| *- w/ language-encoded prognostic priors* | | | | | | | | | |
| | | 0.6128 (± 0.028) | 0.6304 (± 0.065) | 0.7927 (± 0.015) | 0.6351 (± 0.041) | **0.7606** (± 0.037) | 0.6863 | 26.76 | 5 |
| ✓ | | 0.6145 (± 0.024) | 0.6643 (± 0.055) | 0.7973 (± 0.009) | 0.6368 (± 0.034) | 0.7478 (± 0.060) | 0.6921 | 26.53 | 5 |
| | ✓ | 0.6138 (± 0.022) | 0.6293 (± 0.067) | 0.7975 (± 0.013) | 0.6361 (± 0.036) | 0.7592 (± 0.036) | 0.6872 | 25.23 | 5 |
| ✓ | ✓ | **0.6176** (± 0.025) | **0.6652** (± 0.057) | **0.8002** (± 0.010) | **0.6370** (± 0.027) | 0.7571 (± 0.045) | **0.6954** | **25.15** | 5 |

**Discussion** We note that there are two possible reasons for the marginal CI gain of two ordinal inductive biases when $P$ is used in VLSA. **(1) Ordinal incidence function**, often helps to improve MAE, not CI, observed from the results of Table 3. **(2) The effect of the ordinal inductive bias imposed on survival prompts** may be pared down, when the learned visual features become more discriminative (due to the use of $P$). This is because our survival prompts are learnable and they can be tuned better in optimization to align with the discriminative visual features. As a result, the effect of the ordinal bias term for prompts could be alleviated. We note that, although there is a marginal gain in CI when $P$ is present, the effect of ordinal inductive biases on VLSA is often notable in the other terms, such as MAE and the ordinal Acc in Figure 3(a).

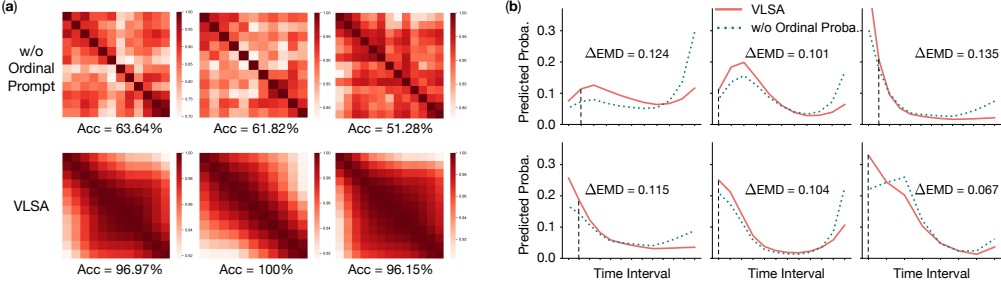

Figure 3: **Ordinality visualization**. (a) Heatmap of the similarity between any two learned survival prompts. Its horizontal axis places the first prompt to the last prompt from left to right; its vertical one does so from top to bottom. Acc is the accuracy of prompt ranking. The results are from the first three datasets. More results are shown in Appendix D.2. (b) Predictive probability (Proba.) w/o and w/ ordinality. The patients from test set are used for prediction. $\Delta$EMD is equal to $\text{EMD}(\boldsymbol{y}, \hat{\boldsymbol{y}}_{\text{w/o}}) - \text{EMD}(\boldsymbol{y}, \hat{\boldsymbol{y}}_{\text{w/}})$. Vertical dashed line indicates individual time-to-event.

**Ordinality Visualization (1) Survival prompts:** we calculate the similarity between any two survival prompts learned by the model and show the similarity in heatmap. By comparing the heatmaps

from the VLSA model with and without ordinal prompts, the effect of ordinal prompts can be examined qualitatively. As shown in Figure 3(a), we can see that survival prompts are better in ordinality when considering an ordinal inductive bias for them. **(2) Incidence function:** the case study shown in Figure 3(b) suggest that the ordinality constraint imposed on IF could help to decrease the EMD between $\hat{y}$ and the ground truth. This often leads to the gains in MAE, as seen from Table 3.

## 4.5 ANALYSIS OF DATA EFFICIENCY (FEW-SHOT SURVIVAL ANALYSIS)

Here we focus on a few-shot learning (FSL) scenario in which only a few samples are available for training. We use this scenario to simulate the case of scarce training data and assess the performance of different SA methods in terms of data efficiency. Since existing works on SA rarely study FSL, this experiment follows common few-shot settings. Concretely, we randomly sample $s$ patients from each time-discrete class. For censored patients, their true class labels are partially given, so we adopt the KM method (Kaplan & Meier, 1958) to estimate them in advance for few-shot sampling. Moreover, due to small patient numbers, we run each single experiment 5 times and report the median metrics, following CONCH (Lu et al., 2024). MI-FewShot (Xu et al., 2024) is a non-parametric method designed for FSL, so we adopt it as an additional baseline for comparisons.

Table 4: **Few-shot survival analysis for data efficiency evaluation**.

| Method | # Shots | | | | | Average | Full-shots |
|---|---|---|---|---|---|---|---|
| | 1 | 2 | 4 | 8 | 16 | | |
| ABMIL | 0.5061 | 0.5136 | 0.5339 | 0.5541 | 0.5798 | 0.5375 | 0.6426 |
| TransMIL | 0.5607 | 0.5856 | 0.6145 | 0.6236 | 0.6388 | 0.6046 | 0.6414 |
| ILRA | 0.5503 | 0.5658 | 0.5891 | 0.6023 | 0.6033 | 0.5822 | 0.6136 |
| R²T-MIL | 0.5725 | 0.5835 | 0.5924 | 0.5927 | 0.6072 | 0.5897 | 0.6245 |
| DeepAttnMISL | 0.5182 | 0.5252 | 0.5522 | 0.5682 | 0.5877 | 0.5503 | 0.5736 |
| Patch-GCN | 0.5627 | 0.5497 | 0.5872 | 0.6022 | 0.6186 | 0.5841 | 0.6726 |
| MI-FewShot | 0.5478 | 0.5439 | 0.5499 | 0.5539 | 0.5873 | 0.5567 | 0.5692 |
| ABMIL$_{Prompt}$ | 0.5314 | 0.5588 | 0.5737 | 0.5958 | 0.6068 | 0.5733 | 0.6500 |
| CoOp | **0.5846** | 0.5870 | 0.6081 | 0.6056 | 0.6237 | 0.6018 | 0.6482 |
| OrdinalCLIP | 0.5710 | 0.5983 | 0.6118 | 0.6106 | 0.6246 | 0.6033 | 0.6604 |
| VLSA | 0.5787 | **0.6068** | **0.6271** | **0.6465** | **0.6592** | **0.6237** | **0.6954** |

The results are shown in Table 4. **(1)** From these results, we find that our VLSA still often obtains the best performance in FSL scenarios. This indicates the data efficiency of our VLSA models. **(2)** When using only 16 shots, our VLSA can obtain comparable performance with nearly-all full-shot baseline models. **(3)** Furthermore, we observe that, in standard full-shot setting, two VL-based baselines, *i.e.*, CoOp and OrdinalCLIP with ABMIL as their MIL aggregator, only show an improvement of 0.6% and 1.8% over vision-only ABMIL, respectively. By contrast, the average improvements seen in few-shot settings are around 6.5%, much larger than those in full-shot. This result also suggests the advantage of VL-based schemes over vision-only ones in data efficiency.

## 5 CONCLUSION

This paper presents a new Vision-Language Survival Analysis (VLSA) paradigm. It designs the first VL-based SA framework for histopathology WSIs. Different from current SA paradigm in CPATH, VLSA proposes to leverage language-encoded prognostic priors as auxiliary signals to improve weakly-supervised WSI representation learning. Moreover, considering the intrinsic characteristics of SA, VLSA proposes ordinal survival prompt learning and ordinal incidence function prediction. Owing to the introduce of language-encoded prognostic priors, our VLSA enjoys a good interpretability, supported by its Shapley values-based interpretation method. The extensive experiments on five datasets verify our scheme's effectiveness: VLSA obtains new state-of-the-art performances in SA tasks and it could often show clear superiority over existing schemes in terms of data and computation efficiency. Notably, our empirical results suggest that VL-based learning paradigm can offer weakly-supervised MIL an effective means to learn valuable prognostic clues from gigapixel WSIs. This finding is likely to provide an alternative solution for improving the weak supervision in MIL, thereby paving a new way for survival analysis in computational pathology.

## 6 ACKNOWLEDGMENT

This work is supported by the National Natural Science Foundation of China (NSFC) under Grant No. 62476049, partly by the Fundamental Research Funds for the Central Universities under Grant No. ZYGX2022YGRH015 and the project of Sichuan Provincial Department of Science and Technology under Grant No. 2023YFQ0011. The authors would like to thank Mahmood Lab for making CONCH public to facilitate the research, and anonymous reviewers for their constructive comments to improve this work.

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

APPENDIX

# A PRELIMINARIES

## A.1 SURVIVAL ANALYSIS

**Survival analysis (SA)** Also known as *time-to-event analysis*, SA is one of the primary statistical approaches for analyzing data on time to event (Kaplan & Meier, 1958; Cox, 1975). Time-to-event data is usually denoted by $\mathcal{D} = \{\boldsymbol{X}_i, t_i, \delta_i\}_{i=1}^N$, where $\boldsymbol{X}_i$ represents individual characteristics and $\delta_i$ is the individual status of event of interest (*e.g.*, death and machine failure) observed at time $t_i$. Particularly, $\delta = 0$, called censorship in SA, indicate that event does not occur during follow-up observation. They are also considered in analysis and modeling. One common task in SA is to predict the risk of event occurrence. It covers a wide range of applications in healthcare and finance. Next, we give some key concepts in SA for reference.

**Survival Function** It quantifies the probability one will survive past a given time $t$, often written as $S(t) = Pr(T > t)$ where $T$ is a random variable indicating survival time. This function is *non-increasing*, *i.e.*, the probability of survival will not increase as the observation process continues. Given individual survival function, the corresponding time-to-event can be estimated by $\hat{t} = \mathbb{E}_t[S(t)] = \int_0^\infty S(t)\mathrm{d}t$.

**Hazard Function** It quantifies the probability that one will experience event at time $t$ given no event occurrence before $t$. By definition, it is written as $h(t) = \lim_{\Delta t \to 0} \frac{Pr(t \leq T \leq t+\Delta t | T > t)}{\Delta t}$. Thus, survival function can be derived from it by $S(t) = \exp(-\int_0^t h(z)\mathrm{d}z)$.

**Incidence Function (IF)** It aims to capture the probability distribution of *the first hitting time*, $p(t)$, often seen in SA under competing risks (Fine & Gray, 1999; Lee & Whitmore, 2006; Lee et al., 2018). Cumulative IF (CIF) gives the probability that one experiences event at or before time $t$, namely, $\text{CIF}(t) = \int_0^t p(z)\mathrm{d}z$. Thus, survival function can be calculated by $S(t) = 1 - \text{CIF}(t)$.

**Modeling Approach** To estimate individual risks, most survival modeling approaches take the following functions as prediction target.

- *Proportional hazard* (Cox, 1975). As the most popular one (CoxPH) in SA, it assumes that individual hazard function is proportional to a cohort-level hazard function (computable); it predicts a scalar indicating individual proportional hazard.

- *Hazard function* (Tutz & Schmid, 2016). Traditional SA models usually assume a specific form for the distribution of individual hazard function, and then optimize the parameter of this specified distribution. Modeling hazard function without any explicit assumption is also studied in SA. It often performs better than traditional ones in real-world datasets.

- *Incidence function*. In DeepHit (Lee et al., 2018), it is first taken as prediction target for deep learning-based SA. Its optimization are similar to classification with partial labels (refer to Eq. (9)). One biggest difference is that modeling incidence function requires to handle the censored individuals (whose true time-to-event is partially known) in $\mathcal{D}$.

Additionally, *survival time* is also directly adopted as prediction target in some SA models (Liu et al., 2024a). Readers could refer to Tutz & Schmid (2016) for more details about SA.

When prediction target is incidence function or hazard function, survival time is usually preprocessed as discrete time labels for analysis. We compare these two prediction targets and try different time discretization settings for our VLSA. The results can be found in Appendix D.6.

## A.2 WEAKLY-SUPERVISED MIL

**Multi-Instance Learning** (MIL) is a fundamental machine learning problem that has been studied for decades (Dietterich et al., 1997; Ilse et al., 2018; Liu & Ji, 2024). Different from conventional learning settings, a sample in MIL is a bag of multi-instances, *i.e.*, $\boldsymbol{X} = \{\boldsymbol{x}_k\}_{k=1}^K$. $\boldsymbol{x}_k$ is the $k$-th instance. In weakly-supervised MIL, bag-level labels are given while *instance-level labels are unknown*. A MIL model is often required to learn from size-varied bags and make bag-level or instance-level predictions. Its applications cover pathology image analysis, video anomaly detection, point could analysis, etc.

**Embedding-level approach** (Ilse et al., 2018) is often adopted for bag-level prediction in the era of deep learning. This approach provides a principled and general three-step framework for MIL. (1) *Instance-level embedding*: $\boldsymbol{x}_k$ is transformed into a low-dimensional embedding $\boldsymbol{h}_k$. (2) *Permutation-invariant pooling*: $\{\boldsymbol{h}_k\}_{k=1}^K$ are combined into a bag-level representation $\boldsymbol{h}_{\text{bag}}$ via permutation-invariant pooling. (3) *Bag classification*: bag-level representation is passed through a classifier for prediction. Common pooling strategies contain mean, max, and attention. Particularly, attention-based pooling is a classic method proposed by Ilse et al. (2018), frequently adopted in weakly-supervised MIL. It can be written as follows:

$$\boldsymbol{h}_{\text{bag}} = \sum_{k=1}^K a_k \cdot \boldsymbol{h}_k, \;\; a_k = \text{Softmax}\left(\varphi(\boldsymbol{h}_k)\right) = \frac{\exp\left(\varphi(\boldsymbol{h}_k)\right)}{\sum_{i=1}^K \exp\left(\varphi(\boldsymbol{h}_i)\right)}, \tag{12}$$

where $\varphi(\cdot)$ is a transformation function in the attention network, parameterized by a multi-layer perceptron. Embedding-level approach usually assumes $\boldsymbol{x}_1, \boldsymbol{x}_2, \cdots, \boldsymbol{x}_K$ are *i.i.d.* Subsequent works (Shao et al., 2021b) improve it by assuming and handling the relation between instances. They often perform better in bag classification tasks.

## A.3 EARTH MOVER'S DISTANCE

Earth Mover's Distance (EMD), as known as the Wasserstein distance ($p = 1$), is utilized to the measure the distance between two distributions (Kolouri et al., 2017). Intuitively, considering two simple one-dimensional discrete probability distributions, $P(x)$ and $Q(x)$, EMD captures the optimal total transportation costs in moving the probability mass of $P$ from any point $a$ to $b$ to make $P$ equal to $Q$ in probability density. In every moving process, the cost is the product of moving mass and *moving distance* ($|a - b|$). From this intuitive example, we could find that EMD considers the geometry property of distribution in distance measurement. This makes EMD different from other metrics like Kullback-Leibler divergence and Euclidean distance.

Formally, EMD is defined as an infimum over joint probabilities:

$$\text{EMD}(P, Q) = \inf_{\gamma \in \Pi(P,Q)} \mathbb{E}_{(x,y) \sim \gamma} d(x, y). \tag{13}$$

$\Pi(P, Q)$ is the set of all joint distributions whose marginal distributions are $P$ and $Q$. $d(x, y)$ is the distance between $x$ and $y$. As shown in Levina & Bickel (2001), for two one-dimensional discrete distributions, $P$ and $Q$, EMD can be calculated by

$$\text{EMD}(P, Q) = \left(\frac{1}{C}\right)^{\frac{1}{l}} ||\text{CDF}(P) - \text{CDF}(Q)||_l, \tag{14}$$

where CDF means cumulative density function and $C$ is the number of discrete values.

## A.4 SHAPLEY VALUE

Shapley value is a classic solution concept in cooperative game theory (Shapley, 1953). It assigns a unique contribution to each player according to the payoff earned by the coalition of players. In machine learning, Shapley value is often adopted to interpret model predictions (Lundberg, 2017), *e.g.*, the importance of input features. This application enables us to interpret black-box, complex nonlinear models, *e.g.*, deep networks, particularly popular in the medical domain.

Formally, to calculate the importance of each feature for an original model $f(\cdot)$, there is $f(x) = g(x') = f(h_x(x'))$ for a corresponding explanation model $g(\cdot)$, where $x'$ is a simplified input that maps to the original input $x$ through a function $x = h_x(x')$. *Additive feature attribution* methods provide such explanation models. Concretely, $g(\cdot)$ is formulated as a linear function of binary variables. It is expected to have a property of *local accuracy*, expressed as

$$f(x) = g(x') = \phi_0 + \sum_{i=1}^A \phi_i x_i', \tag{15}$$

where $\phi_i \in \mathbb{R}$ indicates the importance of the $i$-th feature, $x' \in \{0, 1\}^A$, and $A$ is the number of simplified input features. Moreover, $g(\cdot)$ should satisfy two other desirable properties: i) *missingness*,

missing features ($x'_i = 0$) have no attributed impact; and ii) *consistency*, some features' attribution should not decrease if these features' contribution (*i.e.*, marginal reward) does not decrease. Theorem 1 in Lundberg (2017) states that only one possible $g(\cdot)$ follows those three properties:

$$\phi_i(f, x) = \sum_{z' \subseteq x'} \frac{|z'|!(A - |z'| - 1)!}{A!} \left[ f_x(z') - f_x(z' \setminus i) \right], \tag{16}$$

where $|z'|$ is the number of non-zero entries in $z'$ and $z' \subseteq x'$ represents all $z'$ vectors where the non-zero entries are a subset of the non-zero entries in $x'$. This theorem follows from combined cooperative game theory results. $\phi_i$ is known as Shapley value (Shapley, 1953). Readers could refer to Lundberg (2017) for proofs and related details.

## B MORE DISCUSSIONS ON ORDINAL INDUCTIVE BIAS

Our VLSA considers two ordinal inductive biases: i) *ordinal survival prompts* and ii) *ordinal incidence function*. For better understanding, in this section we first explain them in detail and then discuss the connection and difference between them.

**Further Explanation** (1) Ordinal survival prompts. For $C$ survival classes after time discretization, there are $C$ prompts that encode the information of survival labels. Concretely, from the first class to the last class, *i.e.*, for time-to-event ranging from $[T_0, T_1)$ to $[T_{C-1}, T_C)$, the respective class prompt describes a risk from high to low. Therefore, if two classes are closer in ordering, *e.g.*, the $c$-th and $(c+1)$-th classes, their class prompts would be more similar in encoded risk information. Formally,

$$\text{Sim}(\boldsymbol{V}_{\text{cls}}^c, \boldsymbol{V}_{\text{cls}}^i) > \text{Sim}(\boldsymbol{V}_{\text{cls}}^c, \boldsymbol{V}_{\text{cls}}^j) \text{ if } |c - i| < |c - j|, \tag{17}$$

where $\text{Sim}(\cdot, \cdot)$ is a function measuring two inputs' similarity. (2) Ordinal incidence function. It captures the probability distribution of discrete first-hitting time, *i.e.*, $\boldsymbol{y} = [y_1, \cdots, y_C]$. Since this distribution also characterize ordered survival classes like the class prompt aforementioned, there is a similar ordinality in incidence function,

$$|y_c - y_i| < |y_c - y_j| \text{ if } |c - i| < |c - j|, \tag{18}$$

as depicted in Figure 1(b). Our experiments (Table 3) confirm the effectiveness of these two ordinal inductive biases: i) ordinal survival prompts tend to enhance the model's ability in risk discrimination; and ii) ordinal incidence function could often reduce the bias of predicted survival time (MAE).

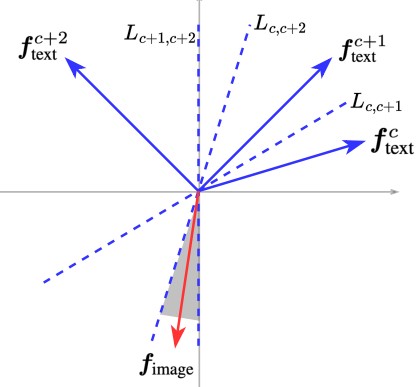

Figure 4: An example to illustrate the case that incidence function is not ordinal whereas prompts are. $L_{c,c+1}$ is the bisector of the angle between $\boldsymbol{f}_{\text{text}}^c$ and $\boldsymbol{f}_{\text{text}}^{c+1}$. When $\boldsymbol{f}_{\text{image}}$ falls into the gray area, $\cos(\boldsymbol{f}_{\text{image}}, \boldsymbol{f}_{\text{text}}^{c+1}) > \cos(\boldsymbol{f}_{\text{image}}, \boldsymbol{f}_{\text{text}}^{c+2})$ does not hold.

**Connection and Difference between Two Ordinal Inductive Biases** Taking two inductive biases into one formula, the prediction of individual incidence function can be written as

$$\hat{\boldsymbol{y}} = \text{Softmax} \left( \tau \cdot \left[ \cos(\boldsymbol{f}_{\text{image}}, \boldsymbol{f}_{\text{text}}^1), \cos(\boldsymbol{f}_{\text{image}}, \boldsymbol{f}_{\text{text}}^2), \cdots, \cos(\boldsymbol{f}_{\text{image}}, \boldsymbol{f}_{\text{text}}^C) \right] \right). \tag{19}$$

So, the first inductive bias, introduced into $\{\boldsymbol{f}_{\text{text}}^c\}_{c=1}^C$, could be cast as a *model-level* ordinal constraint, because $\boldsymbol{f}_{\text{text}}^c$ is a part of model parameters and it is used in all individual predictions. The second inductive bias, imposed on individual prediction $\hat{y}$, is a *subject-level* ordinal constraint. We claim that *only* considering model-level ordinal constraint cannot always lead to ordinal incidence function, *i.e.*, Eq. (18) does not hold in some cases even when Eq. (17) holds for all $c, i, j \in [1, C]$. We show an example in Figure 4. In this example, $\boldsymbol{f}_{\text{text}}^c, \boldsymbol{f}_{\text{text}}^{c+1}, \boldsymbol{f}_{\text{text}}^{c+2}$ are ordinal and there are $\cos(\boldsymbol{f}_{\text{image}}, \boldsymbol{f}_{\text{text}}^c) > \cos(\boldsymbol{f}_{\text{image}}, \boldsymbol{f}_{\text{text}}^{c+1})$ and $\cos(\boldsymbol{f}_{\text{image}}, \boldsymbol{f}_{\text{text}}^c) > \cos(\boldsymbol{f}_{\text{image}}, \boldsymbol{f}_{\text{text}}^{c+2})$. However, there exists an area (colored in gray) so that $\cos(\boldsymbol{f}_{\text{image}}, \boldsymbol{f}_{\text{text}}^{c+1}) > \cos(\boldsymbol{f}_{\text{image}}, \boldsymbol{f}_{\text{text}}^{c+2})$ does not hold. Namely, there exists the case in which incidence function is not ordinal even if all survival prompts satisfy the ordinality. Therefore, both model-level and subject-level ordinal constraint should be considered in VLSA for better survival prediction. This is also implied by our empirical results in Table 3.

## C    MORE EXPERIMENTAL DETAILS

### C.1    DATASETS

We provide the statistics on five TCGA datasets (BLCA, BRCA, GBMLGG, LUAD, and UCEC) used in this study. Please see them in Table 5.

The preprocessing of histopathology WSIs contains the following key steps: i) tissue region segmentation, ii) image patching, and iii) patch feature extraction. Following CONCH (Lu et al., 2024), each WSI is divided into patches with size of $448 \times 448$ pixels at $20\times$ magnification. In the last step, CONCH's image encoder is used to obtain patch features. We utilize a toolkit provided by CLAM [1] (Lu et al., 2021) for these steps. The statistical details on processed instances (patches) are provided in Table 5.

There are three notable numbers in Table 5. **(1) The number of patients** in each dataset is smaller than 1,000. This scale is too small to compare with current standard vision datasets. This poses great challenges to survival analysis on WSIs. **(2) The number of instances for one patient** can reach up to 134,016 in UCEC. Thus, it could be challenging for MIL models to learn one effective representation from numerous patches. **(3) The number of time bins** is usually around 10. This means that manually designing fine-grained risk descriptions at all 10 levels is often intractable.

Table 5: **Dataset details**. The last two rows are calculated at patient level.

| Statistic | TCGA | | | | |
| --- | --- | --- | --- | --- | --- |
| | BLCA | BRCA | GBMLGG | LUAD | UCEC |
| # Patients | 373 | 956 | 569 | 453 | 480 |
| # Patients w/ event ($N_e$) | 169 | 130 | 189 | 158 | 75 |
| # Time bins ($\sqrt{N_e}$) | 12 | 11 | 13 | 12 | 8 |
| Event ratio | 45.31% | 13.60% | 33.22% | 34.88% | 15.63% |
| Maximum $t$ (month) | 163.17 | 282.69 | 211.01 | 238.11 | 225.33 |
| # WSIs | 437 | 1,022 | 1,016 | 516 | 539 |
| # WSIs per patient | 1.17 | 1.07 | 1.79 | 1.14 | 1.12 |
| # Instances (sum) | 2,463,402 | 4,113,285 | 3,508,119 | 2,302,933 | 3,229,606 |
| # Instances (mean) | 6,604.3 | 4,302.6 | 6,165.4 | 5,083.7 | 6,728.3 |
| # Instances (max) | 69,843 | 23,263 | 75,012 | 63,658 | 134,016 |

### C.2    BASELINES

**Vision-Only Baselines** We compare the baselines originally proposed for WSI classification, *i.e.*, ABMIL (Ilse et al., 2018), TransMIL (Shao et al., 2021b), ILRA (Xiang & Zhang, 2023), and R$^2$T-MIL (Tang et al., 2024), and those for WSI survival analysis, *i.e.*, DeepAttnMISL (Yao et al., 2020) and Patch-GCN (Chen et al., 2021a). DeepAttnMISL and Patch-GCN are the most representative method based on cluster and graph, respectively.

---

[1]https://github.com/mahmoodlab/CLAM

**Vision-Language Baselines** Since the vast majority of existing works focus on classification or segmentation tasks, we adapt the related methods to WSI survival analysis. These methods include MI-Zero (Lu et al., 2023) for zero-shot WSI classification, ABMIL$_{Prompt}$, CoOp (Zhou et al., 2022) for context prompt optimization, and OrdinalCLIP (Li et al., 2022) for VL-based ordinal regression.

**Implementation Details** Baselines are implemented as follows:

- ABMIL, TransMIL, ILRA, and R$^2$T-MIL: we adapt them to SA by replacing their prediction target with incidence function and using the $\mathcal{L}_{MLE}$ same as our VLSA. Moreover, their time discretization settings (Section 3.2) ares the same as VLSA.

- DeepAttnMISL and Patch-GCN: we follow their original implementations for SA tasks. Concretely, DeepAttnMISL adopts a survival modeling approach same as CoxPH (Cox, 1975). It predicts a proportional hazard. We transform it into discrete survival distribution, based on a standard KM method (Kaplan & Meier, 1958; Qi et al., 2023b). Patch-GCN predicts discrete hazard function for individuals and is optimized by $\mathcal{L}_{MLE}$, where survival time is discretized by quantiles.

- MI-Zero: In vision-end, we try different score aggregation methods proposed by it to obtain WSI representations and report the best one, *i.e.*, mean-based aggregation. In language-end, we try different survival prompts for MI-Zero and finally adopt the best: context = "*the cancer prognosis reflected in the pathology image is*", classes = ["*very poor*", "*poor*", "*good*", "*very good*"]. The final four survival prompts are encoded into four textual vectors by the CONCH's text encoder. They are then used to derive the prediction of IF via Eq. (6), same as our VLSA.

- ABMIL$_{Prompt}$: In vision-end, ABMIL is used to learn WSI-level representations. In language-end, the naive prompts (prefixed and not learnable), same as those used in MI-Zero and described above, are used for survival prediction. It is adopted as a stronger baseline than MI-Zero for comparisons.

- CoOp and OrdinalCLIP: the implementation of CoOp and OrdinalCLIP are stated in the main paper. Unless specified, they use the same setting as our VLSA.

## C.3 VLSA

This section provides the implementation details of our VLSA, including the description of prognostic priors, survival prompt settings, hyper-parameter settings, and training settings. More details could be found in our source code.

**Description of Prognostic Priors** We derive it by asking GPT-4o (a version before 2024-07-01) the following questions one by one: i) "*What visual features of H&E stained pathological images are related to the prognosis of* [*CANCER_TYPE*]? *Please list them point by point.*", ii) "*Please exclude the features that cannot be directly observed from pathology images.*", and iii) "*Please briefly describe each relevant visual feature.*". Then, we further process the answers from GPT-4o by manually extracting the most relevant part of visual feature descriptions. The final results for five datasets are exhibited in Table 6. From this table, we can see that common prognostic features in WSIs are "celluar atypia", "tumor infiltration", and "tumor growth pattern".

Table 6: **Descriptions of prognostic priors**.

| Item | Brief Summary | Description |
|---|---|---|
| **TCGA-BLCA** ($M = 12$) | | |
| 1 | Cellular atypia | Abnormalities in the size, shape, and organization of tumor cells. High degrees of atypia are associated with more aggressive tumors. |
| 2 | Nuclei atypia | Variation in the size and shape of the nuclei within tumor cells. Pronounced pleomorphism typically indicates a higher grade and more aggressive cancer. |
| 3 | Mitotic activity | The presence and frequency of cell division figures (mitoses) within the tumor. High mitotic activity suggests rapid tumor growth and a poorer prognosis. |
| 4 | Tumor arrangement pattern | The structural arrangement of the tumor, such as solid or mixed patterns. |
| 5 | Tumor infiltration | Tumor cells invade deep into the bladder wall layers, especially into detrusor muscle or muscularis propria, perivesical fat and beyond. |

| Item | Brief Summary | Description |
|------|---------------|-------------|
| 6 | Tumor invasion | Tumor cells within blood vessels or lymphatic channels. |
| 7 | Necrotic tumor cells | Areas of dead (necrotic) tumor cells within the tissue. Necrosis often appears as regions lacking viable cells and can indicate rapid tumor growth outpacing its blood supply. |
| 8 | Perineural Invasion | Tumor cells surrounding or invading nerves. |
| 9 | Tumor growth pattern | Dense fibrous or connective tissue growth around the tumor indicates an aggressive response to the tumor, often seen in high-grade cancers. |
| 10 | Tumor infiltration | Tumor cells penetrate the surrounding stroma, muscle, or other tissues, lack of a clear demarcation between the tumor and the normal tissue, known as poorly defined tumor margins. |
| 11 | Formation of new blood vessels | Formation of new blood vessels within the tumor, necessary for tumor growth and spread. High levels of angiogenesis indicate a more aggressive tumor. |
| 12 | Carcinoma in situ | Presence of areas of flat, high-grade, non-invasive cancer, known as carcinoma in situ, identified by high-grade cellular atypia disorganized epithelial structure, increased and atypical mitotic activity, loss of normal epithelial architecture, and absence of umbrella cells. |

**TCGA-BRCA** ($M = 10$)

| Item | Brief Summary | Description |
|------|---------------|-------------|
| 1 | Tumor size | The dimensions of the tumor observed on the slide. |
| 2 | Tumor boundary | The boundary between the tumor and surrounding tissue. |
| 3 | Tumor differentiation | The differentiation of tumor cells. |
| 4 | Tumor invasion | Presence of tumor cells within lymphatic and blood vessels. |
| 5 | Lymph node metastasis | Visual confirmation of metastasis in lymph nodes, seen as clusters of tumor cells. |
| 6 | Cellular morphology | The appearance of tumor cells, including their size, shape, and nuclear features. |
| 7 | Necrotic tumor cells | Areas within the tumor where cells have died, typically due to insufficient blood supply. |
| 8 | Tumor infiltration | The characteristics of the tissue surrounding the tumor, such as fibrosis (desmoplasia) and immune cell infiltration. |
| 9 | Perineural Invasion | Tumor cells surrounding or invading nerves. |
| 10 | Pattern of tumor growth and arrangement | Different patterns of tumor growth and cell arrangement, which vary among subtypes such as ductal, lobular, and mucinous carcinoma. |

**TCGA-GBMLGG** ($M = 7$)

| Item | Brief Summary | Description |
|------|---------------|-------------|
| 1 | Cellular density | High cellular density indicates a large number of closely packed cells, suggesting rapid tumor growth and aggressive behavior. |
| 2 | Nuclei atypia | The abnormal appearance of cell nuclei, characterized by variations in size, shape, and staining intensity. Higher nuclear atypia indicates more aggressive tumor cells with significant genetic abnormalities. |
| 3 | Mitotic activity | Cells undergoing division (mitosis) can be identified by the presence of mitotic figures. A higher number of mitotic figures points to increased tumor cell proliferation and aggressiveness. |
| 4 | Necrotic tumor cells | Areas of dead tumor cells, often surrounded by viable tumor cells. Necrosis is a hallmark of high-grade tumors and is associated with poor oxygen supply and rapid tumor growth. Pseudopalisading necrosis features rows of tumor cells lining the necrotic areas. |
| 5 | Formation of new blood vessels | The abnormal and excessive growth of small blood vessels within the tumor. This indicates the tumor's ability to stimulate new blood vessel formation (angiogenesis), which supports its rapid growth and spread. |
| 6 | Tumor infiltration | Tumor cells diffusely infiltrate into normal brain tissue, identified by scattered tumor cells within normal brain tissue, unclear boundaries between tumor and normal tissue, and a disrupted normal tissue architecture. |
| 7 | Tumor arrangement pattern | Tumor cells arranged in a palisading pattern around necrotic areas. This feature is characteristic of Glioblastoma and indicates aggressive tumor behavior and rapid cell turnover. |

**TCGA-LUAD** ($M = 8$)

| Item | Brief Summary | Description |
|------|---------------|-------------|
| 1 | Tumor differentiation | The degree of cellular differentiation within the tumor. |
| 2 | Tumor size | The dimensions of the tumor mass within the tissue section, often measured in terms of its diameter or extent. |
| 3 | Tumor boundary | The appearance of tumor borders within the tissue. |
| 4 | Nuclei features | The characteristics of tumor cell nuclei, such as size, shape, and staining pattern. |
| 5 | Mitotic activity | The presence of actively dividing cells, identified by the presence of mitotic figures within the tumor tissue. |
| 6 | Necrotic tumor cells | Areas of tissue death within the tumor mass. |
| 7 | Pattern of tumor growth and arrangement | The overall arrangement and growth pattern of tumor cells within the tissue. |
| 8 | Tumor invasion | The presence of tumor cells within lymphatic or blood vessels. |

**TCGA-UCEC** ($M = 10$)

| Item | Brief Summary | Description |
|------|---------------|-------------|
| 1 | Nuclei atypia | The variation in size, shape, and chromatin pattern of cell nuclei. High-grade nuclei appear larger, irregular, and darker due to increased chromatin (hyperchromasia). Prominent nucleoli may also be visible. |
| 2 | Mitotic activity | The number of mitotic figures (cells undergoing division) per high-power field. High mitotic activity indicates rapid cell proliferation, which is a sign of tumor aggressiveness. |
| 3 | Tumor arrangement pattern | The structural pattern of the tumor, including gland formation and architectural complexity. Poorly differentiated tumors may show solid, cribriform, or papillary patterns. |
| 4 | Tumor invasion | The extent of the tumor's penetration into surrounding tissues, particularly the depth of myometrial invasion. It also includes the presence of tumor cells in lymphovascular spaces (LVSI), indicating potential for metastasis. |

| Item | Brief Summary | Description |
|------|---------------|-------------|
| 5 | Tumor infiltration | The appearance of the tumor edges, which can be infiltrative (irregular, spreading into surrounding tissue) or pushing (smooth, well-defined borders). Irregular, jagged margins suggest aggressive growth. |
| 6 | Necrotic tumor cells | Areas of dead (necrotic) tumor cells within the tissue. Necrosis often appears as regions lacking viable cells and can indicate rapid tumor growth outpacing its blood supply. |
| 7 | Inflammatory cell | The presence and density of inflammatory cells (e.g., lymphocytes, macrophages) within and around the tumor. |
| 8 | Tumor infiltration | Changes in the supportive tissue (stroma) surrounding the tumor, including desmoplasia (dense fibrous tissue response). |
| 9 | Tumor differentiation | Areas within the tumor where cells show squamous (flat, scale-like) characteristics. Squamous differentiation can impact the behavior and classification of the tumor. |
| 10 | Tumor heterogeneity | The variability in cellular and structural characteristics within different areas of the tumor. High heterogeneity indicates diverse cell populations, often associated with more aggressive behavior and resistance to treatment. |

**Survival Prompt Settings** In time discretization, the number of discrete time bins is set by $C = \sqrt{N_e}$. Specific numbers for five datasets can be found in Table 5. For context prompt, we initialize $\boldsymbol{V}_{\text{ctx}}$ using the text "*a histopathology image suggesting*". We use 4 base class prompts, *i.e.*, $B = 4$. Their learnable parameters, $\{\boldsymbol{V}_{\text{cls}}^{\lambda_1}, \boldsymbol{V}_{\text{cls}}^{\lambda_2}, \boldsymbol{V}_{\text{cls}}^{\lambda_3}, \boldsymbol{V}_{\text{cls}}^{\lambda_4}\}$, are initialized using "*a very poor prognosis*", "*a poor prognosis*", "*a good prognosis*", and "*a very good prognosis*", respectively. Following the interpolation setting in Li et al. (2022), we set the ordering distance matrix to $\boldsymbol{D}_{c,b} = |(c-1) - (b-1) \cdot (C-1)/(B-1)|$ and adopt the linear interpolation, *i.e.*, $W(\boldsymbol{D}_{c,b}) = 1 - \frac{\boldsymbol{D}_{c,b}}{C-1}$.

**Hyper-Parameter Settings** Since we adopt CONCH in our experiments, most settings for feature dimension is the same as CONCH: $D = 512$ and $D_{\text{emb}} = 768$. We simply set the coefficient of $\mathcal{L}_{\text{EMD}}$ to $\beta = 1$ without fine-tuning, and $\alpha = 100$ by default.

**Training Settings** In model training, we set the number of epochs to 10, learning rate to 0.0002, batch size to 1 (one bag), the step of gradient accumulation to 32, and the optimizer to Adam with a weight decay rate of 0.00001. These setting are shared across five datasets, without dataset-specific fine-tuning. Please refer to our source code [2] for more training details. All experiments are run on a machine with two NVIDIA GeForce RTX 3090 GPUs (24G).

## C.4 EVALUATION METRICS

Our main evaluation metrics contain concordance index (CI), mean absolute error (MAE), and distribution calibration (D-cal). They are adopted to evaluate the performance of SA models in risk discrimination and distribution calibration. We utilize the toolkit from SurvivalEVAL [3] (Qi et al., 2023b) for performance evaluation. The section is adapted from Qi et al. (2023a). Readers could refer to Qi et al. (2023a) for more details. We show the details of these metrics below.

**Concordance Index (CI)** It is the most popular metric in SA (Harrell Jr et al., 1996). It is often used to assess the SA model's ability in predicting high risk for the patient who experiences the event earlier. It is ranged in $[0, 1]$, calculated by

$$\text{CI} = \frac{\sum_{i,j} \mathbb{1}_{t_i < t_j} \cdot \mathbb{1}_{\hat{R}_i > \hat{R}_j} \cdot \delta_i}{\sum_{i,j} \mathbb{1}_{t_i < t_j} \cdot \delta_i}, \tag{20}$$

where $\mathbb{1}(\cdot)$ is an indication function and $\hat{R}_i$ the risk prediction of the $i$-th individual. A larger CI suggests the model is more powerful in risk discrimination.

**Mean Absolute Error (MAE)** It measures the mean of the absolute error between ground truth time-to-event and the predicted event time $\hat{t}_i$. It can be calculated via

$$\text{MAE}(\hat{t}_i, t_i, \delta_i) = \delta_i \cdot |t_i - \hat{t}_i| + (1 - \delta_i) \cdot \max(0, t_i - \hat{t}_i). \tag{21}$$

In this formulation, a hinge loss is adopted to evaluate the MAE of censored patients. Namely, as we only know the actual time-to-event of a censored patient is larger than $t_i$, the MAE is 0 if the prediction $\hat{t}_i$ is larger than $t_i$; otherwise, it should be $t_i - \hat{t}_i$. The smaller the value of MAE, the better the model performance.

---

[2] https://github.com/liupei101/VLSA
[3] https://github.com/shi-ang/SurvivalEVAL

**Distribution Calibration (D-cal)** (Haider et al., 2020) It is a statistical test to check if individual survival distribution is well calibrated with the ground truth survival distribution. Its calculation follows these steps. (1) For any probability interval $[\mu_a, \mu_b] \subset [0, 1]$, we first obtain the uncensored individuals whose predicted survival probabilities are in $[\mu_a, \mu_b]$ at their time-to-event, *i.e.*,

$$\mathcal{D}(\mu_a, \mu_b) = \left\{ (x_i, t_i, \delta_i = 1) \in \mathcal{D} \mid \hat{S}(t_i) \in [\mu_a, \mu_b] \right\}. \tag{22}$$

(2) The model is well calibrated in survival distribution prediction if there is

$$\frac{|\mathcal{D}(\mu_a, \mu_b)|}{\mathcal{D}} \approx \mu_b - \mu_a. \tag{23}$$

Typically, equal-sized, mutually exclusive probability intervals, like the time bins calculated in time discretization (see Section 3.2), are used in these two steps (Haider et al., 2020). Then, Pearson's $\chi^2$ test is applied to examine if $\frac{|\mathcal{D}(\mu_a, \mu_b)|}{\mathcal{D}}$ is uniformly distributed. $p \leq 0.05$ means there is a significant difference between predictive survival distribution and the ground truth, implying a SA model poor in distribution calibration. A well-calibrated model tends to make safer probability prediction, instead of providing over-confident estimations.

### C.5 TIME DISCRETIZATION SETTINGS

This section provides the details of time discretization settings for all compared SA models. For discrete SA models, survival time is often required to be transformed into a discrete variable at first.

**Conventional Settings** As stated in Appendix A.1, discrete SA models are usually categorized into two groups: hazard-based and incidence-based. Their conventional settings in time discretization are as follows:

- Hazard-based SA models: Most works (Chen et al., 2021b; Xu & Chen, 2023; Zhou & Chen, 2023; Zhang et al., 2024b) follow the settings used in Zadeh & Schmid (2020) and Chen et al. (2021a), *i.e.*, using 4 time bins with quantiles as cutoff points.

- Incidence-based SA models: As suggested in Haider et al. (2020), the number of time bins is determined by $\sqrt{N_e}$, which could often lead to better probability calibration in SA models. It is adopted in many works (Yu et al., 2011; Lee et al., 2018; Qi et al., 2023a;b). Particularly, DeepHit (Lee et al., 2018), as the first deep learning-based SA model with incidence function as prediction target, sets the number of time bins to $\sqrt{N_e}$ and the length of each time bin to be equal (*i.e.*, time cutoff points are uniformly distributed).

**Settings for Baselines and VLSA** We follow conventional settings in time discretization:

- For the hazard-based baseline model, Patch-GCN (Chen et al., 2021a), we follow its original setting to use 4 time bins with quantiles as cutoff points. We also examine its performance under the same number of time bins as VLSA for a fair comparison (Table 11).

- For the baseline models not originally proposed for SA, we adopt the same discretization settings as our VLSA, *i.e.*, $\sqrt{N_e}$ time bins with equal length.

- For our incidence-based VLSA, we use $\sqrt{N_e}$ time bins with equal length, as stated in the main paper. Besides, we also evaluate the performance of VLSA using different time discretization settings. Please refer to Table 10 for the results.

## D ADDITIONAL EXPERIMENTAL RESULTS

### D.1 RISK GROUPING AND KAPLAN-MEIER ANALYSIS

This experiment is conducted to examine whether our VLSA could discriminate the patients with high-risk and low-risk. Its results are shown in Figure 5. For each dataset, the median risk of the entire cohort is used as the cutoff for risk grouping. From the results in Figure 5, we can observe that VLSA can discriminate between high- and low-risk patients with a significant difference (P-Value $< 0.05$ by a log-rank test).

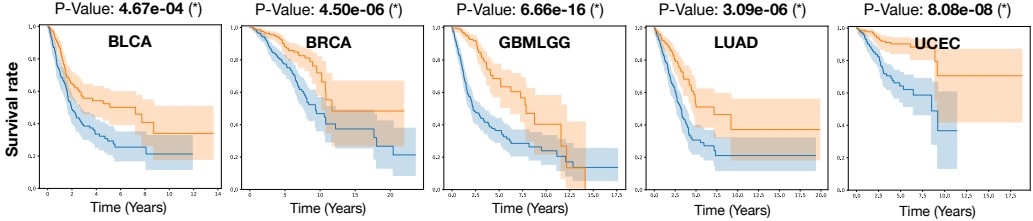

Figure 5: **Risk Grouping and Kaplan-Meier Analysis**. All patients within each dataset are grouped into two risk groups: low-risk (blue) and high-risk (orange). Patients' risk predictions are derived from VLSA. The median risk of the entire cohort is adopted as the cutoff for risk grouping.

## D.2 ADDITIONAL VISUALIZATION RESULTS

**Prediction Interpretation via Language-Encoded Prognostic Priors** Additional results are shown in Figure 6. In most cases, descriptive texts can identify the key patches that help assess cancer prognosis. Detailed descriptive texts are provided in Table 6.

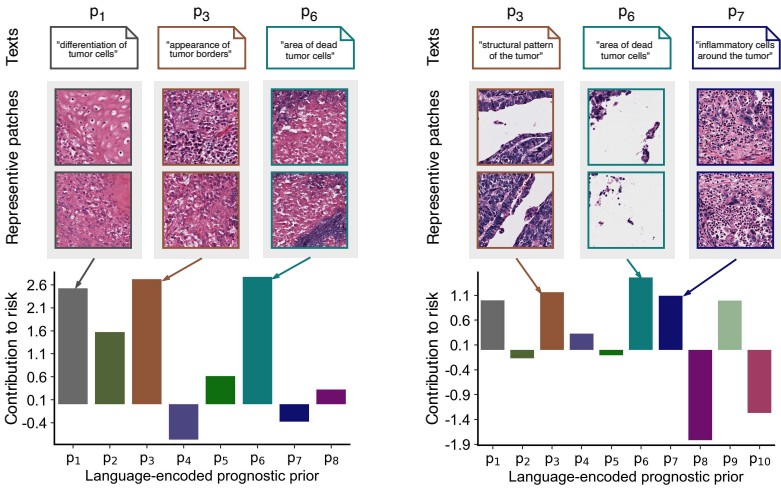

Figure 6: **Additional results of prediction interpretation via language-encoded prognostic priors**. The two examples are from the last two datasets, TCGA-LUAD (left) and TCGA-UCEC (right).

**Visualization of Survival Prompts' Ordinality** Additional results are shown in Figure 7. These results verify that there is often a better ordinality in the survival prompts optimized by VLSA.

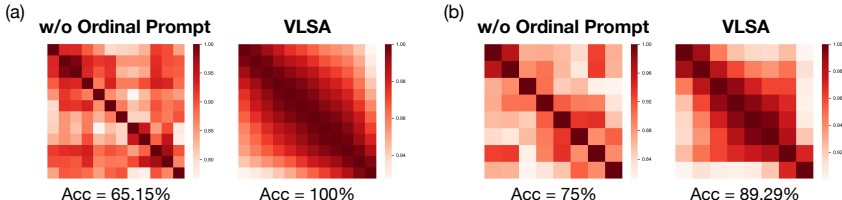

Figure 7: **Additional results of visualizing survival prompts' ordinality**. The visualization results are from the model trained with the last two datasets, TCGA-LUAD (a) and TCGA-UCEC (b).

## D.3 ANALYSIS OF COMPUTATION EFFICIENCY

In this experiment, we evaluate the computation efficiency of all models. The number of learnable parameters (# Params) and Multiply–Accumulate operations (# MACs) are taken as main evaluation metrics. From the results shown in Table 7, we can see that VLSA not only has fewer learnable parameters (284.16K) than all baseline models except for AB-MIL but also is the most efficient one (0.25G) in inference cost. Our further analysis is as follows.

**(1)** Our vision-end model only has two groups of learnable parameters: i) $T_{prog}$ for fine-tuning textual features and ii) one linear layer for projecting image features. Such network is very simple relatively yet it demonstrates strong capabilities (Table 1) in survival prediction. One critical factor behind this is that the well-aligned vision-language embeddings from foundational VLMs are effectively utilized to improve visual representation learning.

Table 7: **Computation efficiency evaluation**. A random WSI (from BLCA) with 20,265 instances is used to measure the # MACs in model inference.

| Model | | # Params | # MACs |
|---|---|---|---|
| | ABMIL | **137.74K** | 2.68G |
| | TransMIL | 683.55K | 14.23G |
| V | ILRA | 3.16M | 34.79G |
| | R$^2$T-MIL | 2.44M | 29.62G |
| | DeepAttnMISL | 329.22K | 2.68G |
| | Patch-GCN | 363.91K | 3.41G |
| | ABMIL$_{Prompt}$ | 394.24K | 2.67G |
| VL | CoOp | 434.18K | 2.68G |
| | OrdinalCLIP | 409.60K | 2.68G |
| | VLSA | 284.16K | **0.25G** |

**(2)** Our survival prompts can be computed in advance and they are subsequently reused in model inference for different individuals. Therefore, $E_{text}$ is not involved explicitly in the process of model forward inference. This enhances the computation efficiency of individual predictions.

## D.4 COMPARISON WITH HAZARD-BASED SURVIVAL MODELING

Since *hazard function* is also utilized in some SA models, we further try to take it as the prediction target of VLSA. Following Zadeh & Schmid (2020), the settings of hazard-based survival modeling for VLSA are as follows: i) survival time is discretized into 4 bins based on quantile time points; ii) we derive the prediction of individual hazard function by applying $\text{Sigmoid}(\cdot)$ to the scaled similarity scores, *i.e.*, $\{\tau \cdot \cos(f_{image}, f_{text}^c)\}_{c=1}^C$; and iii) the MLE loss corresponding to hazard function prediction is adopted to optimize the model.

The results are shown in Table 8. From these results, we observe that there is a decrease of 1.4% in overall CI performance. Additionally, when modeling individual hazard function, the predicted time-to-event is often largely biased. One possible reason is that hazard-based survival modeling may not be suitable for VL paradigm since it relies on a Sigmoid-based probability transformation and this makes it failed to establish connection with conventional multi-class classification. By contrast, incidence-based survival modeling makes SA tasks similar to the *classification with partial labels*, as reflected by Eq. (9), so it could often work better with current VL-based prediction manner.

Table 8: **Comparison with hazard-based survival modeling**.

| Prediction target | TCGA | | | | | Average | | D-cal |
|---|---|---|---|---|---|---|---|---|
| | BLCA | BRCA | GBMLGG | LUAD | UCEC | CI | MAE | Count |
| Hazard function | 0.5934 (± 0.018) | 0.6435 (± 0.072) | 0.7908 (± 0.014) | 0.6089 (± 0.049) | **0.7721** (± 0.019) | 0.6817 | 72.46 | 5 |
| Incidence function | **0.6176** (± 0.025) | **0.6652** (± 0.057) | **0.8002** (± 0.010) | **0.6370** (± 0.027) | 0.7571 (± 0.045) | **0.6954** | **25.15** | 5 |

## D.5 COMPARISON WITH MULTI-MODAL SURVIVAL ANALYSIS METHODS

In this experiment, we compare our VLSA with multi-modal SA models. The textual description of prognostic priors are adopted in VLSA to provide extra auxiliary signals for weakly-supervised learning. An alternative approach to leveraging these textual priors is multi-modal representation learning, *i.e.*, integrating and fusing visual and textual features. Accordingly, here we compare our proposed method with multi-modal SA models.

Four representative multi-modal SA models are adopted as the baselines for comparisons: MCAT (Chen et al., 2021b), MMP$_{Trans.}$ (Song et al., 2024b), MMP$_{OT}$ (Song et al., 2024b), and

MOTCat (Xu & Chen, 2023). We replace their gene input with the language-encoded prognostic priors, *i.e.*, $E_{\text{text}}(\mathcal{T}_{\text{prog}})$, used in VLSA. In addition to this settings, the other settings and model implementations keep the same as those in their official codes.

The results of this experiment are shown in Table 9. These results show that our VLSA could often perform better than the compared multi-modal SA models, with an improvement of 2.85% over the runner-up in average CI. This improvement could indicate the superiority of our scheme.

Table 9: **Comparison with Multi-Modal Survival Analysis Models**. [†] These methods are originally proposed for learning multi-modal representations from WSIs and gene data. We replace their gene input with the language-encoded prognostic priors $E_{\text{text}}(\mathcal{T}_{\text{prog}})$ for comparisons.

| Method | TCGA | | | | | Average | | D-cal |
| | BLCA | BRCA | GBMLGG | LUAD | UCEC | CI | MAE | Count |
|---|---|---|---|---|---|---|---|---|
| MCAT [†] | 0.5899 (± 0.020) | 0.6322 (± 0.035) | 0.7980 (± 0.020) | 0.5768 (± 0.043) | 0.6827 (± 0.022) | 0.6559 | 31.44 | 3 |
| MMP$_{\text{Trans.}}$ [†] | 0.5857 (± 0.040) | 0.5944 (± 0.016) | 0.7917 (± 0.012) | 0.5654 (± 0.055) | 0.6928 (± 0.049) | 0.6460 | 31.08 | 3 |
| MMP$_{\text{OT}}$ [†] | 0.5961 (± 0.031) | 0.6315 (± 0.041) | 0.7948 (± 0.019) | 0.5870 (± 0.071) | 0.6915 (± 0.052) | 0.6602 | 31.35 | 4 |
| MOTCat [†] | **0.6227** (± 0.029) | 0.6301 (± 0.047) | 0.7967 (± 0.026) | 0.5617 (± 0.070) | 0.7234 (± 0.033) | 0.6669 | 29.73 | 3 |
| VLSA | 0.6176 (± 0.025) | **0.6652** (± 0.057) | **0.8002** (± 0.010) | **0.6370** (± 0.027) | 0.7571 (± 0.045) | **0.6954** | **25.15** | 5 |

## D.6 DIFFERENT TIME DISCRETIZATION SETTINGS

**Different Time Discretization Settings for VLSA** Here we examine the impact of time discretization on model performance. Specifically, we try to cut survival time into 4 bins or using quantile time points. The results are shown in Table 10. We can see that quantile time points or 4 time bins could often lead to poor calibration in predictive survival distribution. A poorly-calibrated distribution often indicates over-confident survival predictions. Thus, we follow the setting of Haider et al. (2020) in time discretization, *i.e.*, uniformly discretize time into $\sqrt{N_e}$ bins.

Table 10: **Different time discretization settings for VLSA**. $N_e$ is the number of patients with event.

| Time cutoff | #Cuts | TCGA | | | | | Average | | D-cal |
| | | BLCA | BRCA | GBMLGG | LUAD | UCEC | CI | MAE | Count |
|---|---|---|---|---|---|---|---|---|---|
| Quantile | 4 | 0.6238 (± 0.036) | 0.6483 (± 0.060) | 0.7962 (± 0.014) | 0.6240 (± 0.028) | 0.7295 (± 0.057) | 0.6844 | 24.40 | 0 |
| Quantile | $\sqrt{N_e}$ | **0.6255** (± 0.015) | 0.6567 (± 0.064) | **0.8006** (± 0.013) | 0.6150 (± 0.031) | 0.7300 (± 0.038) | 0.6856 | **21.92** | 2 |
| Uniform | 4 | 0.6124 (± 0.020) | 0.6458 (± 0.056) | 0.7907 (± 0.010) | 0.6312 (± 0.038) | **0.7674** (± 0.041) | 0.6895 | 23.25 | 1 |
| Uniform | $\sqrt{N_e}$ | 0.6176 (± 0.025) | **0.6652** (± 0.057) | 0.8002 (± 0.010) | **0.6370** (± 0.027) | 0.7571 (± 0.045) | **0.6954** | 25.15 | 5 |

**Comparison with Patch-GCN under the Same Number of Time Bins** We use the same number of time bins for Patch-GCN and VLSA to further compare their performance. The results are shown in Table 11. From these results, we can see that our VLSA could often outperform Patch-GCN in terms of average CI and MAE even when the number of time bins is set to the same value.

## D.7 ADOPTING PLIP AS VLM FOR SURVIVAL ANALYSIS

As another foundational VLM in computational pathology, PLIP (Huang et al., 2023) is pretrained on the pathology image-text pairs crawled from Twitter. It is the first work exploring VLMs for pathology, developed before CONCH (Lu et al., 2024). We further adopt it for VL-based methods in this experiment.

Table 11: **Comparison with Patch-GCN under the same number of time bins**.

| #Cuts | Method | TCGA | | | | | Average | | D-cal |
| | | BLCA | BRCA | GBMLGG | LUAD | UCEC | CI | MAE | Count |
|---|---|---|---|---|---|---|---|---|---|
| 4 | Patch-GCN | **0.6124** (± 0.031) | 0.6375 (± 0.033) | **0.7999** (± 0.021) | 0.5922 (± 0.053) | 0.7212 (± 0.025) | 0.6726 | 26.70 | 2 |
| | VLSA | **0.6124** (± 0.020) | **0.6458** (± 0.056) | 0.7907 (± 0.010) | **0.6312** (± 0.038) | **0.7674** (± 0.041) | **0.6895** | **23.25** | 1 |
| $\sqrt{N_e}$ | Patch-GCN | 0.6054 (± 0.049) | 0.6300 (± 0.068) | 0.7890 (± 0.020) | 0.5912 (± 0.029) | 0.6967 (± 0.030) | 0.6625 | 28.65 | 5 |
| | VLSA | **0.6176** (± 0.025) | **0.6652** (± 0.057) | **0.8002** (± 0.010) | **0.6370** (± 0.027) | **0.7571** (± 0.045) | **0.6954** | **25.15** | 5 |

The results of this experiment are shown in Table 12. From these results, we can find that, compared with CONCH, VL-based methods with PLIP often suffer from large decreases in overall performance. In particular, the zero-shot method MI-Zero$_{Surv}$ makes survival predictions like random guessing (CI = 0.519). The other three models with supervised training perform almost identically. One main reason for this could be that CONCH provides a better VL-aligned latent space than PLIP since it is pretrained on high-quality data at a larger scale (Lu et al., 2024).

We note that VL-base methods stand on the shoulder of foundational VLMs and the capability of foundational VLMs could largely determine their performance bounds. Our VLSA is no exception. When vision and language embeddings are well-aligned, language-encoded prognostic priors can effectively guide instance aggregation and improve prognostic visual representation learning.

Table 12: **Adopting PLIP as VLM for survival analysis**.

| Method | VLM | TCGA | | | | | Average | | D-cal |
| | | BLCA | BRCA | GBMLGG | LUAD | UCEC | CI | MAE | Count |
|---|---|---|---|---|---|---|---|---|---|
| MI-Zero$_{Surv}$ [†] | PLIP | 0.4990 (± 0.035) | 0.5048 (± 0.056) | **0.5374** (± 0.048) | 0.4681 (± 0.081) | 0.5855 (± 0.059) | 0.5190 | 27.22 | 4 |
| | CONCH | **0.5541** (± 0.034) | **0.5788** (± 0.028) | 0.3842 (± 0.063) | **0.5209** (± 0.049) | **0.6623** (± 0.059) | **0.5400** | **25.63** | 0 |
| CoOp | PLIP | 0.5753 (± 0.022) | 0.5458 (± 0.026) | **0.7929** (± 0.031) | **0.5838** (± 0.014) | 0.6841 (± 0.064) | 0.6364 | 29.69 | 5 |
| | CONCH | **0.5971** (± 0.033) | **0.5994** (± 0.086) | 0.7853 (± 0.015) | 0.5750 (± 0.064) | **0.6840** (± 0.070) | **0.6482** | **28.70** | 5 |
| OrdinalCLIP | PLIP | 0.5707 (± 0.037) | 0.5467 (± 0.038) | **0.7913** (± 0.035) | 0.5648 (± 0.036) | **0.6850** (± 0.053) | 0.6317 | 29.62 | 5 |
| | CONCH | **0.6037** (± 0.043) | **0.6202** (± 0.046) | 0.7893 (± 0.018) | **0.6053** (± 0.065) | 0.6836 (± 0.036) | **0.6604** | **28.01** | 5 |
| VLSA | PLIP | 0.5780 (± 0.048) | 0.5469 (± 0.034) | 0.7626 (± 0.023) | 0.5657 (± 0.013) | 0.7038 (± 0.029) | 0.6314 | 27.41 | 5 |
| | CONCH | **0.6176** (± 0.025) | **0.6652** (± 0.057) | **0.8002** (± 0.010) | **0.6370** (± 0.027) | **0.7571** (± 0.045) | **0.6954** | **25.15** | 5 |

# E   LIMITATION AND FUTURE WORK

Although promising results are obtained in this study, we note that there are still some limitations in this study. **(1)** The number of datasets and the diversity of cancer types are limited. Due to the extremely-high resolution of histopathology WSIs, it is usually challenging to collect tens of thousands of images and preprocess them into feasible training data. **(2)** The textual descriptions about cancer-specific prognostic prior are derived from GPT-4o. They may be lacking in knowledge completeness and accuracy. More holistic and accurate prognostic prior knowledge are likely to provide stronger guidance for weakly-supervised MIL and perform better in survival prediction. **(3)** The foundational VLM that our empirical results rely on is CONCH (Lu et al., 2024), a recent state-of-the-art VLM for CPATH. More pathology VLMs may be needed to further validate our VLSA. **(4)** Many state-of-the-art approaches for WSI classification (Zhang et al., 2022; Xiong et al., 2023; Tang et al., 2023; Qu et al., 2023; Shi et al., 2024; Fourkioti et al., 2024; Tang et al., 2024) have shown strong capabilities in representation learning. VLSA could take inspiration from or combine with them to further improve its performance in SA tasks.

We note that VL-base approaches stand on the shoulder of foundational VLMs and their performances are closely correlated with the capability of VLMs. Our VLSA is no exception. We strongly believe that VLSA could further refresh the state-of-the-art performances in SA tasks with the continual advancements of VLMs. In the future, we will continue to follow the frontier of VLMs in CPATH and leverage advanced tools to study the potential of VLSA in cancer prognosis.

