# OpenReview forum: "Interpretable Vision-Language Survival Analysis with Ordinal Inductive Bias for Computational Pathology"
_ICLR.cc/2025/Conference — ICLR 2025 Poster_

### Official Review · Reviewer_q9C6 · 2024-10-31

**Soundness:** 2
**Presentation:** 3
**Contribution:** 2
**Rating:** 6
**Confidence:** 5

**Summary:**

The paper proposes a Vision-Language-Based survival analysis (VLSA) paradigm, addressing two key challenges: data scarcity and weakly supervision of gigapixel whole slide image (WSI) analysis. The framework leverages foundational vision-language models with three key innovations: (1) language-encoded prognostic priors to guide instance aggregation, (2) ordinal survival prompt learning for encoding continuous time-to-event labels, and (3) ordinal incidence function prediction. The approach demonstrates strong performance improvements while maintaining interpretability through Shapley values.

**Strengths:**

1.	The authors have proposed a complete framework for VLM-compatible survival analysis, with novel components for handling continuous time labels and maintaining ordinal relationships
2.	The comprehensive experiments demonstrate the efficacy of the method and each proposed module through thorough ablation studies
3.	The method incorporates interpretability using Shapley values, which is crucial for clinical applications, and provides explanations through natural language descriptions

**Weaknesses:**

1.	The fundamental necessity of language priors for addressing data scarcity requires stronger justification. While the authors argue that language priors can mitigate data scarcity, they do not sufficiently demonstrate why this approach is superior to using large-scale vision-only foundation models (e.g., UNI). A direct comparison with vision-only approaches would strengthen this argument
2.	The experimental validation has some limitations in terms of baseline comparisons. The pre-trained VLMs used for comparison (CoOp, OrdinalCLIP) are trained on natural images rather than pathology data, leading to a suboptimal performance
3.	The survival analysis evaluation omits standard clinical protocols, notably Kaplan-Meier analysis and statistical significance testing between risk stratification groups. It would be better to see these experimental results and analysis
4.	The comparative analysis excludes several state-of-the-art methods in WSI classification, including DTFD-MIL [1], HAG-MIL [2], MHIM-MIL [3], and recent VLM approaches such as TOP [4] and ViLa-MIL [5]

Miscellaneous:
1.	The terminology regarding "cross-attention" in line 163 requires revision, as similarity matching differs from the so-called cross-attention mechanisms
2.	Figure 1 contains rendering issues in “context prompt” that should be addressed


[1] DTFD-MIL: Double-Tier Feature Distillation Multiple Instance Learning for Histopathology Whole Slide Image Classification

[2] Diagnose Like a Pathologist: Transformer-Enabled Hierarchical Attention-Guided Multiple Instance Learning for Whole Slide Image Classification

[3] Multiple Instance Learning Framework with Masked Hard Instance Mining for Whole Slide Image Classification

[4] The Rise of AI Language Pathologists: Exploring Two-level Prompt Learning for Few-shot Weakly-supervised Whole Slide Image Classification

[5] Dual-scale Vision-Language Multiple Instance Learning for Whole Slide Image Classification

**Questions:**

1. The paper focuses on vision-language models while foundational vision models like UNI [1] have shown strong performance in computational pathology. Could the authors discuss their rationale for prioritizing vision-language models and perhaps provide comparative experiments with vision-only foundation models to better justify this architectural choice?
2. Given the recent success of multimodal approaches in survival analysis (e.g., MCAT [2], MOTCAT [3], CMTA [4], MoME [5]), could the authors address why these methods were not included in the baseline comparisons? A comparison with these state-of-the-art multimodal methods would help better contextualize VLSA's contributions to the field.
3. To strengthen the clinical relevance of the results, could the authors provide Kaplan-Meier survival curves for risk stratification groups, along with appropriate statistical significance tests (e.g., log-rank test)? This would help validate the model's practical utility in clinical settings.
4. Regarding the experimental comparisons within the VLM group, the baseline models (CoOp and OrdinalCLIP) were originally trained on natural images rather than pathology data. Could the authors discuss how this might affect the fairness of the comparisons and perhaps provide additional experiments with VLMs specifically trained or fine-tuned on pathological images?

[1] Towards a General-Purpose Foundation Model for Computational Pathology

[2] Multimodal Co-Attention Transformer for Survival Prediction in Gigapixel Whole Slide Images

[3] Multimodal Optimal Transport-based Co-Attention Transformer with Global Structure Consistency for Survival Prediction

[4] Cross-Modal Translation and Alignment for Survival Analysis

[5] Mixture of Multimodal Experts for Cancer Survival Prediction

---

> ### Author Response · Authors · 2024-11-26
> **Reply to Reviewer's Comments**
>
> Thanks for your constructive comments and encouraging feedback.
>
> We provide a detailed point-to-point response to your questions and concerns.

---

> ### Author Response · Authors · 2024-11-26
> **Reply to Reviewer's Comments**
>
> **W1**: The fundamental necessity of language priors for addressing data scarcity requires stronger justification. While the authors argue that language priors can mitigate data scarcity, they do not sufficiently demonstrate why this approach is superior to using large-scale vision-only foundation models (e.g., UNI). A direct comparison with vision-only approaches would strengthen this argument.
>
> ---
>
> **AW1**: Thanks for your detailed feedback and insightful comments.
>
> We would like to clarify that we actually have presented a direct comparison with vison-only approaches to support our argument regarding data scarcity mitigation. Next, we will first (1) explain our experiments for this direct comparison, then (2) discuss the comparison with UNI, and finally (3) give our revisions, to address your concerns.
>
> **(1) Our experiments for a direct comparison with vison-only approaches**
>
> We state that VLMs are promising for alleviating the challenge of scarce training data, as written at lines 67 and 68. To justify it, **we design a scenario** where only a few samples can be used for training to simulate the case of data scarcity, called few-shot learning (FSL).
>
> In this FSL scenario, we directly compare our VLSA with vision-only approaches, as shown in Table 4 on page 9. These compared vision-only approaches share characteristics as follows:
> - Language priors are not used in them: **only WSI visual features** are used to predict the survival of patients, namely, they are vision-only approaches.
> - The visual features extracted by the CONCH’s vision encoder are used for them. It is worth noting that, these visual features are **NOT in the VL embedding space**, because they are not projected into the VL embedding space (which involves an additional VL projection layer in CONCH) according to the convention of VLMs, as written at lines 320 and 321 on page 6.
>
> In other words, although CONCH is a large-scale VL foundation model, its text encoder and critical VL projection layer are NOT used for these compared vision-only approaches; instead, only the large-scale pretrained vision model in CONCH is used. This means that, it is the approaches that only use the large-scale vision foundation model are compared.
>
> The comparative results (Table 4) show that our VLSA often outperforms vision-only approaches in FSL scenarios, supporting our argument regarding data scarcity mitigation.
>
> In fact, the claim that VL foundation models often show the advantage of data efficiency (or data scarcity mitigation) has been demonstrated in many works, as written at lines 66 and 67 on page 2. We are inspired by it and thus introduce VL models into WSI survival analysis, given that WSI data is often scarce for training.
>
> **(2) Discussion on the comparison with UNI**
>
> UNI, as another large-scale vision foundation model, has shown great success in computational pathology. Nevertheless, there are some potential concerns when comparing with UNI:
> - Our approach is **driven by VL foundation models**. It thus cannot be adapted to the vision foundational model UNI for further comparisons.
> - The vision-only approaches with UNI features (rather than CONCH visual features) seem to be meaningless baselines. Because the comparison, *i.e.*, existing vision-only approaches with UNI features vs. our vision-language approach with CONCH VL features, makes **two critical factors coupled**: i) the power of pretrained features (UNI vs. CONCH) and ii) the power of the approach itself (existing approaches vs. our VLSA framework). This means that, the comparison **cannot** be used to confirm whether the performance improvement results from the approach itself.
> - Instead, our current presented comparison, *i.e.*, existing vision-only approaches with CONCH visual features vs. our vision-language approach with CONCH VL features, could be used to verify the effectiveness of the framework itself, because the factor, *i.e.*, the power of pretrained features, is not involved in this comparison.
>
> **(3) Our revision**
> - Section 4.5 “Few-Shot Survival Prediction” on page 10 of our revised paper: We have added a few sentences to describe the motivation behind designing FSL experiments, as follows:
> > Here we focus on a few-shot learning (FSL) scenario in which only a few samples can be used for training. We use this scenario to simulate the case of scarce training data and evaluate the performance of different SA methods in terms of data efficiency.
>
> - The caption of Table 4 on page 10 of our revised paper: We have revised it to “Few-shot survival prediction (data efficiency evaluation)”.

---

> ### Author Response · Authors · 2024-11-26
> **Reply to Reviewer's Comments**
>
> **W2**: The experimental validation has some limitations in terms of baseline comparisons. The pre-trained VLMs used for comparison (CoOp, OrdinalCLIP) are trained on natural images rather than pathology data, leading to a suboptimal performance
>
> ---
>
> **AW2**: As written at lines 320 and 321 on page 6, **all methods use the same vision encoder from CONCH**. In other words, the pretrained VLM used for the baselines (CoOp and OrdinalCLIP) is CONCH, rather than CLIP, which is the same as our VLSA.
>
> As you mentioned, using CLIP as the pretrained VLM would lead to suboptimal performance and unfair comparisons. Considering this issue in comparisons, we thus specially replace the pretrained VLM (*i.e.*, CLIP) originally used in CoOp and OrdinalCLIP with CONCH to fairly compare different VLM-driven methods with our VLSA.

---

> ### Author Response · Authors · 2024-11-26
> **Reply to Reviewer's Comments**
>
> **W3**: The survival analysis evaluation omits standard clinical protocols, notably Kaplan-Meier analysis and statistical significance testing between risk stratification groups. It would be better to see these experimental results and analysis
>
> ---
>
> **AW3**: Good suggestion. Thanks for your kind reminder.
>
> According to your suggestion, we have conducted Kaplan-Meier analysis and statistical significance testing between risk stratification groups.
>
> The results, including KM curves and statistical significance test, have been added to our revised paper and now are **shown in Figure 5 on page 25 of our revised paper**. From the additional results in Figure 5, we can observe that VLSA can discriminate between high- and low-risk patients with a significant difference (P-Value $<0.05$ by a log-rank test).
>
> We appreciate your suggestion that does help us to further improve this paper.

---

> ### Author Response · Authors · 2024-11-26
> **Reply to Reviewer's Comments**
>
> **W4**: The comparative analysis excludes several state-of-the-art methods in WSI classification, including DTFD-MIL [1], HAG-MIL [2], MHIM-MIL [3], and recent VLM approaches such as TOP [4] and ViLa-MIL [5]
>
> ---
>
> **AW4**: Thanks for your comments. We also appreciate your efforts in listing them above for our reference.
>
> To address your concerns, we would like to first (1) discuss these listed state-of-the-art methods, then (2) clarify our baseline choice, and finally (3) give our revision details.
>
> **(1) State-of-the-art methods for WSI classification**
>
> After carefully reading these papers and checking the details of these approaches, we observe that most of these approaches originally proposed for WSI classification share the characteristic: one or more critical components of them or their key idea are **specially designed for classification problems**, preventing them from being directly adapted to survival analysis (SA) problems.
>
> Concretely,
> - DTFD-MIL: it relies on the class activation map for the $i$-th class in model inference. The class activation map is obtained through the mechanism of Grad-CAM that is widely used for classification and cannot be directly extended to SA.
> - MHIM-MIL: its key idea is to iteratively mining hard-to-classify instances. However, in survival analysis, the task is not to find the instances that are hard to classify, but to collect the instances that are most relevant to prognosis. For example, given the image patch (instance) which presents the formation of new blood vessels without tumorous cells, it is certainly an easy instance that is not within the scope of MHIM-MIL, but it usually reflects prognosis and is concerned in SA.
> - TOP and ViLa-MIL: they are VLM-based approaches and heavily rely on the textual prompts specially designed for classification tasks, e.g., cancer detection or subtyping. This prevents these two approaches from SA because SA is concerned with other more aspects of tumor diseases (as listed in Table 7 on pages 20 and 21).
>
> For HAG-MIL, while it does not contain the designs tailored for classification tasks, its input involves the patches at multiple resolutions. By contrast, all other compared methods adopt the patches at only one resolution. Such fundamental discrepancy at the input end could lead to unfair comparisons for all other compared single-resolution methods.
>
> Nevertheless, all of these methods have very appealing basic ideas, *e.g.*, multi-scale features and mitigating noisy instances. These could be borrowed and integrated with VLSA for further performance improvements. We would like to leave this as our future work and discuss them in the Section "Limitation and Future Work". Please refer to the last part (3) for our revision details.
>
> Although the above listed SOTA works cannot be directly adapted to SA problems, we still choose an additional SOTA baseline, R$^2$T-MIL (Tang et al., CVPR, 2024), for more comparisons, according to the suggestion of Reviewer f7nn. Its additional results are shown in Table 1, 4, and 5 of our revised paper.
>
> **(2) Our baseline choice**
>
> We would like to clarify that we actually follow Song *et al.* (CVPR, 2024) to choose the most relevant vision-only approaches for comparisons.
>
> Moreover, this work **studies the survival prediction tasks on WSIs**. Thus, we put more attention on the most relevant state-of-the-art SA methods, compared with classification ones.
>
> **(3) Our revision details**
>
> In spite of the above discussed, we would like to closely follow those state-of-the-art classification methods and discuss them in Section 5 “Limitation and Future Work”, as follows:
> > (4) Many state-of-the-art approaches for WSI classification (Zhang et al., 2022; Xiong et al., 2023; Tang et al., 2023; Qu et al., 2023; Shi et al., 2024; Fourkioti et al., 2024; Tang et al., 2024) have shown strong capabilities in representation learning. VLSA could take inspiration from them to further improve its performance in SA.
>
> This revision can be seen at lines 550-553 on page 11 of our revised paper.
>
> [Reference A] Song et al. Morphological prototyping for unsupervised slide representation learning in computational pathology. CVPR 2024.
>
> [Reference B] Tang et al. Feature Re-Embedding: Towards Foundation Model-Level Performance in Computational Pathology. CVPR 2024.

---

> ### Author Response · Authors · 2024-11-26
> **Reply to Reviewer's Comments**
>
> **W5.1**: Miscellaneous: 1. The terminology regarding "cross-attention" in line 163 requires revision, as similarity matching differs from the so-called cross-attention mechanisms
>
> ---
>
> **AW5.1**: We sincerely appreciate your careful reading and kind reminder.
>
> Now we have revised the inappropriate “cross-attention” into “**similarity matching and weighted averaging**”, according to your helpful suggestion.
>
> This revision can be seen at line 165 on page 4 of our revised paper.

---

> ### Author Response · Authors · 2024-11-26
> **Reply to Reviewer's Comments**
>
> **W5.2**: Miscellaneous: 2. Figure 1 contains rendering issues in “context prompt” that should be addressed
>
> ---
>
> **AW5.2**: We are sorry for this inconvenience.
>
> In Figure 1, our intention is to adopt question marks to represent the meaning of unknown parameters. However, such a way could lead to possible confusion, such as the rendering issues you mentioned.
>
> To remedy this, we have improved Figure 1 by **replacing the appropriate question marks with the notation of context prompt**. This revision can be seen in Figure 1 on page 3 of our revised paper.
>
> Thanks for your helpful suggestion.

---

> ### Author Response · Authors · 2024-11-26
> **Reply to Reviewer's Comments**
>
> **Q1**: The paper focuses on vision-language models while foundational vision models like UNI [1] have shown strong performance in computational pathology. Could the authors discuss their rationale for prioritizing vision-language models and perhaps provide comparative experiments with vision-only foundation models to better justify this architectural choice?
>
> ---
>
> **AQ1**: Thanks for your detailed feedback.
>
> To address your concerns, next, we would like to first (1) explain our rationale for prioritizing VLMs, and then (2) discuss the comparison with UNI.
>
> **(1) Rationale for prioritizing VLMs**
> - Our proposed approach, VLSA, is **a VLM-driven framework**. This means that VLSA should be built upon a VL foundation model. However, UNI is a vision-only foundation model that cannot provide aligned textual and visual representations. Therefore, if using UNI as the foundation model of VLSA, VLSA would be unable to provide reasonable predictions for survival analysis because its prediction is derived from the similarity measurement between textual and visual representations.
> - This work is inspired by **the notable merits of VLMs**, as described in the Section Introduction. Concretely, VLMs often show surprisingly-good performance in zero-shot classification or few-shot learning, demonstrating their advantage of data efficiency. Additionally, VLMs could provide auxiliary signals by encoding language priors. These auxiliary signals have the potential to improve weakly-supervised multi-instance learning. Given these notable merits, we prioritize VLMs.
>
> **(2) Discussion on the comparison with UNI**
>
> This part has been explained and provided in our AW1 “(2) Discussion on the comparison with UNI”. Please refer to that for a detailed response. Thanks.

---

> ### Author Response · Authors · 2024-11-26
> **Reply to Reviewer's Comments**
>
> **Q2**: Given the recent success of multimodal approaches in survival analysis (e.g., MCAT [2], MOTCAT [3], CMTA [4], MoME [5]), could the authors address why these methods were not included in the baseline comparisons? A comparison with these state-of-the-art multimodal methods would help better contextualize VLSA's contributions to the field.
>
> ---
>
> **AQ2**: Thanks for your comments.
>
> We would like to first (1) clarify the fundamental discrepancy between our VLSA and current multi-modal SA models and then (2) provide our revisions, to resolve your concerns.
>
> **(1) Clarification on the fundamental discrepancy between our VLSA and current multi-modal SA models**
>
> Current multi-modal SA models, *e.g.*, MCA, MOTCAT, CMTA, and MoME, generally aim to learn the multi-modal representation that **embeds and fuses the features of two modalities** like Gene and WSI. However, our VLSA does not involve multi-modal representation learning. In other words, **in VLSA, there does NOT exist the multi-modal representation that **embeds** and **fuses** the features of two modalities like vision and language**.
>
> To be more specific,
>
> i) **The VLSA’s vision-end does not involve multi-modal representation learning**
>
> Although our VLSA adopts text description in vision-end, the text description is used to guide aggregation and it is **not** an additional modality from the perspective of multi-model representation learning.
>
> The detailed explanation for this is as follows:
> - Our text description is NOT utilized as a conventional text modality that is generally **fused with the other modality** to produce multi-modal representations for better prediction. In fact, our text description is only utilized to provide the weight of each instance for instance aggregation, just like the role of attention weights in aggregation, as written in Eq. (2) on page 4. This means that, our text description has **not been fused with WSI features** to obtain multi-model representations. Our final representation in vision-end **always comes from WSI features only**, not a multi-modal representation.
> - **Current multimodal survival models** using WSI and gene data, such as MCA, MOTCAT, CMTA, and MoME, are standard multi-model representation learning frameworks, in which two modalities (WSI and gene data) are **explicitly fused** to produce multi-modal representations for better prediction.
>
> ii) **Multi-modal representation learning is also not involved in VLSA’s survival prediction**
>
> In VLSA, the prediction is derived by first measuring the similarity between vision and language features. Such operation, *i.e.*, the similarity measurement between vision and language modality, is totally different from multi-modal representation calculation, because
> - multi-modal representation calculation **generally fuses the features of two modalities for prediction** (often by feature addition),
> - **while similarity measurement is an operation for feature fusion**.
>
> **(2) Our revision details**
>
> To make these clearer, we have added a description to discuss existing multi-modal survival models and analyze the difference between them and our VLSA.
>
> This revision can be seen at lines 185-190 on page 4 of our revised paper, as follows:
> > (3) Difference from multi-modal representation learning: note that our text description of prognostic prior is only utilized to provide the weight of each instance for aggregation and mainly plays a guidance role. The final representation in vision-end only comes from WSI features, not a multi-modal representation fusing text and vision features. This is a fundamental discrepancy between our method and current multi-modal survival models, such as MCAT (Chen et al., 2021b), MOTCat (Xu & Chen, 2023), CMTA (Zhou & Chen, 2023), MoME (Xiong et al., 2024), and MMP (Song et al., 2024b).

---

> ### Author Response · Authors · 2024-11-26
> **Reply to Reviewer's Comments**
>
> **Q3**: To strengthen the clinical relevance of the results, could the authors provide Kaplan-Meier survival curves for risk stratification groups, along with appropriate statistical significance tests (e.g., log-rank test)? This would help validate the model's practical utility in clinical settings.
>
> ---
>
> **AQ3**: We agree with your insightful feedback and appreciate your kind reminder.
>
> As replied in our AW3, we have added additional results, including the Kaplan-Meier survival curves of two risk groups and the results of log-rank tests, into our revised paper. Please refer to **Figure 5 on page 25 of our revised paper**.
>
> Thanks again for your valuable suggestions.

---

> ### Author Response · Authors · 2024-11-26
> **Reply to Reviewer's Comments**
>
> **Q4**: Regarding the experimental comparisons within the VLM group, the baseline models (CoOp and OrdinalCLIP) were originally trained on natural images rather than pathology data. Could the authors discuss how this might affect the fairness of the comparisons and perhaps provide additional experiments with VLMs specifically trained or fine-tuned on pathological images?
>
> ---
>
> **AQ4**: In fact, we adopt **the same CONCH for the VLM group** (*e.g.*, CoOp and OrdinalCLIP) in experimental comparisons, as replied in our AW2.
>
> We would like to appreciate your comments with many details and elaborations.

---

> > ### Comment · Reviewer_q9C6 · 2024-12-02
> >
> > I would like to express my gratitude to the authors for their comprehensive response, which largely addressed my concerns. Consequently, I have raised my scores.
> >
> > Despite this, I still have some further suggestions for the authors to enhance the paper’s quality.
> >
> > 1. The potential for broader impact could be realized if the genomic data were utilized, as the current paradigm integrates both WSIs and genomics as input sources, as my previous review.
> >
> > 2. There appears to be an error in line 564.

---

> > > ### Author Response · Authors · 2024-12-02
> > >
> > > Dear Reviewer q9C6,
> > >
> > > We are happy to hear that our responses have resolved you concerns. Thanks for your constructive and helpful suggestions.
> > >
> > > We will carefully revise our paper according to your further suggestions.
> > >
> > > Again, we deeply appreciate your valuable time and efforts you have dedicated as a reviewer for ICLR 2025.
> > >
> > > Best

---

### Official Review · Reviewer_f7nn · 2024-11-01

**Soundness:** 2
**Presentation:** 2
**Contribution:** 3
**Rating:** 6
**Confidence:** 4

**Summary:**

This paper aims to overcome two challenges in survival analysis including overfitting due to data scarcity in CPATH, and insufficient supervision for survival analysis. To address the challenge of scarce training data, the authors proposed to apply visual-language models to survival analysis by utilizing the power of the recent VL pathology foundation model. To compensate for supervision signals, the authors introduced some extra prior descriptions of prognosis by querying GPT-4o. Some ordinal prompts are designed to adapt VL models to survival analysis, although their performance improvements are marginal.

**Strengths:**

1. This is the first work to adapt the VL foundation model to survival tasks by designing ordinal prompts tailored to survival analysis tasks.
2. The introduction of prognostic prior description was empirically validated to significantly improve the performance of survival analysis.
3. The motivation is basically clear although the design of the ordinal incidence function is a bit confusing.
4. Interpretation brought by language priors for survival tasks is interesting and makes the prediction more transparent.

**Weaknesses:**

1. The motivation of ordinal IF is confusing. The authors assume there is a consecutive decline in probability for those classes away from the time when the event occurs, which raises doubts about its reasonableness when events occur multiple times. This assumption is only concerned with the event that first occurs.
2. Ordinal Survival Prompt seems to ignore the status of event indicators (i.e. censorship) when designing the prompts. For different censorships, there should be different outcomes reflected in the survival prompts even when their follow-up time are the same. When the case is censored, the only thing we can know is the event didn’t occur before the follow-up time. For example, we cannot simply assign a case with a short follow-up time into a bin of “very poor prognosis”.
3. Experiment comparison is **NOT** adequate in the following aspects:

    a. To validate the effectiveness of prompts tailored to survival tasks, a stronger baseline equipped with naive prompts that simply formulate survival tasks as classification tasks, should be compared.

    b. **To support the authors’ claim that "highly-expressive modern networks may cause overfitting when facing present small WSI data”**, the current “highly-expressive” models should be compared with the proposed method. Furthermore, the extra text description generated by GPT-4o should be an additional modality. Therefore, some representative multimodal survival models should be compared, **which can validate the efficacy of utilizing the pathology foundation model**. I understand there have been a few models that integrated WSI and text data in the past. However, by replacing the gene data with text data, some multimodal survival models using WSI and gene data can be adapted for comparison, e.g. MCAT [1], MOTCat [2], MMP [3] etc.

    c. Unimodal MIL models are a bit out-of-date. More “highly-expressive” SOTA methods should be compared, such as CAMIL[4], R2T [5], etc.

4. From the results of Tables 3 and 8, performance gains from two ordinal prompts are marginal or even getting worse. The reason should be justified.
5. Comparisons in Few-shot learning may not be fair. The capability of few-shot prediction greatly relies on how the visual and text spaces are well-aligned. CONCH is a pathology VL model that aligns visual and text spaces well. However, CoOp and OrdinalCLIP are equipped with ABMIL, which is not aligned with the text space, for few-shot prediction by using such few samples. For a fair comparison, the author should replace prompts with the ones in CoOp and OrdinalCLIP only based on CONCH to validate the efficacy of survival prompts.
6. The code is unavailable which may affect the reproduction.

If my concerns could be addressed well, I would adjust my scores.

[1] Multimodal co-attention transformer for survival prediction in gigapixel whole slide images, ICCV, 2021

[2] Multimodal optimal transport-based co-attention transformer with global structure consistency for survival prediction, ICCV, 2023

[3] Multimodal Prototyping for cancer survival prediction, ICML, 2024

[4] CAMIL: Context-Aware Multiple Instance Learning for Cancer Detection and Subtyping in Whole Slide Images, ICLR, 2024

[5] Feature re-embedding: Towards foundation model-level performance in computational pathology, CVPR, 2024

**Questions:**

1. I‘m not sure how the authors applied MI-Zero to survival analysis, which was supposed to be designed for classification.
2. What’s the difference in prompts between OrdinalCLIP and Ordinal Survival Prompt?
3. In equations 2 and 3, how can the model ensure learning multi-level representations? It seems there is no constraint to achieving this goal.
4. CONCH is a large foundation model pretrained on a great amount of data including public data. In this case, to avoid data contamination, how can the author ensure test samples never occur in the pretraining materials of CONCH in the few-shot setting?
5. I’m confused about what the specific events are referring in these experiments. If overall survival, there should be only one event. However, the authors mentioned that the events can occur multiple times. Is it longitudinal data?
6. The motivation for using EMD should be elaborated. For example, why is EMD aware of ordinality?
7. It is difficult to understand Figure 3(a) is trying to explain. What do the horizontal and vertical axes represent, respectively?

---

> ### Author Response · Authors · 2024-11-26
> **Reply to Reviewer's Comments**
>
> Thanks for your constructive comments and encouraging feedback.
>
> We provide a detailed point-to-point response to your questions and concerns.

---

> > ### Comment · Reviewer_f7nn · 2024-11-26
> >
> > Thanks for the authors’ careful responses, which address some of my concerns. However, there are still some concerns, which can be found in the point-to-point comments.

---

> > > ### Author Response · Authors · 2024-12-02
> > > **Reply to Further Comment by Reviewer f7nn**
> > >
> > > **For W3a**: As you mentioned in the Q1 response about how to apply MI-Zero to survival, you simply divide the survival labels into several survival classes. Similarly, the prompts used in MI-Zero can be applied here as naive prompts. I agree that censorship cannot be directly predicted from WSI features. However, the morphological features of WSIs for censored patients should differ from those for uncensored patients even with the same survival/observation time. Therefore, they should be regarded as two separate classes. This has been validated in the previous work, PIBD [1], where PIB held the discriminative assumption that combinations of different 't' (time) and 'c' (censorship) are treated as distinct categories
> > >
> > > ---
> > >
> > > **Response**: Thanks for your further explanations. With your detailed explanations, we have understood what you mean by naïve prompts. Thanks again.
> > >
> > > Next, we would like to first (1) show additional experiments for the stronger baseline equipped with naive prompts and then (2) give our revision details.
> > >
> > > **(1) Additional experiments for the stronger baseline equipped with naive prompts**
> > >
> > > We have followed your instructions to conduct additional experiments for the stronger baseline equipped with naïve prompts. Concretely, we apply the prompts used in MI-Zero as naive prompts for vision-language-based survival analysis (SA), as you mentioned. This baseline is called ABMIL$_\\text{Prompt}$, indicating that ABMIL is adapted to VLM-based learning by applying naïve prompts in language-end for survival prediction.
> > >
> > > The additional experimental results of ABMIL$_\\text{Prompt}$ are as follows:
> > >
> > > (a) **Survival prediction**
> > >
> > > **Table 1**: Main comparative results
> > > | Method | TCGA-BLCA | TCGA-BRCA | TCGA-GBMLGG | TCGA-LUAD | TCGA-UCEC | Average CI | Average MAE | D-cal Count|
> > > |:--:|:--:|:--:|:--:|:--:|:--:|:--:|:--:|:--:|
> > > | ABMIL$_\\text{Prompt}$ | 0.5717 ($\pm$ 0.035) | 0.6215 ($\pm$ 0.084) | 0.7825 ($\pm$ 0.020) | 0.5984 ($\pm$ 0.052) | 0.6762 ($\pm$ 0.063) | 0.6500 | 25.68 | 4 |
> > > | VLSA | **0.6176** ($\pm$ 0.025) | **0.6652** ($\pm$ 0.057) | **0.8002** ($\pm$ 0.010) | **0.6370** ($\pm$ 0.027) | **0.7571** ($\pm$ 0.045) | **0.6954** | **25.15** | **5** |
> > >
> > > The above results show that, ABMIL$_\\text{Prompt}$ obtains an average CI of 0.6500, which suggests that it is indeed a strong baseline. Nevertheless, VLSA could often outperform this baseline, with an improvement of 4.54% in average CI.
> > >
> > > (b) **Few-shot learning scenarios**
> > >
> > > Moreover, we have also conducted additional experiments for the additional baseline ABMIL$_\text{Prompt}$ in few-shot learning (FSL) scenarios. Their results are as follows:
> > >
> > > **Table 4**: Few-shot survival prediction (data efficiency evaluation)
> > > | Method | # Shots = 1 | # Shots = 2 | # Shots = 4 | # Shots = 8 | # Shots = 16 | Average | Full-shots |
> > > |:--:|:--:|:--:|:--:|:--:|:--:|:--:|:--:|
> > > | ABMIL$_\text{Prompt}$ | 0.5314 | 0.5588 | 0.5737 | 0.5958 | 0.6068 | 0.5733 | 0.6500 |
> > > | VLSA | **0.5787** | **0.6068** | **0.6271** | **0.6465** | **0.6592** | **0.6237** | **0.6954** |
> > >
> > > (c) **Model efficiency evaluation**
> > >
> > > The evaluation results of model efficiency for ABMIL$_\text{Prompt}$ are given as follows:
> > >
> > > **Table 5**: Model efficiency
> > > | Model | # Params | # MACs |
> > > |:--:|:--:|:--:|
> > > | ABMIL$_\text{Prompt}$ | 394.24K | 2.67G |
> > > | VLSA | **284.16K** | **0.25G** |
> > >
> > > **(2) Our revision**
> > >
> > > We have listed ABMIL$_\text{Prompt}$ as an additional baseline for comparisons, so its relevant parts will be updated, as mentioned above.
> > >
> > > Concretely, we will make revisions as follows:
> > > - The description of baselines at the top of page 7: We will update the description of ABMIL$_\text{Prompt}$, as follows:
> > > > ABMIL$_\text{Prompt}$ is a baseline of ABMIL with naive (not learnable) textual prompts for prediction in language-end.
> > >
> > > - Table 1 on page 7: The main results of ABMIL$_\text{Prompt}$ shown above will be added to it.
> > > - Table 4 on page 10: The FSL results of ABMIL$_\text{Prompt}$ shown above will be added to it.
> > > - Table 5 on page 10: The model efficiency results of ABMIL$_\text{Prompt}$ shown above will be added to it.

---

> > > ### Author Response · Authors · 2024-12-02
> > > **Reply to Further Comment by Reviewer f7nn**
> > >
> > > Continue from Above
> > >
> > > **(2) Our revision**
> > >
> > > We will add a new subsection “Comparison with Multi-Modal Survival Analysis Models” in Appendix E to show this experiment, as follows:
> > >
> > > > In this experiment, we compare our VLSA with multi-modal SA models. The textual description of prognostic priors is adopted in VLSA to provide extra auxiliary signals for weakly-supervised learning. An alternative approach to leveraging these textual priors is multi-modal representation learning, *i.e.*, integrating and fusing visual and textual features. Accordingly, here we compare our proposed method with multi-modal SA models.
> > >
> > > > Four representative multi-modal SA models are adopted as the baselines for comparisons: MCAT (Chen et al., 2021b), MMP (Trans.) (Song et al., 2024b), MMP (OT) (Song et al., 2024b), and MOTCat (Xu & Chen, 2023). We replace their gene input with the language-encoded prognostic priors, *i.e.*, $E_{\text{text}}(\mathcal{T}_\text{prog})$, used in VLSA. In addition to this setting, the other settings and model implementations keep the same as those in their official codes.
> > >
> > > > The results of this experiment are shown in Table 13. These results show that our VLSA could often perform better than the compared multi-modal SA models, with an improvement of 2.85% over the runner-up in average CI. This improvement could indicate the superiority of our scheme.
> > >
> > > The Table 13 given above will also be added into our revised paper. Thanks again for your valuable suggestions.
> > >
> > > [Reference A] Chen et al. Multimodal Co-Attention Transformer for Survival Prediction in Gigapixel Whole Slide Images. ICCV 2021.
> > >
> > > [Reference B] Xu & Chen. Multimodal Optimal Transport-based Co-Attention Transformer with Global Structure Consistency for Survival Prediction. ICCV 2023.
> > >
> > > [Reference C] Song et al. Multimodal Prototyping for cancer survival prediction. ICML 2024.

---

> > > ### Author Response · Authors · 2024-12-02
> > > **Reply to Further Comment by Reviewer f7nn**
> > >
> > > **For W4**: I disagree with the authors’ reason why Table 3 shows marginal CI improvements. The authors claimed that prognostic priors have alleviated ordinal inductive bias, leading to performance saturation. On one hand, if this statement holds true, then the proposed two bias terms are unnecessary. On the other hand, I disagree with this statement because the prognostic priors presented in Appendix D.4 have no relation to ordinal bias. Additionally, C-index exactly focuses on the accuracy regarding the ordinality of the predicted pairs, which should be more sensitive to ordinal inductive biases than MAE. Specifically, C-index is calculated by evaluating whether the predictive model accurately reflects these ordinal relationships. Therefore, I disagree with the author’s explanations that “One of ordinal inductive biases, i.e., ordinal incidence function, often helps to improve MAE, not CI”.
> > >
> > > ---
> > >
> > > **Response**: Thanks for your detailed feedback.
> > >
> > > To address your concerns, we would like to first (1) present the combination of Table 3 and Table 8 for intuitive and direct comparisons and then (2) employ the combined table to clarify the two possible reasons stated in our AW4.
> > >
> > > **(1) Combining Table 3 and Table 8 for intuitive and direct comparisons**
> > >
> > > **Table 3** and **Table 8**: Ablation study on two ordinal inductive bias terms when using **language-encoded prognostic priors** ($P$) in VLSA or not.
> > > | Ordinality Prompt | Ordinality Proba. | TCGA-BLCA | TCGA-BRCA | TCGA-GBMLGG | TCGA-LUAD | TCGA-UCEC | Average CI | Average MAE | D-cal Count|
> > > |:--:|:--:|:--:|:--:|:--:|:--:|:--:|:--:|:--:|:--:|
> > > | **w/o $P$** |
> > > | | | 0.5971 ($\pm$ 0.033) | 0.5994 ($\pm$ 0.086) | 0.7853 ($\pm$ 0.015) | 0.5750 ($\pm$ 0.064) | 0.6840 ($\pm$ 0.070) | 0.6482 | 28.70 | 5 |
> > > | $\checkmark$ | | 0.6037 ($\pm$ 0.043) | **0.6202** ($\pm$ 0.046) | 0.7893 ($\pm$ 0.018) | **0.6053** ($\pm$ 0.065) | 0.6836 ($\pm$ 0.036) | 0.6604 | 28.01 | 5 |
> > > | | $\checkmark$ | 0.5997 ($\pm$ 0.033) | 0.6049 ($\pm$ 0.104) | 0.7846 ($\pm$ 0.016) | 0.5818 ($\pm$ 0.056) | 0.6769 ($\pm$ 0.065) | 0.6496 | 26.99 | 3 |
> > > | $\checkmark$ | $\checkmark$ | **0.6083** ($\pm$ 0.047) | 0.6180 ($\pm$ 0.046) | **0.7908** ($\pm$ 0.017) | 0.6048 ($\pm$ 0.063) | **0.6908** ($\pm$ 0.035) | **0.6625** | **26.78** | 5 |
> > > | **w/ $P$** |
> > > | | | 0.6128 ($\pm$ 0.028) | 0.6304 ($\pm$ 0.065) | 0.7927 ($\pm$ 0.015) | 0.6351 ($\pm$ 0.041) | **0.7606** ($\pm$ 0.037) | 0.6863 | 26.76 | 5 |
> > > | $\checkmark$ | | 0.6145 ($\pm$ 0.024) | 0.6643 ($\pm$ 0.055) | 0.7973 ($\pm$ 0.009) | 0.6368 ($\pm$ 0.034) | 0.7478 ($\pm$ 0.060) | 0.6921 | 26.53 | 5 |
> > > | | $\checkmark$ | 0.6138 ($\pm$ 0.022) | 0.6293 ($\pm$ 0.067) | 0.7975 ($\pm$ 0.013) | 0.6361 ($\pm$ 0.036) | 0.7592 ($\pm$ 0.036) | 0.6872 | 25.23 | 5 |
> > > | $\checkmark$ | $\checkmark$ | **0.6176** ($\pm$ 0.025) | **0.6652** ($\pm$ 0.057) | **0.8002** ($\pm$ 0.010) | **0.6370** ($\pm$ 0.027) | 0.7571 ($\pm$ 0.045) | **0.6954** | **25.15** | 5 |
> > >
> > > **(2) Clarifications on the two possible reasons stated in our AW4**
> > >
> > > We stated two possible reasons regarding the marginal improvements in Table 3.
> > >
> > > Here, we clarify them one by one with the help of the above combined table, as follows:
> > >
> > > **(a) The use of prognostic priors ($P$) could lead to performance saturation**
> > >
> > > This could be observed from
> > > - the comparison between the average CI improvements brought by the ordinal biases used in two baselines: **1.43%** (w/o $P$ + w/ ordinal biases) VS. **0.91%** (w/ $P$ + w/ ordinal biases);
> > > - the comparison between the average CI results of two baselines: **0.6482** (w/o $P$ + w/o ordinal biases) VS. **0.6843** (w/ $P$ + w/o ordinal biases).
> > >
> > > Concretely,
> > > - The **first comparison** shows that the CI improvements are often relatively smaller when $P$ is used while these improvements are larger when $P$ is not used. Therefore, one possible reason firstly hitting our mind is the use of $P$.
> > > - Furthermore, the **second comparison** shows that, when $P$ is used, the baseline without any ordinal biases has obtained a relatively-high start point, an average CI of 0.6843. This performance has already surpassed all compared baselines in Table 1 “Main results”.
> > > - Accordingly, we i) consider it as a signal or trend of performance saturation in VLSA and then ii) state it as one possible reason that could explain the marginal improvements in Table 3, to try to address your concerns raised in W4.
> > >
> > > Although we found that the use of $P$ could alleviate the ordinal inductive bias and lead to performance saturation, we do observe that the proposed two ordinal bias terms are often helpful to VLSA.
> > >
> > > Continue Down

---

> > > ### Author Response · Authors · 2024-12-02
> > > **Reply to Further Comment by Reviewer f7nn**
> > >
> > > Continue from Above
> > >
> > > Specifically, we observe that, when the proposed two ordinal bias terms are **NOT** presented in VLSA,
> > > - there is **a decrease of 1.61** in average MAE in Table 3, suggesting a worse guess of time-to-event for patients;
> > > - there is **a decrease of 0.91%** in average CI in Table 3, although this could be marginal relatively;
> > > - there is **a large degeneration** in the quality of generated survival prompts, with a decrease of 38.79% in average Acc (on the given three datasets), as shown in Figure 3(a).
> > >
> > > The above degeneration suggest the benefits of using the two ordinal bias terms.
> > >
> > > Although the two ordinal bias terms often show marginal improvements in terms of CI, they could often contribute to VLSA in **other more terms** like MAE and the quality of survival prompts.
> > >
> > > **(b) One of ordinal inductive biases, ordinal incidence function, often helps to improve MAE, not CI**
> > >
> > > This could be observed from
> > > - the average MAE improvements brought by the ordinal incidence function in the VLSA w/o $P$: **1.71** (over w/o ordinal biases) and **1.23** (over w/ only ordinal prompts);
> > > - the average CI improvements brought by the ordinal incidence function in the VLSA w/o $P$: **0.14%** (over w/o ordinal biases) and **0.21%** (over w/ only ordinal prompts);
> > > - the average MAE improvements brought by the ordinal incidence function in the VLSA w/ $P$: **1.53** (over w/o ordinal biases) and **1.38** (over w/ only ordinal prompts);
> > > - the average CI improvements brought by the ordinal incidence function in the VLSA w/ $P$: **0.09%** (over w/o ordinal biases) and **0.33%** (over w/ only ordinal prompts);
> > >
> > > Concretely, the above results show that the improvements in MAE are **often positive and consistent** over different settings whereas those in CI are often marginal (0.09% ~ 0.33%).
> > >
> > > These empirical results drive us to think the other possible reason behind the marginal improvement in Table 3, *i.e.*, the ordinal incidence function often helps the improvement of MAE, not CI.
> > >
> > > Yes, as you mentioned, C-index exactly focuses on the accuracy regarding the ordinality of the predicted pairs, and this should mean that C-index is more sensitive to ordinal inductive biases than MAE.
> > >
> > > However, our proposed ordinal inductive biases, both the ordinal prompts and the ordinal incidence function, are imposed on **single individuals**, not pairs. Hence, their effectiveness could not be directly reflected by the C-index metric that is more concerned with predicted pairs. By contrast, the MAE metric, which is measured within predicted single individuals, could often directly reflect the effectiveness of the ordinal inductive biases, since we impose ordinal inductive biases on single individuals.

---

> > > ### Author Response · Authors · 2024-12-02
> > > **Reply to Further Comment by Reviewer f7nn**
> > >
> > > **For W5**: Thanks for your explanations. I agree that the representations encoded by CONCH are projected into the VL space. However, ABMIL is a parametric method that needs to be trained from scratch. I don’t think such a few samples in the few-shot setting can ensure the convergence of ABMIL. A better option is using non-parametric few-shot learning like the few-shot weakly-supervised classification method used in CONCH [2], or MI-FewShot used in mSTAR [3].
> > >
> > > ---
> > >
> > > **Response**: Thanks for your comments and thoughtful considerations.
> > >
> > > We have carefully checked the two papers: CONCH (Lu et al., Nature Medicine, 2024) and mSTAR (Xu et al, arXiv, 2024). We mainly compare with the non-parametric few-shot learning method used in mSTAR, namely, the MI-FewShot illustrated in Fig. 6a on page 13, because it is not only i) a latest non-parametric few-shot learning method for MIL settings but also ii) an improved version of the method used in CONCH.
> > >
> > > Next, we first (1) show the additional experiments for MI-FewShot and then (2) provide our revisions.
> > >
> > > **(1) Additional experiments for MI-FewShot**
> > >
> > > **Experimental settings**: We follow the official implementation of MI-FewShot ([here](https://github.com/Innse/mSTAR/blob/main/downstream_task/fewshot_classification/main.py)) in this experiment. Different from the original MI-FewShot for WSI classification (Fig. 6a on page 13 in the mSTAR paper), the class prompts are replaced with the naïve prompts used in MI-Zero, as you mentioned before.
> > >
> > > **Results**: The experimental results of MI-FewShot are as follows:
> > >
> > > **Table 4**: Few-shot survival prediction (data efficiency evaluation)
> > > | Method | # Shots = 1 | # Shots = 2 | # Shots = 4 | # Shots = 8 | # Shots = 16 | Average | Full-shots |
> > > |:--:|:--:|:--:|:--:|:--:|:--:|:--:|:--:|
> > > | MI-FewShot | 0.5478 | 0.5439 | 0.5499 | 0.5539 | 0.5873 | 0.5567 | 0.5692 |
> > > | VLSA | **0.5787** | **0.6068** | **0.6271** | **0.6465** | **0.6592** | **0.6237** | **0.6954** |
> > >
> > > These results suggest the superiority of VLSA in terms of data efficiency over the non-parametric few-shot learning method.
> > >
> > > **(2) Our revisions**
> > >
> > > - At the top of page 10, above Table 4: We will add a sentence to describe the additional baseline MI-FewShot, as follows:
> > > > MI-FewShot (Xu et al., 2024) is a non-parametric method designed for FSL. We adopt it as an additional baseline for comparisons.
> > >
> > > - Table 4 on page 10: We will add the additional experimental results of MI-FewShot shown above to it.
> > >
> > >
> > > [Reference D] Lu et al. A visual-language foundation model for computational pathology. Nature Medicine, 2024.
> > >
> > > [Reference E] Xu et al. A Multimodal Knowledge-enhanced Whole-slide Pathology Foundation Model. arXiv 2024.

---

> > > > ### Comment · Reviewer_f7nn · 2024-12-03
> > > >
> > > > Thanks for the detailed responses, which basically addressed most of my concerns. However, I still disagree that the reason for the marginal increase of two ordinal items is performance saturation since the presented prognostic priors have no relation to ordinal bias, and 0.6954 of C-Index is far from saturating. Despite the marginal increase of two ordinal regularization items, I would like to adjust my score because the author offers a solution to leverage LLMs for survival prediction, which empirically surpasses the classical SA models.

---

> ### Author Response · Authors · 2024-11-26
> **Reply to Reviewer's Comments**
>
> **W1**: The motivation of ordinal IF is confusing. The authors assume there is a consecutive decline in probability for those classes away from the time when the event occurs, which raises doubts about its reasonableness when events occur multiple times. This assumption is only concerned with the event that first occurs.
>
> ---
>
> **AW1**: Thanks for your insightful feedback.
>
> Yes, our assumption is concerned with the event that first occurs. In fact, the predictive probability that we define as $\hat{y}$ is a probability distribution on **the first hitting time**, as written at lines 241-243 on page 5.
>
> Moreover, we focus on the **overall survival (OS)** of patients in this work. The OS, which only occurs once, is a typical event of interest in survival analysis (Kaplan & Meier, 1958). It is often set as the outcome for the survival analysis of Whole-Slide Images (WSIs) in the field of CPATH.
>
> To avoid possible confusion, we have made revisions as follows:
>
> - Lines 258 and 259 on page 5 of our revised paper: we have added a sentence to further explain our assumption, as follows:
> > Note that this assumption is concerned with the event that first occurs, as $\hat{y}$ is defined as the probability distribution on the first hitting time.
>
> - Lines 320 and 321 on page 6 of our revised paper: we have added a sentence to point out the event of interest this work focuses on, as follows:
> > We follow Chen et al. (2021a) to set overall survival (OS, which only occurs once) as the event of interest in WSI survival analysis.
>
> [Reference A] Kaplan & Meier. Nonparametric estimation from incomplete observations. Journal of the American Statistical Association, 1958.

---

> ### Author Response · Authors · 2024-11-26
> **Reply to Reviewer's Comments**
>
> **W2**: Ordinal Survival Prompt seems to ignore the status of event indicators (i.e. censorship) when designing the prompts. For different censorships, there should be different outcomes reflected in the survival prompts even when their follow-up time are the same. When the case is censored, the only thing we can know is the event didn’t occur before the follow-up time. For example, we cannot simply assign a case with a short follow-up time into a bin of “very poor prognosis”.
>
> ---
>
> **AW2**: Thanks for your thoughtful consideration. We agree with your comments on censorship and the way of handling it.
>
> In fact, our way of handling censorship is the same as that you mentioned. Next, we would like to first (1) explain our way of handling censorship, then (2) clarify that censorship is not handled via designing the prompts in our VLSA, and finally (3) give our revision to make these clear enough.
>
> **(1) Our way of handling censorship**
>
> As written in Eq. (9) on page 6, our objective function to optimize the prediction of incidence function is as follows:
>
> $$\\mathcal{L_{\\text{MLE}}}=-\\left[\\delta\\cdot\\log(\\hat{y_c})+(1-\\delta)\\cdot\\log(1 - \\sum_{i=1}^{c-1} \\hat{y_{i}})\\right].$$
>
> It is a standard maximum likelihood estimation (MLE)-based objective (Tutz & Schmid, 2016) in survival analysis. When a case is censored ($\delta = 0$), this objective handles censorship by maximizing $\\log(1 - \\sum_{i=1}^{c-1} \\hat{y_{i}})$, namely, minimizing the probability that the event first occurs at bins 1, 2, …, $c-1$, *i.e.*, $\sum_{i=1}^{c-1} \hat{y}_{i}$.
>
> **(2) Censorship is not handled via designing the prompts in our VLSA**
>
> $\hat{y_{i}}$ is derived from the cosine similarity between the image feature $f_\text{image}$ and the $i$-th text prompt $f_\text{text}^i$. Since $\hat{y}_i$ is defined as the probability that an event first occurs at the $i$-th time bin (as written at line 241 on page 5), the $i$-th text prompt should indicate the risk of an event first hitting at bin $i$. For example, a text prompt indicates a very high risk of event occurrence by using “very poor prognosis”, or indicates a very low risk by using “very good prognosis”.
>
> In summary, censorship is not handled via designing the prompts in our VLSA, but via the MLE-based training objective.
>
> **(3) Our revision**
>
> To clarify our way of handling censorship, we have added a sentence at lines 285 and 286 on page 6 of our revised paper, as follows:
> > For censored patients ($\delta=0$), $\mathcal{L_{\text{MLE}}}$ minimizes $\sum_{i=1}^{c-1} \hat{y_{i}}$, *i.e.*, the probability that the event of interest first occurs at discrete bins 1, 2, $\dots$, or $c-1$ according to the definition of $\hat{y_{i}}$ in Eq. (6).
>
> [Reference B] Tutz & Schmid. Modeling discrete time-to-event data. Springer, 2016.

---

> ### Author Response · Authors · 2024-11-26
> **Reply to Reviewer's Comments**
>
> **W3a**: To validate the effectiveness of prompts tailored to survival tasks, a stronger baseline equipped with naive prompts that simply formulate survival tasks as classification tasks, should be compared.
>
> ---
>
> **AW3a**: Thanks for your comments. To address your concerns, we would like to clarify that
> 1. Formulating survival tasks as classification ones may not be directly implemented by designing naïve prompts, from our humble understanding (refer to the detailed explanations below);
> 2. Forcing survival tasks to be classification ones would result in unfair comparisons, which thus could not be used to validate the effectiveness of survival prompts (refer to the detailed explanations below);
> 3. In fact, our approach to learning survival prompts is **a basic approach** in the context of VLM-driven survival analysis. It is a simple means to implement survival analysis within VLMs. Replacing it with other prompts used in classification would lead to the model NOT strictly for survival analysis, as shown by the above two points.
>
> Our detailed explanations for 1 and 2 are as follows:
>
> **(1)** From our humble understanding, naïve prompts could not formulate survival tasks as classification ones. It is because survival tasks involve the analysis of censored patients (*i.e.*, censorship). If we simply encode censorship information using naïve prompts, the prompts would be **meaningless for classification** since censorship is a factor independent with WSI features and it thus cannot be directly predicted from WSI features.
>
> We would like to note that the above is from our humble understanding. Please let us know if there is a proper way to design such naïve prompts; we are more than happy to conduct experiments in that way to validate our approach so as to make this work sounder and further improve this paper.
>
> **(2)** To our knowledge, there is a preprocessing approach that formulates survival tasks as classification ones. The commonly-used settings are as follows:
> - defining an endpoint of survival time ($T_\text{end}$),
> - using $T_\text{end}$ to convert survival labels into binary classification labels, *i.e.*, having experienced event vs. No event occurrence at $T_\text{end}$.
>
> However, for the patients censored at $t$ and $t< T_\text{end}$, their binary classification labels are unknown and cannot be determined accordingly. This means that these censored patients must be excluded from the training data, resulting in the training data **without** some censored patients in classification. This baseline could not lead to a head-to-head comparison, thus remaining apparent concerns about whether the effectiveness of survival prompts is validated in a right way.

---

> ### Author Response · Authors · 2024-11-26
> **Reply to Reviewer's Comments**
>
> **W3b**: To support the authors’ claim that "highly-expressive modern networks may cause overfitting when facing present small WSI data”, the current “highly-expressive” models should be compared with the proposed method. Furthermore, the extra text description generated by GPT-4o should be an additional modality. Therefore, some representative multimodal survival models should be compared, which can validate the efficacy of utilizing the pathology foundation model. I understand there have been a few models that integrated WSI and text data in the past. However, by replacing the gene data with text data, some multimodal survival models using WSI and gene data can be adapted for comparison, e.g. MCAT [1], MOTCat [2], MMP [3], etc.
>
> ---
>
> **AW3b**: Thanks for your detailed comments and suggestions. We appreciate your efforts in this review.
>
> Our point-to-point response is as follows:
>
> **(1) Comparing with the current “highly-expressive” models**
>
> In fact, we have compared current “highly-expressive” models, such as the Transformer-based TransMIL, GNN-based Patch-GCN, and the latest ILRA. These models are based on well-known highly-expressive neural network models, like GNNs or Transformers. Moreover, our choice of these representative models follows the latest settings used by Song et al. (CVPR, 2024).
>
> **(2) Comparing with some representative multimodal survival models**
>
> We would like to clarify that our text description used to guide aggregation is **NOT** an additional modality from the perspective of multi-model representation learning.
>
> Our detailed clarification is as follows:
> - Our text description is NOT utilized as a conventional text modality that is generally fused with the other modality to produce multi-modal representations for better prediction. In fact, our text description is only utilized to provide the weight of each instance for instance aggregation, just like the role of attention weights in aggregation, as written in Eq. (2) on page 4. This means that, our text description has **not been fused** with WSI features to obtain multi-model representations. Our final representation in the vision-end **always comes from WSI features only**, NOT a multi-modal representation.
> - Current multimodal survival models using WSI and gene data, such as MCAT, MOTCat, and MMP, are standard multi-model representation learning frameworks, in which two modalities (WSI and gene data) are **explicitly fused** to produce multi-modal representations for better prediction.
>
> From the above discussion, we could find that there is **a fundamental discrepancy** between our approach and existing multi-modal survival models. In view of this, we choose to pay more attention on the comparison with existing VLM-driven or vision-only survival analysis models.
>
> **(3) Our revision**
>
> To avoid possible misunderstanding, we have added a description to discuss existing multi-modal survival models and analyze the difference between them and our VLSA. It can be seen at lines 185-190 on page 4 of our revised paper, as follows:
> > (3) Difference from multi-modal representation learning: note that our text description of prognostic prior is only utilized to provide the weight of each instance for aggregation and mainly plays a guidance role. The final representation in vision-end only comes from WSI features, not a multi-modal representation fusing text and vision features. This is a fundamental discrepancy between our method and current multi-modal survival models, such as MCAT (Chen et al., 2021b), MOTCat (Xu & Chen, 2023), CMTA (Zhou & Chen, 2023), MoME (Xiong et al., 2024), and MMP (Song et al., 2024b).
>
> [Reference C] Song et al. Morphological prototyping for unsupervised slide representation learning in computational pathology. CVPR 2024.

---

> ### Author Response · Authors · 2024-11-26
> **Reply to Reviewer's Comments**
>
> **W3c**: Unimodal MIL models are a bit out-of-date. More “highly-expressive” SOTA methods should be compared, such as CAMIL [4], R2T [5], etc.
>
> ---
>
> **AW3c**: Thanks for your suggestion.
>
> To address your concerns, we have carefully read these two papers and checked the implementation details of them. We find that
> - the official implementation of R2T is compatible with patient-level survival analysis tasks;
> - however, CAMIL could not be directly adapted to patient-level survival analysis, because it involves the slide-level graph construction and is designed for slide-level cancer subtyping tasks while survival analysis is a **patient-level** task and involves **all available slides of one patient** (possibly five slides). Moreover, since there could more than 130000 patches for one patient (as shown in Table 6), there are the concerns about the overflow of GPU memory in graph-based learning.
>
> **(1) Additional experimental results**
>
> We provide the additional experimental results of R2T as follows:
>
> (a) **Survival prediction**
>
> The additional experimental results for R2T are as follows:
>
> **Table 1**: Main comparative results
> | Method | TCGA-BLCA | TCGA-BRCA | TCGA-GBMLGG | TCGA-LUAD | TCGA-UCEC | Average CI | Average MAE | D-cal Count|
> |:--:|:--:|:--:|:--:|:--:|:--:|:--:|:--:|:--:|
> | R$^2$T-MIL | 0.5775 ($\pm$ 0.024) | 0.5473 ($\pm$ 0.095) | 0.7757 ($\pm$ 0.024) | 0.5711 ($\pm$ 0.076) | 0.6510 ($\pm$ 0.087) | 0.6245 | 32.54 | 4 |
> | VLSA | **0.6176** ($\pm$ 0.025) | **0.6652** ($\pm$ 0.057) | **0.8002** ($\pm$ 0.010) | **0.6370** ($\pm$ 0.027) | **0.7571** ($\pm$ 0.045) | **0.6954** | **25.15** | **5** |
>
> From this table, we can see that our VLSA often outperforms R2T in terms of CI and MAE.
>
> (b) **Few-shot learning scenarios**
>
> Moreover, we have also conducted additional experiments for R2T in few-shot learning (FSL) scenarios. Their results are as follows:
>
> **Table 4**: Few-shot survival prediction (data efficiency evaluation)
> | Method | # Shots = 1 | # Shots = 2 | # Shots = 4 | # Shots = 8 | # Shots = 16 | Average | Full-shots |
> |:--:|:--:|:--:|:--:|:--:|:--:|:--:|:--:|
> | R$^2$T-MIL | 0.5725 | 0.5835 | 0.5924 | 0.5927 | 0.6072 | 0.5897 | 0.6245 |
> | VLSA | **0.5787** | **0.6068** | **0.6271** | **0.6465** | **0.6592** | **0.6237** | **0.6954** |
>
> These results could also verify the effectiveness of our method.
>
> (c) **Model efficiency evaluation**
>
> The model efficiency evaluation results of R2T are given as follows:
>
> **Table 5**: Model efficiency
> | Model | # Params | # MACs |
> |:--:|:--:|:--:|
> | R$^2$T-MIL | 2.44M | 29.62G |
> | VLSA | **284.16K** | **0.25G** |
>
> This result indicates that our model is more efficient than R2T in term of # Params and # MACs.
>
> **(2) Our revision**
>
> (a) **Discussion on the state-of-the-art methods (like CAMIL and R2T) for WSI classification**
>
> We have added a discussion for the state-of-the-art methods (*e.g.*, CAMIL and R2T) for WSI classification, as follows:
> > (4) Many state-of-the-art approaches for WSI classification (Zhang et al., 2022; Xiong et al., 2023; Tang et al., 2023; Qu et al., 2023; Shi et al., 2024; Fourkioti et al., 2024; Tang et al., 2024) have shown strong capabilities in representation learning. VLSA could take inspiration from them to further improve its performance in SA.
>
> This revision can be seen at lines 550-553 on page 11 of our revised paper.
>
> (b) **Additional experimental results for R2T**
>
> We have listed R2T as the additional baseline for comparisons, so their relevant parts have been updated, as mentioned above. Concretely, our revision details are as follows:
> - Lines 326 on page 7 of our revised paper: We have updated the description of baselines.
> - Table 1 on page 7 of our revised paper: The main results of the additional baseline have been added to Table 1.
> - Table 4 on page 10 of our revised paper: The FSL results of the additional baseline have been added to Table 4.
> - Table 5 on page 10 of our revised paper: The model efficiency results of the additional baseline have been added to Table 5.

---

> ### Author Response · Authors · 2024-11-26
> **Reply to Reviewer's Comments**
>
> **W4**: From the results of Tables 3 and 8, performance gains from two ordinal prompts are marginal or even getting worse. The reason should be justified.
>
> ---
>
> **AW4**: Thanks for your careful reading.
>
> To address your concerns, we would like to first (1) present the performance gains from two ordinal inductive biases in Table 3 and 8, then (2) explain the reason, and finally (3) show our revision that discusses the reason.
>
> Our detailed response is as follows:
>
> **(1) Performance gains from two ordinal inductive biases in Table 3 and 8**
>
> In **Table 8** which shows the effects of two ordinal inductive biases on the VLSA **without prognostic priors**, the CI performance gains from two ordinal inductive biases are **1.12%**, **1.86%**, **0.55%**, **2.98%**, and **0.68%** on BLCA, BRCA, GBMLGG, LUAD, and UCEC, respectively. Moreover, the improvement in average CI is **1.43%**, larger than 1%.
>
> To our knowledge, most of these improvements could be **comparable** with those improvements given by representative survival analysis works. For example, Patch-GCN (Chen et al., MICCAI, 2021) shows an improvement of **1.6%** in average CI, as written in Table 1 on page 8 of PatchGCN’s original paper; MCAT (Chen et al., ICCV, 2021) shows an improvement of **1.9%** in average CI, as written in Table 1 on page 7 of MCAT’s original paper.
>
> In **Table 3** which shows the effects of two ordinal inductive biases on **our complete VLSA**, the CI performance gains from two ordinal inductive biases are 0.48%, 3.48%, 0.75%, 0.19%, and -0.35% on BLCA, BRCA, GBMLGG, LUAD, and UCEC, respectively. However, most of these improvements are marginal, smaller than those improvements shown in Table 8.
>
> **(2) The reason why Table 3 shows marginal CI improvements**
>
> There are two main possible reasons:
> - When prognostic prior is used in VLSA, there could be limited room for CI performance improvements by using ordinal inductive biases. This could be indicated by this empirical fact: when there is no ordinal inductive bias, the average CI obtained by the VLSA **without** prognostic prior is **0.6482** (Table 8) while that obtained by the VLSA **with** prognostic prior is **0.6863** (Table 3). This means that getting further large CI improvements over the VLSA with prognostic prior could be more difficult, since the VLSA with prognostic prior has already obtained a relatively-high CI (0.6863) and it shows **a trend of performance saturation**.
> - One of ordinal inductive biases, *i.e.*, ordinal incidence function, often helps to **improve MAE**, not CI, as written at line 419 on page 8 and line 1186 on page 22.
>
> **(3) Our revision**
>
> We have added a paragraph, the second paragraph in Section 4.4 on page 9 of our revised paper, to discuss the reason why Table 3 shows marginal CI improvements, as follows:
> > Discussion We note that in this study two ordinal inductive biases often show marginal improvements in CI. One possible reason for this is that VLSA shows a trend of saturation in CI performance when it uses prognostic priors. It could be observed from that the average CI has already reached a high level (0.6863) when there is no any ordinal inductive bias. By contrast, for the counterpart, *i.e.*, the VLSA without prognostic priors (see Table 8), its average CI is only 0.6482 when there is no any ordinal inductive bias. This means there could leave room for CI improvements. This is indicated by the experimental results of Table 8: i) two ordinal inductive biases often show larger improvements on five datasets and ii) the average improvement in CI is 1.43%.
>
> [Reference D] Chen et al. Whole slide images are 2d point clouds: Context-aware survival prediction using patch-based graph convolutional networks. MICCAI 2021.
>
> [Reference E] Chen et al. Multimodal Co-Attention Transformer for Survival Prediction in Gigapixel Whole Slide Images. ICCV 2021.

---

> ### Author Response · Authors · 2024-11-26
> **Reply to Reviewer's Comments**
>
> **W5**: Comparisons in Few-shot learning may not be fair. The capability of few-shot prediction greatly relies on how the visual and text spaces are well-aligned. CONCH is a pathology VL model that aligns visual and text spaces well. However, CoOp and OrdinalCLIP are equipped with ABMIL, which is not aligned with the text space, for few-shot prediction by using such few samples. For a fair comparison, the author should replace prompts with the ones in CoOp and OrdinalCLIP only based on CONCH to validate the efficacy of survival prompts.
>
> ---
>
> **AW5**: Thanks for your detailed feedback. We agree with your comment on the key to the capability of few-shot prediction.
>
> To address your concerns, we would like to clarify that CoOp and OrdinalCLIP are equipped with **the ABMIL whose visual outputs are aligned with the text space**.
>
> Our detailed explanation is as follows:
>
> Since CoOp and OrdinalCLIP are originally proposed for natural images, not gigapixel WSIs, we adapt them to WSI survival analysis, as follows:
> - **Vision-end**: the input is a group of instances. Each instance is a feature vector extracted by the CONCH’s vision encoder; and this vision encoder has projected the feature vector into **the VL embedding space** (written at lines 320 and 321 on page 6). To aggregate a group of instances into a vector-like representation for prediction, we adopt the classical aggregator ABMIL. Concretely, ABMIL is able to learn an adaptive weight for each instance via the attention mechanism. The learned weights (one weight for one instance) are then used for aggregating all instances through weighted averaging. This process yields a vector-like representation for prediction. This representation is **still in the VL embedding space**, because its inputs, *i.e.*, all instances, come from the VL embedding space as aforementioned.
> - **Language-end**: same as VLSA, CoOp and OrdinalCLIP **also use the CONCH’s text encoder** to obtain the textual features of survival prompts, as written at lines 318 and 319 on page 6.
>
> The above implementation is similar to the conventional way of calculating VL-aligned features (Zhang *et al.*, IEEE TPAMI, 2024):
> - extracting image features using the VLM’s vision encoder,
> - projecting the image features into the VL embedding space,
> - and finally, obtain the textual features of prompts by passing the prompts through the VLM’s text encoder.
>
> The only difference is that instance features are aggregated into a vector-like representation; yet the aggregation is required in MIL.
>
> In summary, although CoOp and OrdinalCLIP are equipped with ABMIL, ABMIL is only used to aggregate instances and **its output is still in the VL embedding space**. This means that the output of CoOp and OrdinalCLIP in the vision-end are aligned with the text space.
>
> [Reference F] Zhang et al. Vision-language models for vision tasks: A survey. IEEE TPAMI, 2024.

---

> ### Author Response · Authors · 2024-11-26
> **Reply to Reviewer's Comments**
>
> **W6**: The code is unavailable which may affect the reproduction.
>
> ---
>
> **AW6**: We had already uploaded our source code as a supplementary file in our initial submission.
>
> To make it clearer, we have added a placeholder (`Our code is available at [this URL]`) in the Abstract to present the URL of our source code. It will be filled with a valid Github link once the anonymous policy is not active.

---

> ### Author Response · Authors · 2024-11-26
> **Reply to Reviewer's Comments**
>
> **Q1**: I‘m not sure how the authors applied MI-Zero to survival analysis, which was supposed to be designed for classification.
>
> ---
>
> **AQ1**: Yes, the original MI-Zero is a zero-shot approach for WSI classification. It is not a learning-based approach.
>
> We adapt MI-Zero to survival analysis, as follows:
> - **Vision-end**: A group of instances are aggregated into a vector-like representation via the aggregation methods proposed by MI-Zero.
> - **Language-end**: Four survival prompts with the survival context = “the cancer prognosis reflected in the pathology image is” and the survival classes = [“very poor”, “poor”, “good”, “very good”] are encoded into four textual features by the CONCH’s text encoder. Note that we tried different survival prompts (refer to our code) and selected the best one to report the results, as written at lines 994-998 on page 19.
> - **Prediction**: Same as our VLSA, we use the vector-like representation and the four textual features to derive the prediction of incidence function using Eq. (6) on page 5.
>
> To make the MI-Zero’s implementation clearer, we have incorporated the above details into our revised paper. This revision can be found at lines 1102-1108 on page 21 of our revised paper.

---

> ### Author Response · Authors · 2024-11-26
> **Reply to Reviewer's Comments**
>
> **Q2**: What’s the difference in prompts between OrdinalCLIP and Ordinal Survival Prompt?
>
> ---
>
> **AQ2**: No difference. Namely, OrdinalCLIP and our VLSA **share** the approach to calculating Ordinal Survival Prompt.
>
> To make it clearer, we have added a sentence, as follows:
> > As a result, OrdinalCLIP and our VLSA share the approach to calculating ordinal survival prompts in language-end.
>
> This revision can be seen at lines 333 and 334 on page 7 of our revised paper.

---

> ### Author Response · Authors · 2024-11-26
> **Reply to Reviewer's Comments**
>
> **Q3**: In equations 2 and 3, how can the model ensure learning multi-level representations? It seems there is no constraint to achieving this goal.
>
> ---
>
> **AQ3**: Equations 2 and 3 are used to show **the specific steps** of calculating multi-level representations, not the constraint to achieve multi-level WSI representation learning.
>
> Multi-level WSI representations are **ensured by prognostic priors**. It is because the textual descriptions of prognostic priors are generated by GPT-4o and they contain different and diverse prognostic biomarkers, as shown in Table 7 “Description of prognostic priors”.

---

> ### Author Response · Authors · 2024-11-26
> **Reply to Reviewer's Comments**
>
> **Q4**: CONCH is a large foundation model pretrained on a great amount of data including public data. In this case, to avoid data contamination, how can the author ensure test samples never occur in the pretraining materials of CONCH in the few-shot setting?
>
> ---
>
> **AQ4**: All five datasets used in this study are from TCGA. The pretraining of CONCH **excludes** the data from TCGA, as described in the Section Methods “Dataset curation” presented at the top of page 13 in CONCH’s original paper.
>
> Thus, there is no data contamination in the few-shot setting.

---

> ### Author Response · Authors · 2024-11-26
> **Reply to Reviewer's Comments**
>
> **Q5**: I’m confused about what the specific events are referring in these experiments. If overall survival, there should be only one event. However, the authors mentioned that the events can occur multiple times. Is it longitudinal data?
>
> ---
>
> **AQ5**: We are sorry for your confusion. The event of interest in this paper is **overall survival (OS)**. Yes, it should occur only once.
>
> After checking the writing throughout this paper, we fail to find the sentence that explicitly describes multiple event occurrences in this paper. Still, we would be more than happy to fix the wrong statements if further details could be provided.
>
> We appreciate your selfless efforts that make this paper better.

---

> ### Author Response · Authors · 2024-11-26
> **Reply to Reviewer's Comments**
>
> **Q6**: The motivation for using EMD should be elaborated. For example, why is EMD aware of ordinality?
>
> ---
>
> **AQ6**: Thanks for your kinder reminder. Due to limited space, we elaborate on it in Appendix B.3.
>
> Now, we have added a concise statement in our main paper to explain the EMD’s awareness of ordinality, as follows:
> > EMD is a measure aware of distribution ordinality as it considers the geometry property of distribution in distance measurement and it is smaller when the geometry (shape) of two distributions is closer. Please refer to Appendix B.3 for detailed explanations.
>
> This revision can be seen at lines 265-268 on page 5 of our revised paper.

---

> ### Author Response · Authors · 2024-11-26
> **Reply to Reviewer's Comments**
>
> **Q7**: It is difficult to understand Figure 3(a) is trying to explain. What do the horizontal and vertical axes represent, respectively?
>
> ---
>
> **AQ7**: We are sorry for the inconvenience and appreciate your patience in kindly reminding us of these omissions.
>
> Our detailed response is as follows:
>
> **(1) Clarification on Figure 3(a)**
>
> Figure 3(a) tries to **further examine the effect of ordinal prompt**, *i.e.*, the ordinal inductive bias imposed on survival prompts, in a **qualitative** way. We obtain this figure by
> - calculating the similarity between any two survival prompts learned by the model,
> - and showing the similarity in a heatmap.
>
> Thus, by intuitively comparing the heatmaps from the VLSA model with and without ordinal prompt, we can qualitatively examine the effect of ordinal prompt.
>
> For each heatmap of Figure 3(a),
> - its **horizontal axis** places the first prompt to the last prompt from left to right;
> - its **vertical one** does so from top to bottom.
>
> **(2) Our revision**
>
> To remedy this, we have made revisions as follows:
>
> - Lines 444-447 on page 9 of our revised paper: We have added a detailed description below Figure 3(a), as follows:
> > (a) Heatmap of the similarity between any two learned survival prompts. Its horizontal axis places the first prompt to the last prompt from left to right; its vertical one does so from top to bottom.
>
> - Lines 479-482 on page 9 of our revised paper: We have added a few sentences to describe the origin and purpose of Figure 3(a), as follows:
> > We calculate the similarity between any two survival prompts learned by the model and show the similarity in a heatmap. By intuitively comparing the heatmaps from the VLSA model with and without ordinal prompt, we further examine the effect of ordinal prompt in a qualitative way.

---

> ### Comment · Reviewer_f7nn · 2024-11-26
> **For W3a**
>
> Thanks for your responses. There are some misunderstandings of naive prompts about formulating the task to classification, which doesn’t mean the neglect of censorship. Here I try to make it more clear.
>
> As you mentioned in the Q1 response about how to apply MI-Zero to survival, you simply divide the survival labels into several survival classes. Similarly, the prompts used in MI-Zero can be applied here as naive prompts.
>
> I agree that censorship cannot be directly predicted from WSI features. However, the morphological features of WSIs for censored patients should differ from those for uncensored patients even with the same survival/observation time. Therefore, they should be regarded as two separate classes. This has been validated in the previous work, PIBD [1], where PIB held the discriminative assumption that combinations of different 't' (time) and 'c' (censorship) are treated as distinct categories.
>
> [1] Prototypical Information Bottlenecking and Disentangling for Multimodal Cancer Survival Prediction, ICLR, 2024.

---

> > ### Comment · Reviewer_f7nn · 2024-11-26
> > **For W3b**
> >
> > I don't think comparing with multimodal methods is unnecessary for the following reasons.
> >
> > 1. The motivation for leveraging the generated texts is to alleviate the challenge of scarce training data, and the generated texts can provide extra auxiliary signals
> > to boost learning efficiency. If the performance of directly combining the generated text via existing multimodal methods is better, which would have achieved the goal mentioned in the motivation, why would we need the method proposed in the paper?
> > 2. Although the text modality is not explicitly fused into the final representation, the additional information provided by text has already been implicitly injected into the learning process, making it naturally superior to an unimodal approach. Therefore, it would be unfair to solely compare unimodal models. However, if explicitly integrating the textual modality leads to better performance, why not do so? This approach could also achieve the desired goal.
> >
> > Therefore, comparing with existing multimodal methods is essential to validate the necessity of the proposed method.

---

> > > ### Comment · Reviewer_f7nn · 2024-11-26
> > > **For W4**
> > >
> > > I disagree with the authors’ reason why Table 3 shows marginal CI improvements. The authors claimed that prognostic priors have alleviated ordinal inductive bias, leading to performance saturation. On one hand, if this statement holds true, then the proposed two bias terms are unnecessary. On the other hand, I disagree with this statement because the prognostic priors presented in Appendix D.4 have no relation to ordinal bias.
> > >
> > > Additionally, C-index exactly focuses on the accuracy regarding the ordinality of the predicted pairs, which should be more sensitive to ordinal inductive biases than MAE. Specifically, C-index is calculated by evaluating whether the predictive model accurately reflects these ordinal relationships. Therefore, I disagree with the author’s explanations that “One of ordinal inductive biases, *i.e.*, ordinal incidence function, often helps to improve MAE, not CI”.

---

> > > > ### Comment · Reviewer_f7nn · 2024-11-26
> > > > **For W5**
> > > >
> > > > Thanks for your explanations. I agree that the representations encoded by CONCH are projected into the VL space. However, ABMIL is a parametric method that needs to be trained from scratch. I don’t think such a few samples in the few-shot setting can ensure the convergence of ABMIL. A better option is using non-parametric few-shot learning like the few-shot weakly-supervised classification method used in CONCH [2], or MI-FewShot used in mSTAR [3].
> > > >
> > > > [2] A visual-language foundation model for computational pathology, 2024.
> > > >
> > > > [3] A Multimodal Knowledge-enhanced Whole-slide Pathology Foundation Model, 2024.

---

> ### Author Response · Authors · 2024-12-02
> **Reply to Further Comment by Reviewer f7nn**
>
> Thanks for your further feedback and detailed clarifications. Seriously, we deeply appreciate your great patience and responsibility in reviewing this paper at ICLR 2025. This paper could NOT be better if without your constructive suggestions and insightful comments.
>
> Next, we provide a detailed point-to-point response, supplemented with additional experiments, to address your additional questions and concerns.

---

> ### Author Response · Authors · 2024-12-02
> **Reply to Further Comment by Reviewer f7nn**
>
> **For W3b**: I don’t think comparing with multimodal methods is unnecessary for the following reasons. 1. The motivation for leveraging the generated texts is to alleviate the challenge of scarce training data, and the generated texts can provide extra auxiliary signals to boost learning efficiency. If the performance of directly combining the generated text via existing multimodal methods is better, which would have achieved the goal mentioned in the motivation, why would we need the method proposed in the paper? 2. Although the text modality is not explicitly fused into the final representation, the additional information provided by text has already been implicitly injected into the learning process, making it naturally superior to an unimodal approach. Therefore, it would be unfair to solely compare unimodal models. However, if explicitly integrating the textual modality leads to better performance, why not do so? This approach could also achieve the desired goal. Therefore, comparing with existing multimodal methods is essential to validate the necessity of the proposed method.
>
> ---
>
> **Response**: Thanks for your detailed explanations.
>
> Yes, as you mentioned, explicitly integrating the textual modality is indeed another approach to leveraging language priors, in additional to our proposed one. Thus, it is necessary to conduct experiments to further compare our method with multi-modal survival analysis (SA) methods. We deeply appreciate your insightful and high-quality reviews at ICLR 2025. We believe these would contribute a lot to this work and provide us valuable guidance to further improve this paper.
>
> Next, we would like to first (1) present the comparisons with multi-modal SA methods and then (2) provide our revision details.
>
> **(1) Comparisons with multi-modal survival analysis methods**
>
> **Experimental settings**: According to your suggestion mentioned in W3b, we have conducted additional experiments for MCAT (Chen et al., ICCV 2021), MOTCat (Xu & Chen, ICCV 2023), and MMP (Song et al., ICML 2024). Some notable settings are as follows:
> - Since these multi-modal SA methods take WSI and gene data as inputs, we replace their gene input with our language-encoded prognostic priors for comparisons, following your instructions. For the other settings, we strictly follow their official codes in this experiment: [MCAT](https://github.com/mahmoodlab/MCAT/blob/master/models/model_coattn.py), [MMP](https://github.com/mahmoodlab/MMP/blob/main/src/mil_models/model_multimodal.py), and [MOTCat](https://github.com/Innse/MOTCat/blob/main/models/model_motcat.py). Thanks for the author’s selfless contributions.
> - Two approaches, Transformer-based and optimal transportation feature fusion, are adopted in MMP, so we compare with MMP’s two variants, *i.e.*, MMP (Trans.) and MMP (OT).
>
> **Results**: The additional experimental results of the compared multi-modal SA methods are as follows:
>
> **Table 13**: Comparison with multi-modal survival analysis models. * These methods are originally proposed for learning multi-modal representations from WSIs and gene data. We replace their gene input with the language-encoded prognostic priors $E_{\text{text}}(\mathcal{T}_\text{prog})$ for comparisons.
> | Method | TCGA-BLCA | TCGA-BRCA | TCGA-GBMLGG | TCGA-LUAD | TCGA-UCEC | Average CI | Average MAE | D-cal Count|
> |:--:|:--:|:--:|:--:|:--:|:--:|:--:|:--:|:--:|
> | MCAT * | 0.5899 ($\pm$ 0.020) | 0.6322 ($\pm$ 0.035) | 0.7980 ($\pm$ 0.020) | 0.5768 ($\pm$ 0.043) | 0.6827 ($\pm$ 0.022) | 0.6559 | 31.44 | 3 |
> | MMP$_\text{Trans.}$ * | 0.5857 ($\pm$ 0.040) | 0.5944 ($\pm$ 0.016) | 0.7917 ($\pm$ 0.012) | 0.5654 ($\pm$ 0.055) | 0.6928 ($\pm$ 0.049) | 0.6460 | 31.08 | 3 |
> | MMP$_\text{OT}$ * | 0.5961 ($\pm$ 0.031) | 0.6315 ($\pm$ 0.041) | 0.7948 ($\pm$ 0.019) | 0.5870 ($\pm$ 0.071) | 0.6915 ($\pm$ 0.052) | 0.6602 | 31.35 | 4 |
> | MOTCat * | **0.6227** ($\pm$ 0.029) | 0.6301 ($\pm$ 0.047) | 0.7967 ($\pm$ 0.026) | 0.5617 ($\pm$ 0.070) | 0.7234 ($\pm$ 0.033) | 0.6669 | 29.73 | 3 |
> | VLSA | 0.6176 ($\pm$ 0.025) | **0.6652** ($\pm$ 0.057) | **0.8002** ($\pm$ 0.010) | **0.6370** ($\pm$ 0.027) | **0.7571** ($\pm$ 0.045) | **0.6954** | **25.15** | **5** |
>
> The above results show that MOTCat (Xu et al., ICCV 2023) is a strongest and competitive multi-modal SA baseline, obtaining an average CI of 0.6669 on five datasets.
>
> Moreover, compared with these multi-modal baselines specially designed for WSI and gene data, our method specially for vision-language data could often perform better in SA. This experiment could further confirm the effectiveness of VLSA.
>
> Continue Down

---

> ### Author Response · Authors · 2024-12-04
>
> **Response**: Clarification on the reason behind the marginal CI increase of two ordinal items
>
> For the reason behind the marginal CI increase of two ordinal items, we present two in AW4: **i)** the increase or effect could be reflected in other metrics like MAE (Table 3) and Ordinal Acc (Fig. 3a), not CI; and **ii)** performance saturation.
>
> **For the second reason**, although the presented prognostic priors have no direct or explicit relation to ordinal bias, we notice that these two are **closely related in final prediction**, after careful thinking, as follows:
> - the presented prognostic priors largely help to learn more discriminative visual features in **vision-end** (3.5% gain in Table 2),
> - ordinal prompts help to preserve the intrinsic inductive bias on ordinal survival classes in **language-end** (Fig. 3a),
> - survival prediction is made by **combining the two above**.
>
> Such close relation in the prediction-end seems to indicate that the effect of ordinal prompts on prediction may be pared down intuitively if the learned visual features are *sufficiently discriminative*, because
> - the ordinal prompts are **learnable** and
> - they can be adjusted in optimization to **align with visual features**.
>
> Therefore, to address your remaining concern, we would like to revise our old statement on the second reason and restate it as follows:
> > The other possible reason is that, the effect of the ordinal inductive bias imposed on survival prompts may be pared down, when the learned visual features are sufficiently discriminative. This is because our survival prompts are learnable and they can be adjusted in optimization to align with the discriminative features from vision-end.
>
> Thanks for your thoughtful comments.
>
> ---
>
> Dear Reviewer `f7nn`,
>
> We are so happy to hear that our replies have basically addressed most of your concerns. It is your high-quality feedback that continuously drives us to keep enhancing this paper. Thank you so much! Definitely, ICLR 2025 will be proud of having a group of professional and responsible reviewers like you.
>
> Additionally, we deeply appreciate your encouraging evaluation and your recognition of this work's contribution. These will certainly motivate us to do more cutting-edge works and pursuit more meaningful research. Thanks!
>
> Thank you for your consideration.
>
> The Authors of Submission 631

---

### Official Review · Reviewer_8wxk · 2024-11-03

**Soundness:** 3
**Presentation:** 3
**Contribution:** 2
**Rating:** 5
**Confidence:** 3

**Summary:**

This paper introduces a Vision-Language-based Survival Analysis (VLSA) paradigm to improve cancer prognosis prediction from histopathology whole-slide images by leveraging pathology foundation models for enhanced data efficiency and weakly-supervised learning. VLSA incorporates prognostic language priors and ordinal survival prompts to guide visual feature aggregation. The authors have conducted multiple experiments and ablation studies to evaluate the method proposed.

**Strengths:**

Overall, the paper is well-written and tries to address an important challenge in digital pathology. The proposed solution seems to also outperform the reported benchmark in the paper.

**Weaknesses:**

Major Comments:
1. The introduction to the paper highlights data scarcity for survival analysis as a major problem that this paper claims to solve. However, it seems like this paper addresses data efficiency in terms of model size and MACs, not the limited amount of training data available for survival analysis.
2. The number of bins being set at sqrt(N events) is given little justification. It also calls into question the comparison with other models (eg. PatchGCN used 4 bins for all datasets in the original paper). Additionally in the Appendix “Comparison with Hazard-Based Survival Modeling” 4 bins were used as opposed to the number used in the original experiments. The additional experiments in Appendix E don’t provide sufficient justification for using uniform sqrt(N events).
3. PatchGCN was trained on 4 bins in the original paper. The VLSA model should be compared with the baselines with the different bin sizes being considered for a fair comparison.
4. Given the variability of survival evaluation metrics (eg c-index) and the fairly marginal improvement over the baselines the authors should do a deeper analysis to showcase the improvements of their model. This could be done by plotting the survival curves of “high-risk” vs “low-risk” cases.
5. In Appendix E, there appear to be patches comprising of mostly white-space. Was any preprocessing done to filter out patches like this, or patches with significant blur or artifacts.

Minor Comments:
Spelling error at line 44 “While existing approaches have made exciting progress…”
In Appendix Table 6, there seems to be little point in labeling the maximum values.

**Questions:**

mentioned in weaknesses.

---

> ### Author Response · Authors · 2024-11-26
> **Reply to Reviewer's Comments**
>
> Thanks for your encouraging feedback and constructive comments.
>
> We provide a detailed point-to-point response to your questions and concerns.

---

> ### Author Response · Authors · 2024-11-26
> **Reply to Reviewer's Comments**
>
> **W1**: The introduction to the paper highlights data scarcity for survival analysis as a major problem that this paper claims to solve. However, it seems like this paper addresses data efficiency in terms of model size and MACs, not the limited amount of training data available for survival analysis.
>
> ---
>
> **AW1**: Thanks for your feedback. To resolve your concerns, we would like to clarify that our two core designs, including scheme design and experimental design, are presented in the paper to mitigate the challenge of data scarcity in survival analysis (SA).
>
> Our detailed response is as follows:
>
> **(1) Our two core designs to address data efficiency**
>
> - **Scheme design**: We improve the model’s data efficiency by **leveraging a pathology vision-language (VL) foundation model** and adapting it to WSI survival analysis, as written at lines 65-68 on page 2. Concretely, recent studies (Huang *et al.*, Nature Medicine, 2023; Lu *et al.*, Nature Medicine, 2024; Javed *et al.*, CVPR, 2024) have demonstrated that state-of-the-art pathology VL foundation models can obtain competitive performances with traditional models when they are fine-tuned with only a few training samples. Inspired by this finding, we leverage pathology vision-language (VL) foundation models to improve SA models’ data efficiency.
> - **Experimental design**: To validate whether our VLSA is of better data efficiency, we design **a few-shot learning (FSL) scenario** in which only a few samples are used for training, as presented in Table 4. FSL is usually adopted to test the data efficiency of models, as presented in recent studies (Huang *et al.*, Nature Medicine, 2023; Lu *et al.*, Nature Medicine, 2024). The results in Table 4 show that our VLSA could often obtain better performance in FSL, confirming the data efficiency advantage of our VLSA.
>
> **(2) Clarification on model efficiency and data efficiency**
>
> - For the **model size and MACs** presented in Table 5, we would like to clarify that they are reported to examine models’ efficiency **in terms of parameter numbers and computation consumption**, **NOT** the models’ data efficiency.
> - For the **limited amount of training data** available for SA, we state it as a challenge (written at line 43 on page 1). We try to mitigate this challenge by leveraging a pathology VL foundation model, because VL foundation models have shown clear advantages over traditional models in terms of **data efficiency**, as aforementioned.
>
> **(3) Our revision**
>
> To clearly state our approach for addressing data efficiency, we have made revisions as follows:
>
> - Line 67 on page 2 of our revised paper: We further highlight the data efficiency advantage of VL models by formatting “data efficiency” in bold.
>
> - Section 4.5 “Few-Shot Survival Prediction” on page 10 of our revised paper: We have added a few sentences to clearly describe the motivation behind designing FSL experiments, as follows:
> > Here we focus on a few-shot learning (FSL) scenario in which only a few samples can be used for training. We use this scenario to simulate the case of scarce training data and evaluate the performance of different SA methods in terms of data efficiency.
>
> - The caption of Table 4 on page 10 of our revised paper: We have improved it to show the aim of this FSL experiment more clearly, as follows:
> > Table 4: Few-shot survival prediction (data efficiency evaluation). The CI averaged on five datasets is presented.
>
> [Reference A] Huang et al. A visual–language foundation model for pathology image analysis using medical twitter. Nature Medicine, 2023.
>
> [Reference B] Lu et al. A visual-language foundation model for computational pathology. Nature Medicine, 2024.
>
> [Reference C] Javed et al. CPLIP: Zero-shot learning for histopathology with comprehensive vision-language alignment. CVPR 2024.

---

> ### Author Response · Authors · 2024-11-26
> **Reply to Reviewer's Comments**
>
> **W2**: The number of bins being set at sqrt(N events) is given little justification. It also calls into question the comparison with other models (eg. PatchGCN used 4 bins for all datasets in the original paper). Additionally in the Appendix “Comparison with Hazard-Based Survival Modeling” 4 bins were used as opposed to the number used in the original experiments. The additional experiments in Appendix E don’t provide sufficient justification for using uniform sqrt(N events).
>
> ---
>
> **AW2**: Thanks for your careful reading and constructive comments. We understand your concerns regarding the setting of time bins. To address your concerns, we (1) first clarify the conventional settings of time bins in SA, (2) then provide a point-to-point response to your comments, and (3) finally give our revision.
>
> **(1) The conventional settings of time bins in SA**
>
> - For **hazard**-based SA models, the conventional setting is the number of time bins being set to `4`. Since the inception of the representative work (Zadeh & Schmid, IEEE TPAMI, 2020), this conventional setting is commonly adopted (Chen *et al.*, MICCAI, 2021; Chen *et al.*, ICCV, 2021; Xu & Chen, ICCV, 2023; Zhou & Chen, ICCV, 2023; Zhang *et al.*, ICLR, 2024) in hazard-based SA on WSIs. PatchGCN (Chen *et al.*, MICCAI, 2021) is a hazard-based SA model.
> - For **incidence**-based SA models, the conventional setting is the number of time bins being set to $\sqrt{N_e}$. It is widely adopted in incidence-based survival modeling (Yu *et al.*, NIPS, 2011; Lee *et al.*, AAAI, 2018; Haider *et al.*, JMLR, 2020; Qi *et al.*, AAAI Symposium, 2023; Qi *et al.*, ICML, 2023). Our VLSA is an incidence-based SA model.
>
> **(2) Our point-to-point response**
>
> **W2.1**: The number of bins being set at sqrt(N events) is given little justification
>
> **AW2.1**: As written at lines 191 and 192 on page 4, we follow the convention of discrete-time SA (Haider et al., JMLR, 2020) to set the number of time bins to $\sqrt{N_e}$. Besides, we have examined different settings of time bins and present their results in Table 9 (on page 23). The results show that $\sqrt{N_e}$ could often lead to **better calibration** on five datasets.
>
> We use the above two parts to justify the setting of $\sqrt{N_e}$. To provide more explanations to justify the setting of $\sqrt{N_e}$, we have added a new subsection Appendix D.5 “Time Discretization Settings”. Revision details are elaborated in the final part “(3) Our revision”.
>
> **W2.2**: It also calls into question the comparison with other models (e.g. PatchGCN used 4 bins for all datasets in the original paper)
>
> **AW2.2**: In all comparisons, we strictly follow Patch-GCN’s experimental settings and do not make modifications to its settings, as written at lines 988 and 989 on page 19 (in our code as well). In other words, given that Patch-GCN used `4` bins for all datasets in its original paper, we also set `4` bins for Patch-GCN in baseline experiments.
>
> **W2.3**:  Additionally in the Appendix “Comparison with Hazard-Based Survival Modeling” 4 bins were used as opposed to the number used in the original experiments
>
> **AW2.3**: As mentioned in the first part “(1) The conventional settings of time bins in SA”, hazard-based survival modeling generally adopts `4` time bins (written at lines 1218-1220 on page 23), so we set the number of bins to `4` for hazard-based survival modeling.
>
> For incidence-based survival modeling, we set bins to $\sqrt{N_e}$, which is the same as that in our original experiments. Its results (Table 10) are also the same as those in the original experiments in Table 1.
>
> In summary, we always keep in mind to follow conventional settings in experiments and try our best to conduct fair experimental comparisons throughout this work.
>
> **W2.4**: The additional experiments in Appendix E don’t provide sufficient justification for using uniform sqrt(N events)
>
> **AW2.4**: To provide sufficient justification, we have conducted the additional experiment in which `time cutoff` is set to `Quantile` and `#Cuts` is set to `4`. Their results are as follows:
>
> **Table 9**: Different time discretization settings for VLSA. $N_{e}$ is the number of patients with event.
> | Time cutoff | #Cuts | TCGA-BLCA | TCGA-BRCA | TCGA-GBMLGG | TCGA-LUAD | TCGA-UCEC | Average CI | Average MAE | D-cal Count|
> |:--:|:--:|:--:|:--:|:--:|:--:|:--:|:--:|:--:|:--:|
> | Quantile | 4 | **0.6238** ($\pm$ 0.036) | 0.6483 ($\pm$ 0.060) | 0.7962 ($\pm$ 0.014) | 0.6240 ($\pm$ 0.028) | 0.7295 ($\pm$ 0.057) | 0.6844 | **24.40** | 0 |
> | Uniform | $N_{e}$ | 0.6176 ($\pm$ 0.025) | **0.6652** ($\pm$ 0.057) | **0.8002** ($\pm$ 0.010) | **0.6370** ($\pm$ 0.027) | **0.7571** ($\pm$ 0.045) | **0.6954** | 25.15 | **5** |
>
> Continue Down

---

> ### Author Response · Authors · 2024-11-26
> **Reply to Reviewer's Comments**
>
> Continue from Above
>
> From these additional results, we can find that using quantile 4 bins leads to **worse calibration** (`D-cal Count = 0`) than using uniform $\sqrt{N_e}$. Besides, from the complete Table 9 on page 26 of our revised paper, we can also observe that using uniform $\sqrt{N_e}$ performs better than other settings, especially in terms of **averaged CI** and **calibration**. Given these encouraging results, we adopt uniform $\sqrt{N_e}$ for our VLSA.
>
> **(3) Our revision**
>
> - Line 202 on page 4 of our revised paper: To address W2.1, we have added a sentence to guide readers to see our more justification regarding time discretization settings, as follows:
> > Refer to Appendix D.5 for more setting details and Appendix E for this setting's experiments.
>
> - On page 24 of our revised paper: To address W2.1 and W2.2, we have added a new subsection Appendix D.5 “Time Discretization Settings” to further clarify the settings of time bins used for all compared methods.
>
> - Line 1406 on page 27 of our revised paper: To address W2.3, we have improved our description to clearly describe the setting in “Comparison with Hazard-Based Survival Modeling”, as follows:
> > Following Zadeh & Schmid (2020), the settings of hazard-based survival modeling for VLSA are as follows…
>
> - Table 9 on page 26 of our revised paper: To address W2.4, we have added the above new results to this table, according to your valuable feedback.
>
> [Reference D] Zadeh & Schmid. Bias in cross-entropy-based training of deep survival networks. IEEE TPAMI, 2020.
>
> [Reference E] Chen et al. Whole slide images are 2d point clouds: Context-aware survival prediction using patch-based graph convolutional networks. MICCAI 2021.
>
> [Reference F] Chen et al. Multimodal Co-Attention Transformer for Survival Prediction in Gigapixel Whole Slide Images. ICCV 2021.
>
> [Reference G] Xu & Chen. Multimodal Optimal Transport-based Co-Attention Transformer with Global Structure Consistency for Survival Prediction. ICCV 2023.
>
> [Reference H] Zhou & Chen. Cross-modal translation and alignment for survival analysis. ICCV 2023.
>
> [Reference I] Zhang et al. Prototypical Information Bottlenecking and Disentangling for Multimodal Cancer Survival Prediction. ICLR 2024.
>
> [Reference J] Yu et al. Learning patient-specific cancer survival distributions as a se-quence of dependent regressors. NIPS 2011.
>
> [Reference K] Lee et al. DeepHit: A deep learning approach to survival analysis with competing risks. AAAI 2018.
>
> [Reference L] Haider et al. Effective ways to build and evaluate individual survival distributions. JMLR, 2020.
>
> [Reference M] Qi et al. SurvivalEVAL: A comprehensive open-source python package for evaluating individual survival distributions. AAAI Symposium 2023.
>
> [Reference N] Qi et al. An effective meaningful way to evaluate survival models. ICML 2023.

---

> ### Author Response · Authors · 2024-11-26
> **Reply to Reviewer's Comments**
>
> **W3**: PatchGCN was trained on 4 bins in the original paper. The VLSA model should be compared with the baselines with the different bin sizes being considered for a fair comparison.
>
> ---
>
> **AW3**: Thanks for your suggestion. We have conducted the additional experiment that the baseline Patch-GCN uses a different bin size, $\sqrt{N_e}$. Its results are shown as follows:
>
> **Table 10**: Comparison with Patch-GCN under the same number of time bins.
> | #Cuts | Method | TCGA-BLCA | TCGA-BRCA | TCGA-GBMLGG | TCGA-LUAD | TCGA-UCEC | Average CI | Average MAE | D-cal Count|
> |:--:|:--:|:--:|:--:|:--:|:--:|:--:|:--:|:--:|:--:|
> | 4 | Patch-GCN | **0.6124** ($\pm$ 0.031) | 0.6375 ($\pm$ 0.033) | 0.7999 ($\pm$ 0.021) | 0.5922 ($\pm$ 0.053) | 0.7212 ($\pm$ 0.025) | 0.6726 | 26.70 | **2** |
> | 4 | VLSA | **0.6124** ($\pm$ 0.020) | **0.6458** ($\pm$ 0.056) | **0.7907** ($\pm$ 0.010) | **0.6312** ($\pm$ 0.038) | **0.7674** ($\pm$ 0.041) | **0.6895** | **23.25** | 1 |
> | $N_{e}$ | Patch-GCN | 0.6054 ($\pm$ 0.049) | 0.6300 ($\pm$ 0.068) | 0.7890 ($\pm$ 0.020) | 0.5912 ($\pm$ 0.029) | 0.6967 ($\pm$ 0.030) | 0.6625 | 28.65 | **5** |
> | $N_{e}$ | VLSA | **0.6176** ($\pm$ 0.025) | **0.6652** ($\pm$ 0.057) | **0.8002** ($\pm$ 0.010) | **0.6370** ($\pm$ 0.027) | **0.7571** ($\pm$ 0.045) | **0.6954** | **25.15** | **5** |
>
> From these additional results, we can see that our VLSA model still often shows better performance than Patch-GCN even when Patch-GCN is trained on the same bin size, *i.e.*, 4 or $\sqrt{N_e}$.
>
> According to your suggestion, we have added the above results to our revised paper. Please refer to the new subsection “Comparison with Patch-GCN under the Same Number of Time Bins” in Appendix E, on page 26 of our revised paper.

---

> ### Author Response · Authors · 2024-11-26
> **Reply to Reviewer's Comments**
>
> **W4**: Given the variability of survival evaluation metrics (eg c-index) and the fairly marginal improvement over the baselines the authors should do a deeper analysis to showcase the improvements of their model. This could be done by plotting the survival curves of “high-risk” vs “low-risk” cases.
>
> ---
>
> **AW4**: Thanks for your insightful comment and valuable suggestion. We believe these would make this paper better.
>
> According to your suggestion, we have done a deeper analysis by plotting the survival curves of “high-risk” vs “low-risk” cases.
>
> The results have been presented in Figure 5 on page 25 of our revised paper. These results indicate that VLSA can discriminate between high- and low-risk patients with a significant difference (P-Value $<0.05$ by a log-rank test).
>
> Thanks again for your constructive feedback.

---

> ### Author Response · Authors · 2024-11-26
> **Reply to Reviewer's Comments**
>
> **W5**: In Appendix E, there appear to be patches comprising of mostly white-space. Was any preprocessing done to filter out patches like this, or patches with significant blur or artifacts.
>
> ---
>
> **AW5**: Thanks for your careful reading.
>
> In WSI preprocessing, we filter out the patches without any tissue or cell using the efficient OTSU algorithm. As a result, the patches with mostly white-space or significant blur or artifacts may not be filtered out in preprocessing.
>
> To address your concerns, we would like to provide our clarification, as follows:
> - In fact, we strictly follow a standard protocol in computational pathology, CLAM (Lu *et al.*, Nature Biomedical Engineering, 2021), to preprocess WSIs. CLAM does not include the functionality, *i.e.*, automatically filtering out the patches with mostly white-space or significant blur or artifacts.
> - From our understanding, doing the above could introduce large computational overheads, because i) almost each WSI contains hundreds of thousands of patches and ii) recognizing noisy patches would involve intensive computation on images.
>
> Nevertheless, we believe filtering out noisy patches could help to improve the performance. We would like to leave it as our future work.
>
> [Reference O] Lu et al. Data-efficient and weakly supervised computational pathology on whole-slide images. Nature Biomedical Engineering, 2021.

---

> ### Author Response · Authors · 2024-11-26
> **Reply to Reviewer's Comments**
>
> **W6**: Minor Comments: Spelling error at line 44 “While existing approaches have made exciting progress…” In Appendix Table 6, there seems to be little point in labeling the maximum values.
>
> ---
>
> **AW6**: Thanks for your kind reminder. We deeply appreciate your valuable feedback and good suggestions. We are sorry for these issues.
>
> To remedy these, we have corrected our spelling error at line 44: `exiting` -> `existing`. Moreover, we have removed the bold labeling on maximum values in Table 6, following your advice. These revisions can be seen in our revised paper.
>
> To avoid the similar errors you mentioned, we have carefully checked our writing throughout the paper and have revised them one by one.
>
> Thanks again for your efforts in reviewing.

---

> ### Author Response · Authors · 2024-12-03
>
> Thanks for your further feedback.
>
> We would like to express our gratitude to you for your recognition of our efforts put into revising this paper. All of these efforts made here are to address your concerns and questions, since this paper could NOT be improved if without your valuable suggestions and insightful comments.
>
> You mentioned that you have also reviewed the feedback from other reviewers and our replies to them. We would like to thank you once again for your careful and responsible reviewing. Fortunately, our point-to-point responses have basically addressed all other reviewers' concerns & questions. We sincerely hope our replies to you could do so, as well.
>
> Furthermore, we deeply appreciate your encouraging words above, plus your recognition of our meaningful contribution made to the field.
>
> Next, we provide a detailed point-to-point response to your further questions and concerns.
>
> ---
>
> **Q1**: However, the current structure of the paper makes it challenging to follow, and significant restructuring is necessary to better highlight the improvements the model achieves in few-shot contexts.
>
> ---
>
> **AQ1**: Thanks for your constructive comment.
>
> We understand you concerns about this paper’s structure organization for the few-shot relevant part. You mentioned that this paper needs to better highlight the few-shot relevant part by significant restructuring.
>
> To address your concerns, we would like to clarify that this paper actually intends to highlight **a new paradigm for WSI survival analysis**—Vision-Language Survival Analysis (VLSA)—as reflected by this paper’s title. Our detailed explanations are as follows:
>
> We **choose to highlight this new paradigm for WSI survival analysis**, because
> - previous studies **have not yet** adopted vision-language models (VLMs) for SA on WSIs;
> - this work is **the first study** that proposes a new approach to adapting VLMs to SA.
>
> Considering the above, the first idea coming to our mind is that we should let the public first know this new SA paradigm and emphasize it in the main paper.
>
> We **do not consider to highlight the few-shot relevant part**, because
> - few-shot learning is not this paper’s focus, but **is employed as an experimental scenario** to test the model’s performance when only a few samples can be used for training;
> - few-shot learning is **just one of the notable capabilities** in VLSA;
> - VLSA’s complete advantages contain **i**) better SA performance (Table 1), **ii**) good explainability & prediction transparency (Figure 2), **iii**) better few-shot learning performance, and **iv**) better model efficiency (Table 5);
>
> Therefore, in view of i) VLSA as a new paradigm for WSI survival analysis and ii) more than one highlight in VLSA, our idea of organizing this paper is
> - highlighting the VLSA itself in the paper and
> - designing different experiments to demonstrate its owned capabilities including few-shot learning.
>
> By contrast, if we choose to highlight the few-shot relevant part by significant restructuring, there would be **more possible concerns** raised accordingly, such as
> - **Is this approach specially designed for few-shot learning**? However, as aforementioned, few-shot learning is not this paper’s focus, but used as a test scenario to further examine the model’s performance. The new paradigm for SA on WSIs, *i.e.*, VLSA, is our focus.
> - I am more concerned with model efficiency, so **why do not the authors choose to highlight the model’s efficiency**? In fact, model efficiency is also one of the VLSA’s advantages.
> - I am more concerned with model explainability, so **why do not the authors choose to highlight the model’s explainability**? In fact, model explainability is also among the VLSA’s advantages.

---

> ### Author Response · Authors · 2024-12-03
>
> **Q2**: Additionally, the gains outside of the few-shot settings are marginal, which limits the overall impact of the work.
>
> ---
>
> **AQ2**: Thanks for your further feedback. To address your concerns, we would like to clarify **our gains outside of the few-shot settings**, as follows:
>
> **(1) Main results in Table 1**
>
> To our knowledge, compared with the gains obtained by previous studies, **our gains are comparable and competitive**. Our observations and evidence are as follows:
> - In Table 1 which shows the main comparative results, our gains in CI over runner-up are 0.52%, 2.77%, 0.03%, 2.49%, and 3.59% on BLCA, BRCA, GBMLGG, LUAD, and UCEC, respectively. Moreover, **our gain in average CI over runner-up is 2.28%**, larger than 1%.
> - Most of our gains above are comparable with those obtained by representative survival analysis works. For example,
> - Patch-GCN (Chen et al., MICCAI, 2021) shows a gain of **1.6% in average CI**, as written in Table 1 on page 8 of PatchGCN’s original paper;
> - MCAT (Chen et al., ICCV, 2021) shows a gain of **1.9% in average CI**, as written in Table 1 on page 7 of MCAT’s original paper.
>
> **(2) Results of ablation study in Table 2**
>
> Both the last two rows with “Prognostic texts” in Table 2 are proposed by us. They at least show
> - **2.87% gains in average CI** over the “Attention” baseline;
> - **3.46% gains in average CI** over the “Learnable prototypes” baseline.
>
> These gains are also comparable and competitive with those obtained by previous studies, as discussed above.
>
> **(3) Results of ablation study in Table 3**
>
> As written in our response to Reviewer f7nn ([the first half](https://openreview.net/forum?id=trj2Jq8riA&noteId=1CyeS23miO), [the second half](https://openreview.net/forum?id=trj2Jq8riA&noteId=04xvcZsGWa), [the latest one](https://openreview.net/forum?id=trj2Jq8riA&noteId=cBexgtXFUL)), the relatively-small gains in CI (Table 3) may be due to
> - the performance saturation possibly caused by our use of prognostic priors ($P$);
> - one of ordinal inductive biases, ordinal incidence function, often helps to improve MAE, not CI.
>
> Additionally, although the proposed two ordinal bias terms often show relatively-small improvements **in terms of CI**, they could often contribute to VLSA **in other more terms like MAE (Table 3) and the quality of survival prompts (Figure 3a)**.
>
> **(4) Evaluation of model efficiency in Table 5**
>
> Our VLSA has **the lowest cost** (**0.25G**) in model inference, apparently more efficient than the runner-up’s inference cost (**2.68G**).
>
> **(5) Summary**
>
> In summary, our gains are comparable and competitive in most cases, compared with those presented in previous studies.
>
> Most importantly, our approach shows superiority over existing survival analysis models **in many aspects**, such as
> - survival prediction,
> - few-shot learning (data efficiency),
> - model efficiency,
> - the explainability of survival prediction.
>
> The above listed advantages or merits **have the potential to broaden this work’s impact**. In addition, this work also could contribute to the field by **offering a solution to leverage LLMs for survival prediction**, as commented by the Reviewer f7nn.
>
> Once again, thanks for your consideration, as well as your valuable time and efforts in reviewing this paper.
>
> ---
>
> [Reference A] Chen et al. Whole slide images are 2d point clouds: Context-aware survival prediction using patch-based graph convolutional networks. MICCAI 2021.
>
> [Reference B] Chen et al. Multimodal Co-Attention Transformer for Survival Prediction in Gigapixel Whole Slide Images. ICCV 2021.

---

> ### Author Response · Authors · 2024-12-04
>
> Dear Reviewer `8wxk`,
>
> As we approach the end of the discussion period, we'd like to check if our further responses have addressed your concerns & questions.
>
> Overall, we had deep, effective, and happy discussions with the other reviewers in ICLR 2025:
> - Reviewer ` f7nn` mentioned that we have basically addressed most of his/her concerns; he/she acknowledges this work's contribution and have adjusted the score from **5 (conf. 4)** to **6 (conf. 4)**.
> - Reviewer `q9C6` mentioned that we have largely addressed his/her concerns; and have also raised the score from **5 (conf. 5)** to **6 (conf. 5)**.
>
> Hope we may have that discussions with you, too.
>
> We thank your again for your valuable time and insightful feedback.
>
> The Authors of Submission 631

---

### Author Response · Authors · 2024-11-26
**Reply to All Reviewers**

Dear Reviewers,

We would like to thank you for your valuable time and efforts in reviewing this paper. We deeply appreciate all your helpful and insightful suggestions. This paper cannot be improved and better if without your detailed feedback.

To address your concerns, **we have uploaded our revised paper**. Please check this revised version and let us know if there are further questions. We would be more than happy to reply to them.

We summarize our main revisions as follows:
- Discussion on the comparison with multi-modal survival models (page 4);
- Additional justification (page 24) and experimental results (Table 9 and 10 on page 26) for time discretization settings;
- Additional experimental results for a new baseline (Table 1 on page 7; Table 4 and 5 on page 10);
- Discussion on the performance of two inductive biases (page 9);
- More elucidations for the few-shot survival prediction experiments that verify models' data efficiency (page 10);
- Discussion on the state-of-the-art methods for WSI classification (page 11);
- Additional results of risk grouping and Kaplan-Meier analysis (page 24 and Figure 5 on page 25).

All revisions are colored in red. We hope our revisions and responses could address your concerns well.

Thanks again for the time and effort you have dedicated as a reviewer for ICLR 2025.

---

> ### Public Comment · ~Pei_Liu5 · 2025-02-12
> **Overview of our Camera Ready Revision**
>
> Dear PCs, SACs, ACs & Reviewers,
>
> We'd like to thank you again for your efforts in reviewing this paper. Based on the suggestions from Reviewers & ACs, we have carefully revised this paper and uploaded its camera-ready PDF.
>
> To let you know the changes made in this version, we summarize our major revisions over the paper revised in the rebuttal, as follows:
> - Presentation and Clarity Improvements
>   - On page 10: the title of Subsection 4.5 has been changed to "Analysis of Data Efficiency (Few-Shot Survival Analysis)" to clarify the experiments on data efficiency evaluation (mentioned by Reviewer `8wxk`).
>   - Previous Subsection 4.6 "Analysis of Computation Efficiency" and Section 5 "Limitation and Future Work" have been moved to the Appendix, due to the limit of 10 pages for the main paper.
>   - On page 16: we have added a table of contents for the Appendix of this paper, to let the readers have an overview of our additional results and further enhance the presentation.
>   - Other minor changes throughout this paper to improve the representation and clarity (mentioned by Reviewer `8wxk` & ACs).
> - Experiments
>   - Table 1 on page 7: the results of the baseline $\text{ABMIL}_{\text{Prompt}}$ are added (mentioned by Reviewer `f7nn`).
>   - Table 3 on page 9: the results of the VLSA model without language priors (previously shown in Table 8 on page 26) are added (mentioned by Reviewer `f7nn`).
>   - On page 9: the discussions on the marginal CI gains are revised according to the additional feedback from Reviewer `f7nn` (also mentioned by Reviewer `8wxk`, and ACs).
>   - Table 4 on page 10: the results of the baseline MI-FewShot are added (mentioned by Reviewer `f7nn`).
>   - On pages 26 & 27: Appendix D.5 "Comparison with Multi-Modal Survival Analysis Methods" is added to present the additional results of the comparisons with multi-modal SA methods (mentioned by Reviewer `f7nn`).
>
> Thank you again for the invaluable time and effort you have dedicated as the Program Committee for ICLR 2025.
>
> The Authors of Submission 631

---

### Meta-Review · Area_Chair_xWTG · 2024-12-20

**Metareview:**

First, there appears to be a misalignment between your paper's stated contribution and its actual focus.
Second, critical methodological choices lack sufficient justification.
Third, the experimental validation needs substantial strengthening.
Fourth, the ordinal survival prompt design raises concerns about its handling of censored cases and multiple events, suggesting that the core methodology may need refinement.

**Additional Comments On Reviewer Discussion:**

The paper received mixed reviews and the authors were able to address some of the concerns raised by the reviewers. While there is no final consensus, the AC acknowledges both the merits outlined by the positive reviewers & shortcomings of the paper.
The authors are suggested to provide a revision in final version to clarify main issues raised by reviewers, i.e.,  substantial improvements in its presentation and clarity.

---

### Decision · Program_Chairs · 2025-01-22

Accept (Poster)